# Spatial dysregulation of T follicular helper cells impairs vaccine responses in aging

Alyssa Silva-Cayetano [1], Sigrid Fra-Bido[1], Philippe A. Robert [2,12], Silvia Innocentin[1], Alice R. Burton[1], Emily M. Watson [1], Jia Le Lee [1], Louise M. C. Webb[1], William S. Foster [1], Ross C. J. McKenzie [1], Alexandre Bignon[1], Ine Vanderleyden[1], Dominik Alterauge[3], Julia P. Lemos[4,5,13], Edward J. Carr[1,6,7], Danika L. Hill[1,8], Isabella Cinti[9], Karl Balabanian[4,5], Dirk Baumjohann [3,10], Marion Espeli[4,5], Michael Meyer-Hermann [2,11], Alice E. Denton [9] & Michelle A. Linterman [1] ✉

The magnitude and quality of the germinal center (GC) response decline with age, resulting in poor vaccine-induced immunity in older individuals. A functional GC requires the co-ordination of multiple cell types across time and space, in particular across its two functionally distinct compartments: the light and dark zones. In aged mice, there is CXCR4-mediated mislocalization of T follicular helper ($T_{FH}$) cells to the dark zone and a compressed network of follicular dendritic cells (FDCs) in the light zone. Here we show that $T_{FH}$ cell localization is critical for the quality of the antibody response and for the expansion of the FDC network upon immunization. The smaller GC and compressed FDC network in aged mice were corrected by provision of $T_{FH}$ cells that colocalize with FDCs using CXCR5. This demonstrates that the age-dependent defects in the GC response are reversible and shows that $T_{FH}$ cells support stromal cell responses to vaccines.

T cell regulation of antibody-mediated immunity is critical for health, as an enduring antibody response after vaccination or infection can generate protective immunity against subsequent infections. However, there are members of our society who are less able to generate high-titer antibody responses upon vaccination, the largest cohort being older people[1]. This age-dependent deficit in antibody production has been evident in the global COVID-19 vaccine rollout as, despite the success of these vaccines, older people generate lower antibody titers than younger persons[2–8]. Furthermore, because antibody titers decrease over time[7,8], this culminates in a faster reduction in vaccine efficacy in older individuals[9]. This has been known for decades, but the underlying mechanism(s) remains unclear. Antibody production upon vaccination or infection can occur via two cellular pathways: the extrafollicular response, which produces an initial burst of antibodies

[1]Immunology Program, Babraham Institute, Cambridge, UK. [2]Department of Systems Immunology and Braunschweig Integrated Centre of Systems Biology, Helmholtz Centre for Infection Research, Braunschweig, Germany. [3]Institute for Immunology, Faculty of Medicine, Biomedical Center, LMU Munich, Munich, Germany. [4]Université Paris Cité, Institut de Recherche Saint Louis, EMiLy, INSERM U1160, Paris, France. [5]OPALE Carnot Institute, The Organization for Partnerships in Leukemia, Hôpital Saint-Louis, Paris, France. [6]Department of Medicine, University of Cambridge, Cambridge, UK. [7]The Francis Crick Institute, London, UK. [8]Department of Immunology and Pathology, Monash University, Melbourne, Victoria, Australia. [9]Department of Immunology and Inflammation, Imperial College London, London, UK. [10]Medical Clinic III for Oncology, Hematology, Immuno-Oncology and Rheumatology, University Hospital Bonn, University of Bonn, Bonn, Germany. [11]Institute for Biochemistry, Biotechnology and Bioinformatics, Technische Universität Braunschweig, Braunschweig, Germany. [12]Present address: Translational Immunology, Department of Biomedicine, University of Basel, Basel, Switzerland. [13]Present address: Sorbonne Université, INSERM, Institut de Myologie, Centre de Recherche en Myologie, Paris, France. ✉e-mail: michelle.linterman@babraham.ac.uk

early after antigenic challenge[10]; or the GC reaction, which produces memory B cells and high-affinity plasma cells with somatically mutated immunoglobulin genes that can persist long-term[11]. In aged mice, the extrafollicular response is intact but the formation of GCs is delayed and, once formed, is of reduced size at its peak compared with younger animals[12–17]. A diminished GC reaction is also observed in reactive lymph nodes (LNs) from older people compared with younger adults[18] and this is reflected by circulating biomarkers of an ongoing GC response after vaccination in older people[19]. Therefore, understanding why the size of the GC, and its output, is impaired in aging is key to determining why older people are less capable of mounting persistent antibody responses to vaccines.

The GC is polarized into two functionally distinct regions known as the light and dark zones. GC B cells localize to the dark zone via expression of CXCR4, which facilitates their migration to the CXCL12-producing reticular stromal cells[20,21]. Here, GC B cells proliferate and the genes encoding the B cell receptor undergo somatic hypermutation, then selection is required to test whether the introduction of mutations has impacted the function or specificity of the B cell receptor. To undergo selection in the light zone, GC B cells exit the cell cycle and downregulate CXCR4, enabling CXCR5-dependent migration towards the CXCL13-rich FDC stromal network that defines the light zone. In the light zone, a functional B cell receptor enables GC B cells to collect antigen and present it to $T_{FH}$ cells[22], those that successfully engage a $T_{FH}$ cell by presenting cognate antigen receive help, in the form of CD40L-dependent costimulation and cytokines. This protects B cells from death and induces cMyc expression, which enables re-entry into the cell cycle and promotes survival[11]. These GC B cells will then migrate back to the dark zone and either undergo further rounds of proliferation and mutation or exit the GC as fully differentiated memory B cells or long-lived antibody-secreting cells (ASCs).

Here, we investigated the mechanistic reasons for the suboptimal GC response in aged mice. We showed that the spatial organization of the GC is changed in aging; the FDC network failed to expand in the GC after immunization, leading to a smaller light zone, and $T_{FH}$ cells were dispersed throughout the GC, rather than being polarized to the light zone. In silico modeling, in vivo vaccination studies using genetically modified mice, human vaccination cohorts and cell transfers into aged mice were combined to demonstrate that the mislocalization of $T_{FH}$ cells is a main driver of the smaller and poorer quality GC response in aging. In our endeavor to understand how the GC response changes with age, this study identified a role for $T_{FH}$ cells supporting the FDC response to vaccination.

## Results

### GC magnitude is impaired with age in a B cell extrinsic way
The proportion and absolute number of GC B cells were diminished at the peak of the GC response in both aged (90–108-week-old) BALB/c and C57BL/6 mice compared with younger adult (8–12-week-old) mice

(Extended Data Figs. 1 and 2), consistent with previous reports[13–15,23]. GC formation and maintenance relies on $T_{FH}$ cells, which provide several B cell-supporting cues necessary throughout the GC reaction[24]. The proportion of $T_{FH}$ cells was increased with age throughout the GC reaction in both BALB/c and C57BL/6 mice, but the absolute number of $T_{FH}$ cells was diminished in aged mice of both strains (Extended Data Figs. 1 and 2). Comparative quantitative imaging of draining inguinal lymph node (iLN) sections showed that aged mice generated significantly fewer GCs 14 d postimmunization than adult mice (Fig. 1a–c, Supplementary Fig. 1 and Extended Data Fig. 3). This confirms that aging is associated with a poor GC response upon vaccination in two genetically different mouse strains (BALB/c and C57BL/6); however, the causal mechanism is unknown.

Accumulation of T follicular regulatory cells, which are key negative regulators of the GC response, has been postulated to contribute to poor GC responses in aged mice[25,26], yet we found no significant differences in the proportion of T follicular regulatory cells and a reduction in their total number with age (Extended Data Fig. 4a–c). Furthermore, genetically halving the number of T follicular regulatory cells[27] did not enhance the magnitude of the GC response in aged $Foxp3^{Cre/+}Cxcr5^{fl/fl}$ mice (Extended Data Fig. 4d–i), indicating that the reduced GC magnitude in aging is not caused by increased T follicular regulatory cell number.

To address whether the observed defects in GC formation could be driven by age-associated B cell-intrinsic effects, we used an adoptive transfer system of $SW_{HEL}$ B cells, which carry antigen specificity for hen egg lysozyme (HEL)[28], from either young adult or aged $SW_{HEL}$ mice into young adult mice (Fig. 1d). At 10 d after immunization, both the proportion and total number of GC B cells derived from aged $SW_{HEL}$ donors were comparable to those derived from younger donors (Fig. 1e–g and gating strategy in Supplementary Fig. 2), implicating B cell extrinsic factors as contributors to the diminished GC formation and magnitude with age. We confirmed that there were no differences in the frequency and total number of transferred HEL-binding B cells and $T_{FH}$ cells derived from transferred OT-II cells from young mice between the two recipient groups (Supplementary Fig. 2). These data show that there are defects in GC formation and magnitude with age that cannot be rescued through the reduction of T follicular regulatory cell numbers and could not be accounted for by intrinsic defects of B cells in aging.

### Aging impairs GC selection and affinity maturation
GC B cells increase their affinity for antigen by undergoing somatic hypermutation and selection. Sequencing of the $V_H186.2$ heavy chain region of 4-hydroxy-3-nitrophenyl acetyl (NP)-specific GC B cells revealed that there was a near twofold reduction in the frequency of the affinity-enhancing W33L mutation in aged mice compared with younger mice (Fig. 1h). In vivo assessment of positive selection of NP-specific B cells (B1-8i-Tg) transferred from a younger adult mouse into either young or aged mice (Fig. 1i and Supplementary Fig. 2) revealed there was a lower frequency of cMyc+ GC B cells in aged recipient mice compared

**Fig. 1 | The GC response, and its output, is diminished in aged mice.**
**a**, Representative confocal images of GCs at ×40 from adult and aged BALB/c mice 14 d after immunization with NP-KLH in alum; scale bars, 50 μm. LN sections were stained for IgD (green), CD35 (white), Ki67 (blue) and CD3e (magenta).
**b**, Enumeration of GCs per LN. **c**, Quantification of the total GC area (n = 16). **d**, Experimental outline of the cotransfer of $SW_{HEL}$ B cells from either adult or aged donors alongside OT-II T cells from adult donors into adult C57BL/6 recipient mice in which GC formation was analyzed 10 d after immunization with HEL-OVA in alum. **e**, Representative flow cytometry plots identifying $SW_{HEL}$-derived GC B cells (CD95+CD38−CD45.1+B220+HEL+) in recipient mice. The values next to the gates indicate the population percentage. **f,g**, Quantification of the frequency (**f**) and total number (**g**) of $SW_{HEL}$-derived GC B cells (n = 12). **h**, Pie charts indicating the frequency of the affinity-inducing mutation W33L in the CDR1 region of $V_H186.2$ sequenced from single-cell sorted NP+IgG1+ GC B cells of adult and aged C57BL/6 mice 21 d postimmunization with 1W1K-NP/alum. The values in the chart center

indicate the total number of cells sequenced per group (n = 16). **i**, Experimental outline of the transfer of B1.8i $myc^{GFP/GFP}$ B cells from adult donors into adult or aged C57BL/6 recipient mice in which GC formation was analyzed 10 d after NP-OVA in alum immunization. **j**, Quantification of the frequency of B1.8i-derived cMyc+ GC B cells in adult or aged mice; data are pooled from two independent experiments, first experiment in black, second experiment in white (n = 23). **k,l**, Representative ELISpot well images (left) and quantification (right) of bone marrow NP23-(**k**) and NP2- (**l**) specific IgG1 ASCs in BALB/c mice 21 d after immunization with NP-KLH in alum. **m**, Affinity maturation of bone marrow ASCs from BALB/c mice as determined by the ratio of NP2/NP23-specific ASCs (n = 15). For all experiments, 2–4 experimental repeats were performed with biologically independent samples. In bar graphs, each symbol represents a mouse, and the bar height represents the median. The P values were generated by performing an unpaired two-tailed Mann–Whitney U test. ELISpot, enzyme-linked immunosorbent spot; s.c., subcutaneous; wo, weeks old.

with younger recipients (Fig. 1j and Supplementary Fig. 2). In both aged BALB/c (Fig. 1k–m) and C57BL/6 (Extended Data Fig. 2) mice there was a clear reduction in the number of IgG1-secreting plasma cells specific

for both NP23 and NP2 in the bone marrow, as well as a reduction in the ratio of high-affinity NP2-binding to NP23-binding plasma cells, compared with younger adult mice. This was accompanied by a reduction

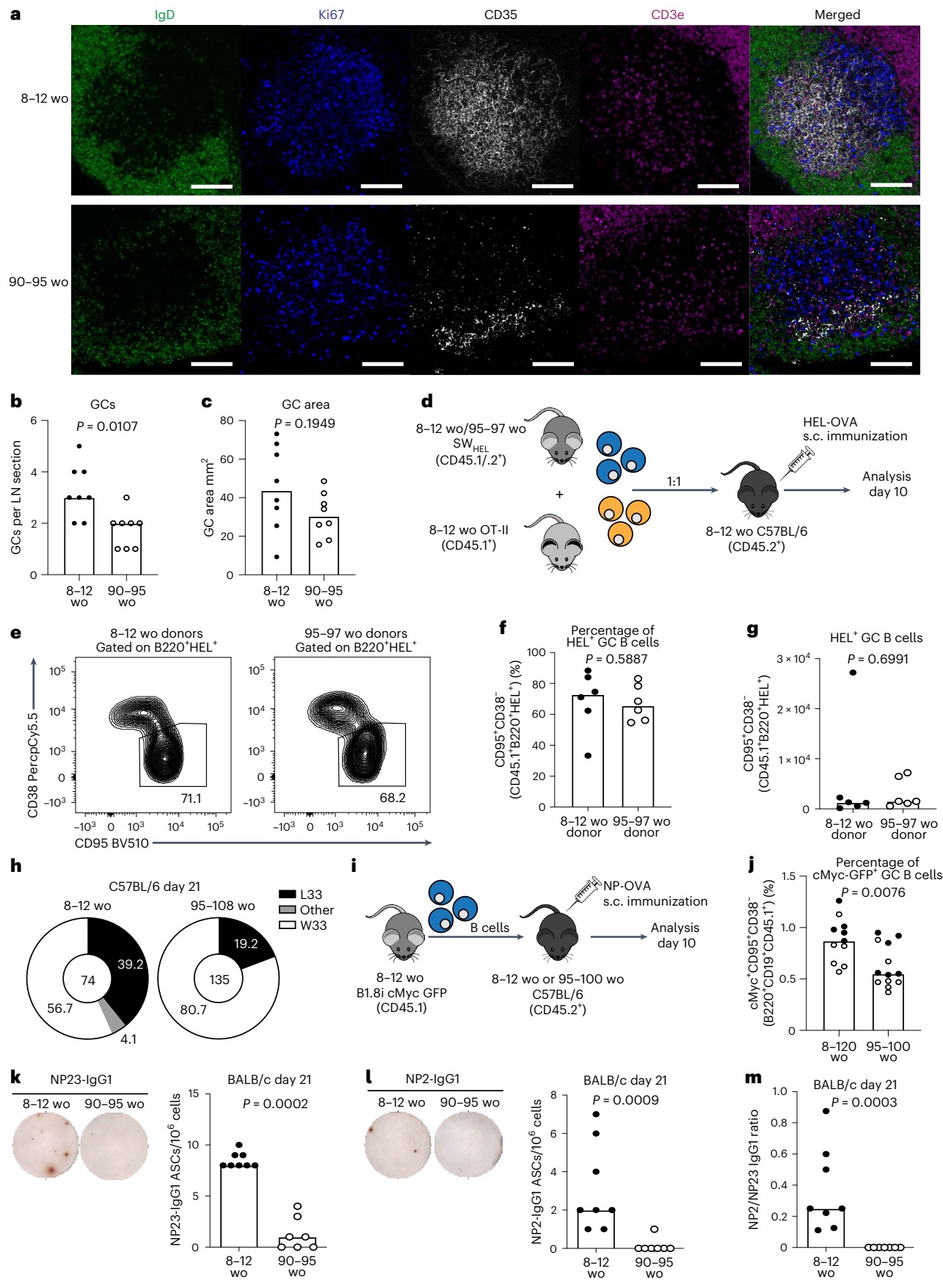

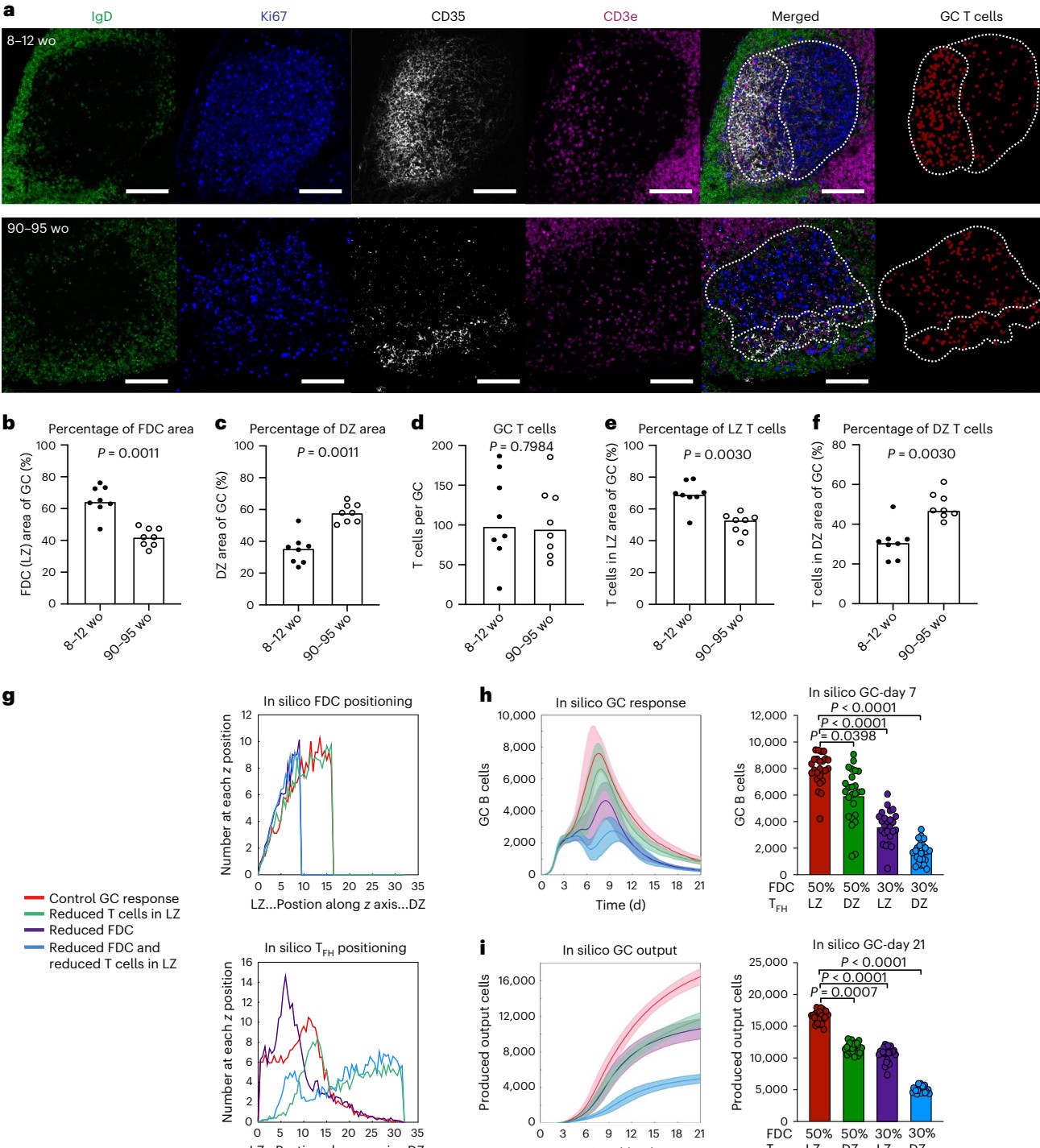

**Fig. 2 | The spatial organization of the GC is altered in aged mice.**
**a**, Representative confocal images of GCs at ×20 magnification from adult and aged BALB/c mice 14 d after immunization with NP-KLH in alum; scale bars, 100 μm. LN sections were stained for IgD (green), CD35 (white), Ki67 (blue) and CD3e (magenta). Representative masked images identifying GC T cells (red) generated by Cell Profiler used for enumeration (right panel). **b–f**, Quantification of the CD35⁺ FDC network light zone area (**b**), the Ki67⁺CD35⁻ dark zone area (**c**) of the GC, the number of CD3⁺ T cells (**d**) within the GC area, and the proportion of T cells positioned in the CD35⁺ FDC light zone area (**e**) and Ki67⁺CD35⁻ dark zone (**f**) of the GC. For **b–f**, the data are representative of four independent experiments ($n = 16$) where each symbol on the graph represents a mouse and the bar height represents the median. The $P$ values were generated by performing an unpaired, two-tailed Mann–Whitney $U$ test. **g**, Computational modeling of the age-associated changes to the spatial organization of the GC (control GC

response in red, GC response with reduced $T_{FH}$ cell positioning in the light zone in green, GC response with reduced FDC network in purple and the combined effect of both defects in blue). Graphical representation of the FDC network size (top) and $T_{FH}$ cell positioning (bottom) within the GC compartment for the simulations performed. The units on $x$ axes of plots are the positions of cells along the $z$ axis in ×10 μm. **h**, Computational modeling of the impact on the number of GC B cells (left) with quantification at day 7 of the GC response (right). **i**, Computational modeling of the impact on the total number of produced output cells (left) with quantification at day 21 of the GC response (right). For **h** and **i**, lines on time-course graphs show the mean of 25 independent simulations and the shaded areas indicate the standard deviation. For bar graphs, the bar height represents the mean ($n = 25$). The $P$ values indicated on the graphs were generated by performing a Kruskal–Wallis test with Dunn's multiple comparison correction.

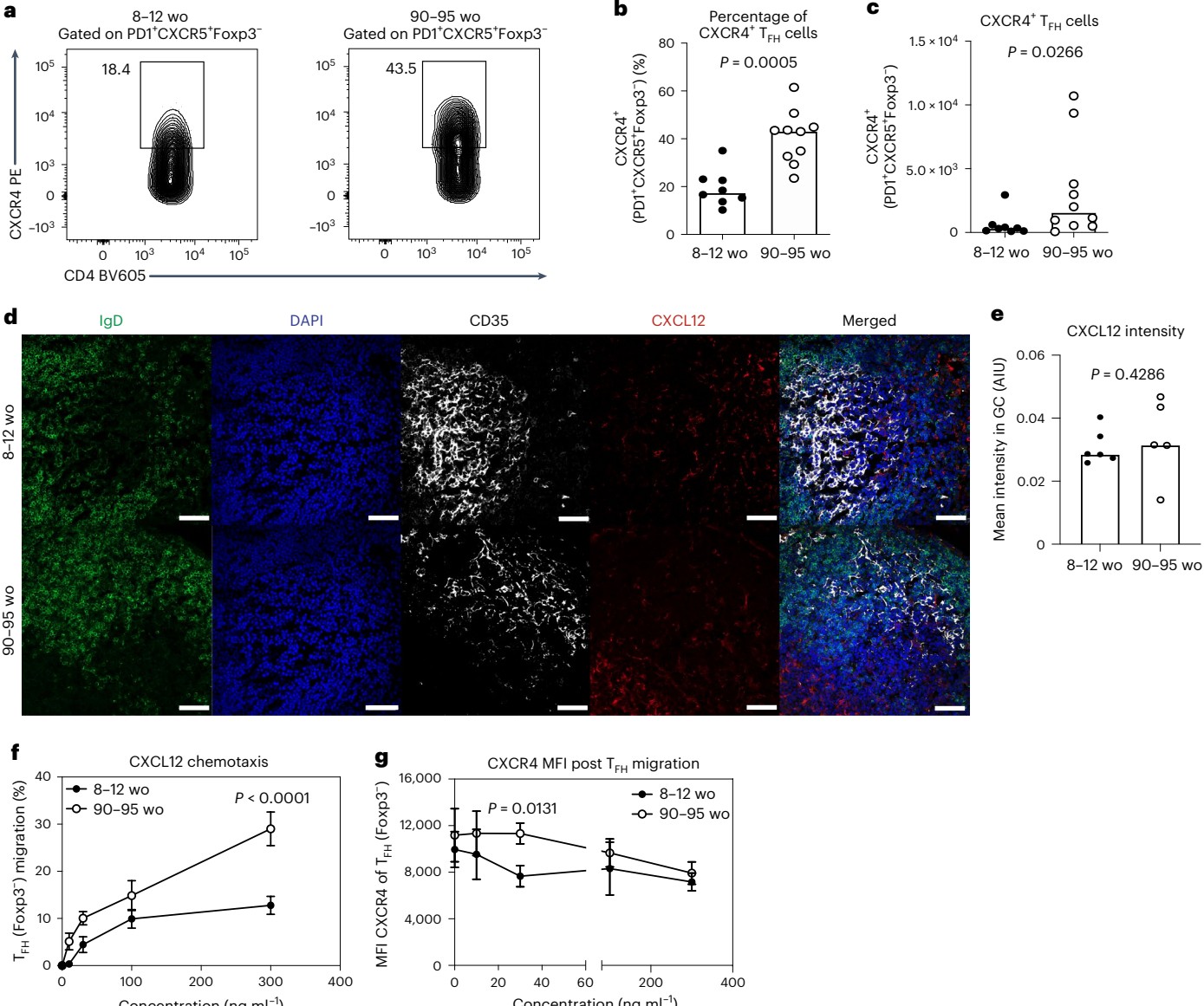

**Fig. 3 | CXCR4 expression is increased in $T_{FH}$ cells from aged mice.**
**a**, Representative flow cytometry plots showing CXCR4$^+$PD1$^+$CXCR5$^+$Foxp3$^-$ $T_{FH}$ cells in adult and aged BALB/c mice 14 d after immunization with NP-KLH in alum. Values adjacent to gates represent percentages. **b,c**, Quantification of the percentage (**b**) and total number (**c**) of CXCR4$^+$PD1$^+$CXCR5$^+$Foxp3$^-$ $T_{FH}$ cells in adult and aged BALB/c mice ($n = 18$). Data are representative of two independent experiments. **d,e**, Representative ×40 confocal images (**d**) and quantification (**e**) of CXCL12 in red within the dark zone of GCs in the iLNs of adult and aged mice at day 14 postimmunization with NP-KLH in alum; IgD (green), DAPI (blue) and CD35 (white). AIU, arbitrary intensity units. $n = 11$ biologically independent samples.

Scale bars, 50 μm. Bar heights represent the median and $P$ values were obtained by performing an unpaired, two-tailed Mann–Whitney $U$ test. Each symbol represents a single mouse. **f,g**, CXCL12 chemotaxis assays with $T_{FH}$ cells isolated 14 d after NP-KLH in alum immunization. **f,g**, Percentage of PD1$^+$CXCR5$^+$Foxp3$^-$ $T_{FH}$ cells of the total input cells that migrated to the indicated concentrations of CXCL12 (**f**) and median fluorescence intensity (MFI) of cell-surface CXCR4 expression (**g**) ($n = 16$). Each symbol represents the mean ± s.d. and $P$ values are from two-way ANOVA with Sidak's multiple comparisons test. Data are representative of two independent experiments performed with biologically independent samples. PE, phycoerythrin.

in the NP-specific antibody titers and reduced affinity maturation of serum antibodies of both aged BALB/c and C57BL/6 mice (Extended Data Figs. 1 and 2), consistent with previous reports[15,29,30]. These data show that both the magnitude and the quality of the GC are impaired with age and that B cells from younger adult donors do not receive positive selection signals in the aged GC as well as in younger animals.

**$T_{FH}$ cell positioning and FDC expansion are altered in aging**
Correct structural organization of the GC allows multiple cell types to interact at the right place and at the right time. Yet, the impact of aging on the spatial organization of the GC remains unexplored.

Quantitative confocal imaging of iLN sections from adult and aged mice (Fig. 2a and Supplementary Fig. 1) revealed that the area of the mesenchyme-derived FDC network within GCs of aged mice was significantly reduced compared with younger adult mice, resulting in larger dark zone areas and altered GC structure (Fig. 2b,c). The reduced FDC area was also observed in aged mice before vaccination (Extended Data Fig. 3). Despite this, the proportion of centroblasts and centrocytes as quantified by flow cytometry was not altered with age (Extended Data Fig. 1), suggesting that the structural changes observed in the GC zones are caused by the stromal compartment. The number of $T_{FH}$ cells per GC was comparable between adult and aged BALB/c and C57BL/6 mice

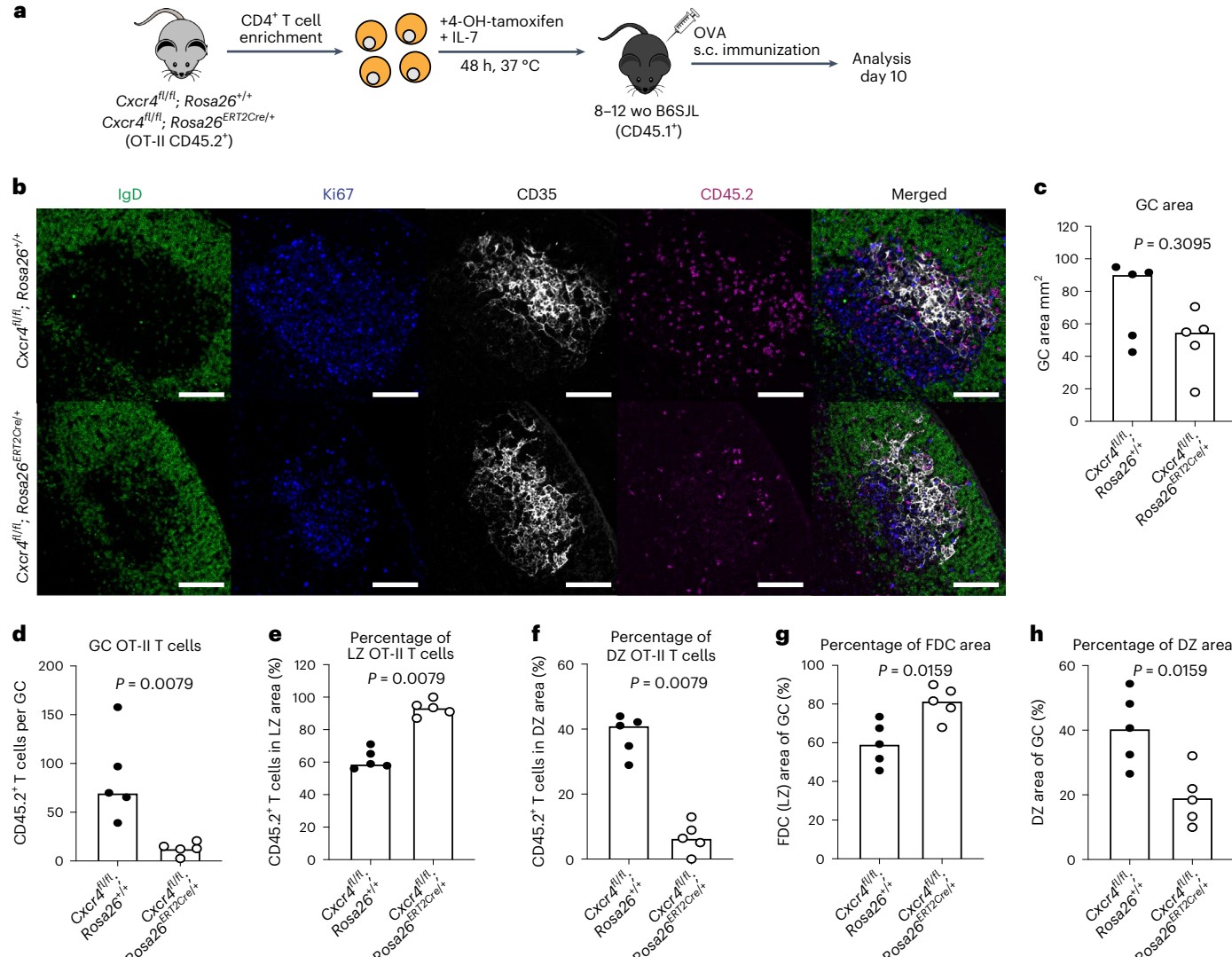

**Fig. 4 | CXCR4 expression determines T cell dark zone positioning.**
**a**, Experimental outline of in vitro 4-OH-tamoxifen treatment of CD4+ T cells isolated from $Cxcr4^{fl/fl};Rosa26^{ERT2Cre/+}$ OT-II mice that were treated for 48 h, after which the cells were transferred into adult B6SJL recipient mice. Recipient mice were immunized subcutaneously with OVA in alum and analysis was performed after 10 d. **b**, Representative ×20 magnification confocal images of the GCs from the iLNs of B6SJL mice that received tamoxifen-treated OT-II cells from either $Cxcr4^{fl/fl};Rosa26^{+/+}$ or $Cxcr4^{fl/fl};Rosa26^{ERT2Cre/+}$ mice; scale bars, 100 µm. LN sections were stained for IgD (green), CD35 (white), Ki67 (blue) and CD45.2 (magenta).

**c–h**, Quantification of the GC area (**c**), the number of CD45.2+ transferred cells in the GC (**d**), percentages of OT-II $T_{FH}$ cells in the CD35+ FDC light zone area (**e**) and Ki67+CD35– dark zone area (**f**), percentage of the GC occupied by the CD35+ FDC network (**g**) and percentage of the GC occupied by the dark zone (**h**), from the iLNs of recipient B6SJL mice ($n = 10$). Data are representative of two independent experiments performed with biologically independent samples. In graphs, bar heights represent the median and P values were obtained by performing an unpaired, two-tailed Mann–Whitney $U$ test. Each symbol represents a single mouse. LZ, light zone. DZ, dark zone.

(Fig. 2d), but in GCs from aged mice there was a lower proportion of $T_{FH}$ cells localizing to the light zone (Fig. 2e and Extended Data Fig. 3). Instead, a greater proportion of $T_{FH}$ cells were seen in the dark zone of aged mice (Fig. 2f). Thus, aging alters the structure of the FDC network and the spatial distribution of $T_{FH}$ cells. The potential outcomes of this on the GC response are twofold: (1) the reduced light zone area would reduce the amount of antigen presented to B cells and (2) the $T_{FH}$ cells are not closely associated with B cells that are collecting and presenting antigen in the light zone, thereby limiting the provision of $T_{FH}$ cell help to B cells.

To disentangle the contribution of the disrupted light zone size from $T_{FH}$ cell localization on the GC response, and to establish whether the observed spatial changes could be causative of the poor GC responses observed with age, we performed three-dimensional spatial

computational simulations[31,32] of the GC. These simulations predicted that the number of GC B cells generated would be significantly impaired by either a reduction of $T_{FH}$ cells in the light zone or a reduction in the size of the FDC network, and even more so by the combined effect of both changes (Fig. 2g,h). Similarly, the cellular output (a combined metric of memory and plasma cells generated) of the GC was predicted to be significantly diminished by day 21 in response to the structural changes simulated, and this effect was further exacerbated by both defects simulated in combination (Fig. 2i). The simulations show that the impaired light zone area and $T_{FH}$ localization could explain the poor vaccine response in aged mice and support the hypothesis that impaired FDC expansion and aberrant $T_{FH}$ positioning synergistically contribute to the poor GC response in aging. Yet, the molecular cause(s) of these changes in spatial organization with age are unknown.

## Enhanced CXCR4 localizes $T_{FH}$ cells to the dark zone

$T_{FH}$ cells preferentially localize to the light zone area of the GC using CXCR5 expression to migrate to FDC-expressed CXCL13 (ref. 33). The movement of GC B cells from the light to the dark zone is controlled by CXCR4-dependent migration to the CXCL12-expressing stromal cells in the dark zone[20,21]. A proportion of $T_{FH}$ cells located within GCs have been reported to express CXCR4 (ref. 34), and consistent with this report we observed that GC $T_{FH}$ cells marked by S1PR2 and low levels of CD90 have higher expression of CXCR4 than non-GC $T_{FH}$ cells (Extended Data Fig. 5). We hypothesized that CXCR4 may function to localize $T_{FH}$ cells to the dark zone and this process may be altered in aging.

Flow cytometric analysis revealed that $T_{FH}$ cells from aged mice had more CXCR4+ $T_{FH}$ cells than younger adult mice (Fig. 3a–c), while CXCR5 expression on PD-1+Bcl6+CD4+Foxp3− $T_{FH}$ cells was unaltered by age (Extended Data Fig. 5). This was also observed after immunization with an adenoviral vectored COVID-19 vaccine and after influenza A virus infection (Extended Data Fig. 5). Enhanced CXCR4 expression was likewise seen in older people, with antigen-experienced CD4+ T cells from unvaccinated people over 65 yr of age having increased surface CXCR4 expression compared with 18–36-yr-old adults (Extended Data Fig. 5). The expression of CXCR4 was also higher on circulating $T_{FH}$-like cells from older people 7 d after seasonal influenza vaccination compared with younger individuals, with CXCR5 expression being consistent between the age groups (Extended Data Fig. 5). Immunofluorescence staining indicated that there was comparable expression of CXCL12 within the GCs of adult and aged mice (Fig. 3d,e), suggesting that the increased dark zone positioning of $T_{FH}$ cells with age is likely due to an increase in the expression of CXCR4 rather than increased ligand availability in the GC dark zone. Consistent with this, the increased CXCR4 expression on $T_{FH}$ cells from aged mice was associated with enhanced chemotaxis towards CXCL12 in an in vitro Transwell assay (Fig. 3f). In the chemotaxis assays, $T_{FH}$ cells from younger mice downregulated CXCR4 after migrating to CXCL12, but this did not occur to the same extent in $T_{FH}$ cells from aged mice (Fig. 3g), suggesting impaired ligand-dependent internalization as the cause of enhanced CXCR4 surface expression in aging.

To test the functional role of CXCR4 expression in $T_{FH}$ localization, we used an adoptive transfer system where T cells cannot express this receptor and assessed whether the $T_{FH}$ cells would still localize in the dark zone. Ovalbumin (OVA)-specific (OT-II) T cells from $Cxcr4^{fl/fl}$; $Rosa26^{ERT2Cre/+}$ mice were isolated and treated with tamoxifen in vitro to induce deletion of $Cxcr4$, before being transferred into congenically distinct adult B6SJL mice (Fig. 4a). Deletion of $Cxcr4$ on CD4+ T cells did not alter early T cell division after activation, nor did it affect $T_{FH}$ cell differentiation (Extended Data Fig. 6). At 10 d after immunization, GCs had a nonsignificant trend to be smaller in recipients of CXCR4-deficient T cells (Fig. 4b,c). CXCR4-deficient $T_{FH}$ cells were, however, fewer in number and also enriched in the light zone, with few cells being present in the dark zone (Fig. 4d–f and Supplementary Fig. 3), showing that CXCR4 expression is necessary for dark zone positioning of $T_{FH}$ cells.

These data demonstrate that $T_{FH}$ cell localization to the dark zone is actively controlled by CXCR4 expression, which is dysregulated in aging. Surprisingly, we found that the enrichment of $T_{FH}$ cells in the light zone increased the size of the FDC network in GCs of recipient mice, with a corresponding decrease in the dark zone (Fig. 4g,h), suggesting that T cell localization may influence the GC stroma in response to vaccination, independent of the size of the GC.

## $T_{FH}$ cells in the light zone promote FDC expansion

To investigate how $T_{FH}$ cell polarization to the light zone influences GC size and output, we immunized younger adult $Cxcr4^{fl/fl}$; $Cd4^{Cre/+}$ mice, which give rise to CXCR4-deficient $T_{FH}$ cells (Supplementary Fig. 4). At 10 d after immunization, imaging showed that GC size and $T_{FH}$ cell numbers were comparable in $Cxcr4^{fl/fl}$; $Cd4^{cre/+}$ mice and their littermate controls (Fig. 5a–c). Flow cytometric analysis showed that the percentage and total number of GC B cells and $T_{FH}$ cells were comparable between $Cxcr4^{fl/fl}$; $Cd4^{Cre/+}$ and control mice throughout the GC response at days 10, 21 and 35 postimmunizations (Supplementary Fig. 4). Despite normal numbers of GC $T_{FH}$ cells, the majority of CXCR4-deficient $T_{FH}$ cells were localized to the light zone, with less than 10% of cells in the dark zone (Fig. 5d,e and Supplementary Fig. 5). Consistent with the CXCR4-deficient OT-II transfer data (Fig. 4), we found that the enrichment of $T_{FH}$ cells in the light zone increased the size of the FDC network in GCs of $Cxcr4^{fl/fl}$; $Cd4^{Cre/+}$ mice, with a concomitant decrease in dark zone area compared with littermate control mice (Fig. 5f,g). This increase in FDC number in $Cxcr4^{fl/fl}$; $Cd4^{Cre/+}$ mice was confirmed by flow cytometric analysis (Fig. 5h–j and Supplementary Fig. 6). These data support a role for light zone-localized $T_{FH}$ cells enhancing FDC expansion upon immunization, independent of the size of the GC reaction.

Although we did not observe an impact on GC size or GC $T_{FH}$ cell numbers, 21 d after NP-OVA/alum immunization there was a higher frequency of GC B cells carrying the high-affinity mutation W33L in $Cxcr4^{fl/fl}$; $Cd4^{Cre/+}$ mice compared with control mice (Fig. 5k). This resulted in increased affinity maturation of NP-specific IgG1 antibody in the serum of $Cxcr4^{fl/fl}$; $Cd4^{Cre/+}$ mice, while total titer remained unchanged (Fig. 5l). Consistent with this, the high/low affinity ratio of plasma cells from $Cxcr4^{fl/fl}$; $Cd4^{Cre/+}$ mice was significantly increased (Fig. 5m). We also observed increased high-affinity antibody production in heterozygous $Cxcr4^{fl/+}$; $Cd4^{Cre/+}$ mice, which have a mild skewing of $T_{FH}$ cells to the light zone (Extended Data Fig. 7). However, the effect size was not as big as in homozygous $Cxcr4^{fl/fl}$; $Cd4^{Cre/+}$ mice, likely due to heterozygous mice having only a ~40% reduction in expression of CXCR4 on the $T_{FH}$ cell's surface. Together, these data show that $T_{FH}$ cell restriction to the light zone through CXCR4 deletion can enhance the expansion of the FDC network and results in higher-affinity GC responses without affecting the number of GC B cells or the area of the GC. This demonstrates that $T_{FH}$ cell localization and the expansion of the FDC network are entangled processes which can both be modulated by CXCR4 expression on T cells alone.

---

**Fig. 5 | $T_{FH}$ restriction to the light zone can boost FDC expansion and alter the quality of the GC output. a**, Representative ×20 confocal images of GCs at day 10 after NP-OVA/alum immunization in the iLNs of $Cxcr4^{fl/fl}$; $Cd4^{+/+}$ mice (top) and $Cxcr4^{fl/fl}$; $Cd4^{Cre/+}$ mice (bottom); scale bar, 100 μm. LN sections were stained for IgD (green), Ki67 (blue), CD35 (white) and CD3e (magenta). **b–g**, Quantification of the total GC area (**b**), the total number of CD3+ T cells within the GC (**c**), and the percentage of CD3+ T cells localizing to the CD35+ light zone area (**d**) and the Ki67+CD35− dark zone area (**e**), and the CD35+ FDC light zone area (**f**) and the Ki67+CD35− dark zone area (**g**) ($n = 11$). **h**, Representative plots showing gp38+ICAM+CD31−MadCAM−CD21/35+ FDCs in adult $Cxcr4^{fl/fl}$; $Cd4^{+/+}$ and $Cxcr4^{fl/fl}$; $Cd4^{Cre/+}$ mice 10 d after immunization with NP-OVA/alum. Values adjacent to the gates represent percentages. **i,j**, Quantification of this population frequency (**i**) and total number (**j**) ($n = 15$). **k**, Pie charts indicating the frequency of the affinity-inducing mutation W33L in the CDR1 region of $V_H$186.2 sequenced from NP+IgG1+ GC B cells of $Cxcr4^{fl/fl}$; $Cd4^{+/+}$ and $Cxcr4^{fl/fl}$; $Cd4^{Cre/+}$ mice at 21 d postimmunization with NP-OVA/alum. The values in the center indicate the number of cells sequenced per group ($n = 11$). **l**, Serum titers of NP20- (left) and NP2- (middle) specific IgG1 of $Cxcr4^{fl/fl}$; $Cd4^{+/+}$ and $Cxcr4^{fl/fl}$; $Cd4^{Cre/+}$ mice and antibody affinity maturation indicated by the NP2/NP20 antibody ratio (right) at 35 d postimmunization with NP-OVA ($n = 16$). Titers were normalized to a positive control and are displayed as arbitrary units. **m**, Enumeration of NP20 (left) and NP2 (middle) IgG1 ASCs and affinity maturation indicated by the ratio of NP2/NP20 ASCs (right) in the bone marrow of $Cxcr4^{fl/fl}$; $Cd4^{+/+}$ and $Cxcr4^{fl/fl}$; $Cd4^{Cre/+}$ mice at 35 d postimmunization with NP-OVA ($n = 15$). For all bar graphs, bar height indicates the median, each symbol represents a mouse and $P$ values were obtained by performing an unpaired, two-tailed Mann–Whitney $U$ test. Data are representative of two independent experiments. FDC, follicular dendritic cells. LZ, light zone. DZ, dark zone. ASCs, antibody secreting cells.

## T_FH restriction to the dark zone limits FDC expansion

CXCR5 expression on T cells has previously been shown to localize $T_{FH}$ cells to the GC light zone[35], the loss of which we postulated would mimic the aging GC phenotype. To test this, we generated $Cxcr5^{fl/fl}; Cd4^{Cre/+}$ mice to determine the impact of T cell-specific deletion of $Cxcr5$ (Extended Data Fig. 8). At 10 d after immunization, confocal microscopy showed that both GC size and the number of GC $T_{FH}$ cells in the iLNs of $Cxcr5^{fl/fl}; Cd4^{Cre/+}$ mice were significantly reduced compared with littermate

control mice (Fig. 6a–c and Supplementary Fig. 7). Flow cytometric analysis showed a significant decrease in the percentage and number of GC B cells of $Cxcr5^{fl/fl}; Cd4^{Cre/+}$ mice throughout the GC response at days 10, 21 and 35 postimmunization (Extended Data Fig. 8). However, we found that while the proportion of $T_{FH}$ cells was slightly reduced in mice with T cell-specific CXCR5 deletion, there was no significant impact on the total number of Bcl6+PD-1+ $T_{FH}$ cells compared with control mice (Extended Data Fig. 8). By imaging, CXCR5-deficient $T_{FH}$ cells were

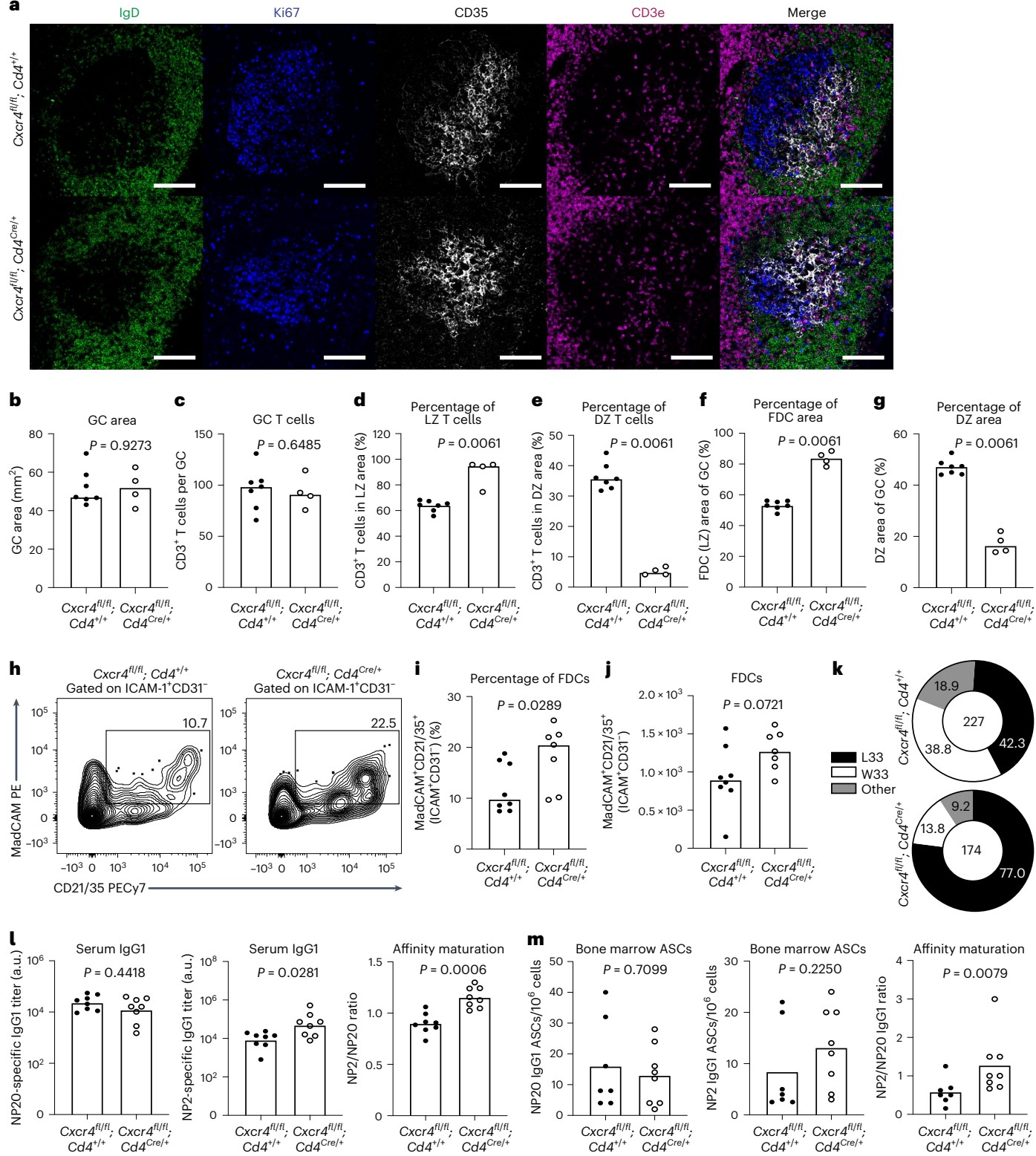

predominantly localized within the GC dark zone with a significantly smaller proportion in the light zone compared with $T_{FH}$ cells from control mice (Fig. 6d,e). The expansion of the FDC network was significantly diminished in GCs of $Cxcr5^{fl/fl};Cd4^{Cre/+}$ mice, resulting in a higher percentage of dark zone areas (Fig. 6f,g and Supplementary Fig. 7). This was independently validated in $Cxcr5^{fl/fl};Cd4^{ERT2Cre}$ mice, in which $Cxcr5$ was ablated in T cells upon tamoxifen administration on days 3 and 4 after immunization with NP-Keyhole Limpet Hemocyanin (KLH)/alum[36]. At 14 d after immunization, confocal microscopy confirmed the localization of CXCR5-deficient $T_{FH}$ cells to the dark zone and a strong trend ($P = 0.07$) towards a diminished FDC network in $Cxcr5^{fl/fl};Cd4^{ERT2Cre}$ mice compared with $Cxcr5^{+/+};Cd4^{ERT2Cre}$ controls (Extended Data Fig. 9). We confirmed the impaired expansion of FDCs by flow cytometric quantification (Fig. 6h–j and Supplementary Fig. 6). These data were reminiscent of the FDC reduction observed in aging and further implicated a role for $T_{FH}$ cell localization to the light zone in FDC expansion. Therefore, improper $T_{FH}$ localization to the dark zone recapitulates the smaller FDC network and smaller GCs observed in aged mice.

Sequencing of the $V_H 186.2$ heavy chain region of NP-specific GC B cells sorted from $Cxcr5^{fl/fl};Cd4^{Cre/+}$ mice 21 d postimmunization showed that the frequency of the high-affinity-inducing mutation W33L was similar between $Cxcr5^{fl/fl};Cd4^{Cre/+}$ and control mice (Fig. 6k). This was reflected in the affinity maturation of NP-specific IgG1 antibody in the serum of $Cxcr5^{fl/fl};Cd4^{Cre/+}$ mice, which was comparable to that of control mice. However, the titers of both NP20- and NP2-specific IgG1 antibody in the serum of $Cxcr5^{fl/fl};Cd4^{Cre/+}$ mice were significantly reduced compared with control mice (Fig. 6l). Likewise, there was only a small effect observed on the affinity ratio of bone marrow NP-specific IgG1 ASCs in $Cxcr5^{fl/fl};Cd4^{Cre/+}$ mice compared with littermate controls, despite an evident reduction in the number of NP20- and NP2-specific IgG1 ASCs of $Cxcr5^{fl/fl};Cd4^{Cre/+}$ mice (Fig. 6m). Taken together, these data indicate that the GC response in $Cxcr5^{fl/fl};Cd4^{Cre/+}$ mice, which display aberrant $T_{FH}$ cell positioning to the dark zone, recapitulates the loss of GC magnitude and cellular output and diminished FDC expansion observed in aged mice. Therefore, the data presented support the hypothesis that aberrant $T_{FH}$ cell positioning contributes to poor GC responses in aging and reveal a role for $T_{FH}$ cell localization in the expansion of the FDC network.

To further evaluate how $T_{FH}$ cell positioning towards the dark zone influences the GC, we performed similar experiments, but with mice whose T cells have only one functional allele of CXCR5, $Cxcr5^{fl/+};Cd4^{Cre/+}$ mice. We hypothesized that $T_{FH}$ cells from these mice would have less surface CXCR5 expression, and an intermediate phenotype between controls and full CXCR5 T cell knockouts. At 10 d after immunization, $T_{FH}$ cells had a ~50% reduction in CXCR5 expression, which resulted in normal-sized GCs, and comparable numbers of GC $T_{FH}$ cells to control $Cxcr5^{+/+};Cd4^{+/+}$ mice, unlike mice that completely lack CXCR5 on their T cells. Nevertheless, heterozygosity of CXCR5 resulted in $T_{FH}$ cell skewing to the dark zone, and the proportion of the GC occupied by

the FDC network was also diminished (Extended Data Fig. 10). This provides further evidence in support of $T_{FH}$ cell localization influencing the GC stroma.

## $T_{FH}$ cell light zone positioning restores aged GCs and FDCs

To test whether $T_{FH}$ cells that localize to the light zone can correct the age-associated GC defects, we transferred CD4+ OT-II T cells from young adult mice into aged mice and assessed the GC response and OT-II $T_{FH}$ cell positioning after NP-OVA immunization (Fig. 7a). Flow cytometry and imaging showed that supplementation of T cells boosted the GC response in aged mice (Fig. 7b–f and Supplementary Fig. 8). Quantitative imaging also revealed that the proportion of the GC occupied by the FDC network was significantly enlarged in the GCs of mice that received OT-II T cells (Fig. 7g), indicating that provision of light zone-localizing $T_{FH}$ cells can support the expansion of aged FDCs upon immunization. Importantly, the transferred OT-II cells correctly localized to the GC light zone (Fig. 7h), indicating that the aged GC microenvironment does not mediate aberrant $T_{FH}$ cell positioning to the dark zone. Of note, transfer of OT-II cells was not able to correct the delayed formation of the GC 7 d after immunization (Supplementary Fig. 8). To test whether correction of these age-dependent defects in the GC impacted humoral immunity in aging, we assessed the ability of OT-II cells to support the formation of high-affinity plasma cells. Indeed, the numbers of NP2-binding GC-derived ASCs were increased in aged mice that received OT-II cells, resulting in enhanced affinity maturation (Fig. 7i). Together, these data support that supply of light zone-localizing T cells to aged mice can expand the FDC network, increase GC size at the peak of the response and enhance GC-derived humoral immunity.

## CXCR5-deficient T cells cannot rescue the aged GC response

To determine whether the observed correction of the GC and FDC response in aged mice was indeed being driven by $T_{FH}$ cell positioning to the light zone rather than simply the transfer of T cells from a young animal, we took advantage of the knowledge that T cells that lack CXCR5 can access the GC but are not enriched in the light zone[35] (Fig. 6). CXCR5-deficient CD4+ OT-II T cells were generated by treating cells from young adult $Cxcr5^{fl/fl};Rosa26^{ERT2Cre/+}$ mice or control OT-II cells with tamoxifen ex vivo, and then these cells were transferred into aged mice who were then immunized (Fig. 8a). At 10 d after NP-OVA immunization, the GC responses in the aged mice that received CXCR5-deficient OT-II cells were significantly reduced compared with the response in aged mice that received CXCR5-sufficient OT-II cells (Fig. 8b–d). This was also visible by confocal microscopy, with only CXCR5-sufficient OT-II cells increasing the GC area and proportion of the GC occupied by FDCs (Fig. 8e–g and Supplementary Fig. 9). Together, the data indicate that CXCR5-dependent colocalization of $T_{FH}$ cells with CXCL13-producing FDCs is essential for facilitating the expansion of the aged FDC network and boosting defective GC responses upon vaccination.

---

**Fig. 6 | $T_{FH}$ restriction to the dark zone in adult mice mimics certain aspects of the aged GC response. a**, Representative ×20 confocal images of GCs at day 10 after NP-OVA/alum immunization in the iLNs of $Cxcr5^{fl/fl};Cd4^{+/+}$ mice (top) and $Cxcr5^{fl/fl};Cd4^{Cre/+}$ mice (bottom); scale bar, 100 μm. LN sections were stained for IgD (green), Ki67 (blue), CD35 (white) and CD3e (magenta). **b–g**, Quantification of the total GC area (**b**), the total number of CD3+ T cells within the GC (**c**), and the percentage of CD3+ T cells localizing to the CD35+ light zone area (**d**) and the Ki67+CD35− dark zone area (**e**), and the CD35+ FDC light zone area (**f**) and the Ki67+CD35− dark zone area (**g**) ($n = 11$). **h**, Representative plots showing gp38+ICAM+CD31−MadCAM+CD21/35+ FDCs in adult $Cxcr5^{fl/fl};Cd4^{+/+}$ and $Cxcr5^{fl/fl};Cd4^{Cre/+}$ mice 10 d after immunization with NP-OVA/alum. Values adjacent to the gates represent percentages. **i,j**, Quantification of this population frequency (**i**) and total number (**j**) ($n = 15$). **k**, Pie charts indicating the frequency of the affinity-inducing mutation W33L in the CDR1 region of $V_H 186.2$ sequenced

from NP+IgG1+ GC B cells of $Cxcr5^{fl/fl};Cd4^{+/+}$ and $Cxcr5^{fl/fl};Cd4^{Cre/+}$ mice at 21 d postimmunization with NP-OVA/alum. The values in the center indicate the number of cells sequenced per group ($n = 11$). **l**, Serum titers of NP20- (left) and NP2- (middle) specific IgG1 of $Cxcr5^{fl/fl};Cd4^{+/+}$ and $Cxcr5^{fl/fl};Cd4^{Cre/+}$ mice and antibody affinity maturation indicated by the NP2/NP20 antibody ratio (right) at 35 d postimmunization with NP-OVA ($n = 16$). Titers were normalized to a positive control and are displayed as arbitrary units. **m**, Enumeration of NP20 (left) and NP2 (middle) IgG1 ASCs and affinity maturation indicated by the ratio of NP2/NP20 ASCs (right) in the bone marrow of $Cxcr5^{fl/fl};Cd4^{+/+}$ and $Cxcr5^{fl/fl};Cd4^{Cre/+}$ mice at 35 d postimmunization with NP-OVA ($n = 15$). For all bar graphs, bar height indicates the median, each symbol represents a mouse and $P$ values were obtained by performing an unpaired, two-tailed Mann–Whitney $U$ test. Data are representative of two independent experiments.

## Discussion

A functional GC response is at the heart of successful responses to vaccination. When the GC is impaired, such as in aging, vaccine efficacy is poor, leading to the need for additional vaccine doses to bolster antibody-mediated immunity. Here, we show that mislocalization of $T_{FH}$ cells to the CXCL12-rich dark zone of aged GCs could be accounted for by the increased expression of CXCR4 on $T_{FH}$ cells from aged mice and their enhanced chemotaxis towards CXCL12, as CXCR5 expression

on CD4[+] T cells is unaltered by age[37]. The increased expression of CXCR4 is a feature of aging T cells that is conserved between mice and humans[38,39], suggesting a common mechanism underpinning its dysregulation. Cell-surface expression of CXCR4 is downregulated after interaction with its ligand[40]. Our confocal imaging of CXCL12 showed no difference in ligand expression in the GC dark zone of aged mice, which, combined with the observation that OT-II cells from young mice do not mislocalize to the GC dark zone in aged hosts, suggests

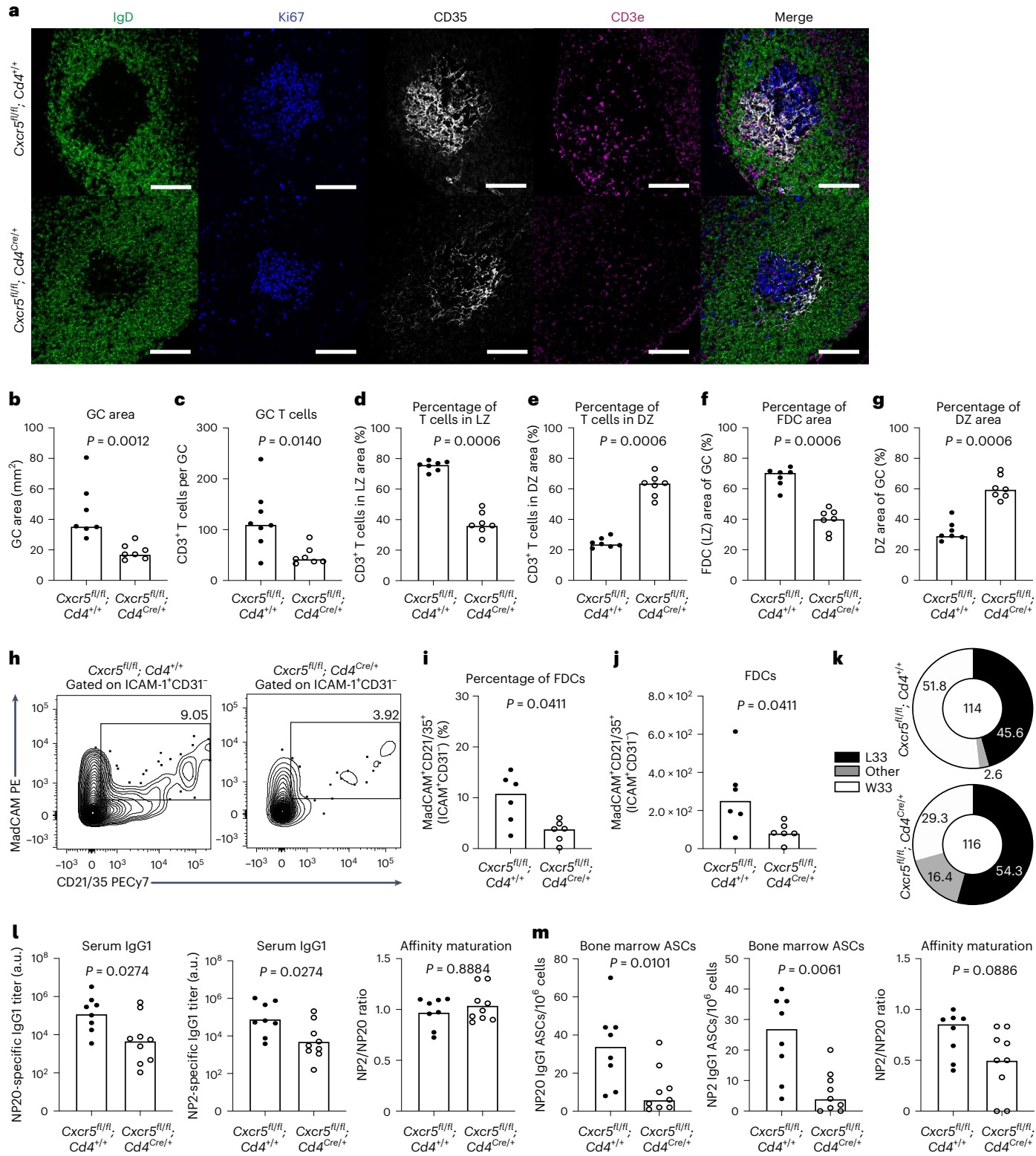

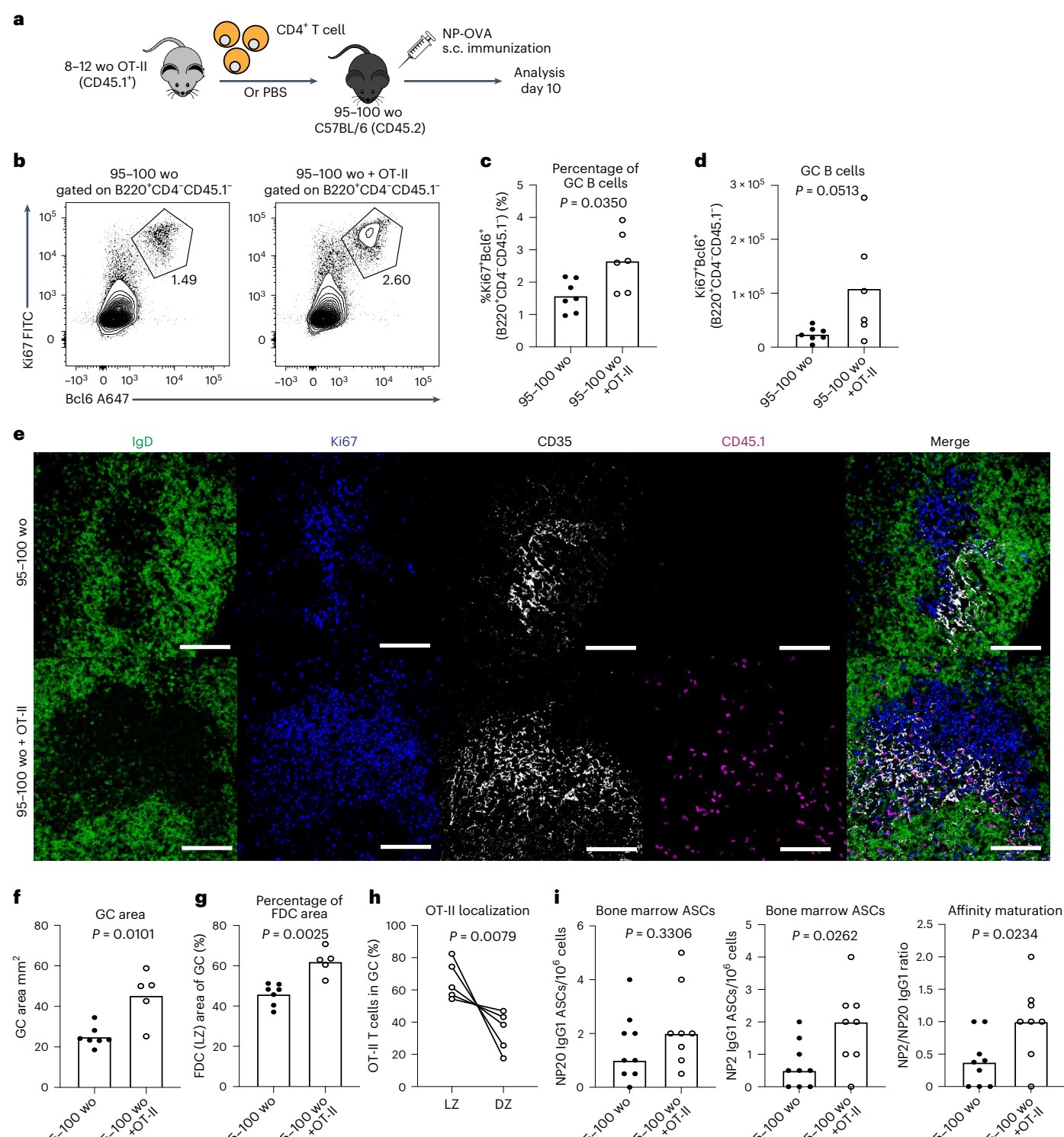

**Fig. 7 | T cell transfer can boost the aged GC response, FDC expansion and humoral immunity. a**, Experimental outline of the adoptive transfer of CD4⁺ T cells, isolated from adult OT-II mice, into aged C57BL6 recipients which were then subcutaneously immunized with NP-OVA/alum and analyzed 10 d after immunization. Control aged C57BL/6 recipient mice were injected with PBS instead of OT-II cells. **b–d**, Representative flow cytometry plots (**b**) and the percentage (**c**) and total number (**d**) of Ki67⁺Bcl6⁺ GC B cells in aged C57BL/6 mice that received either an injection of PBS (left) or CD4⁺ OT-II T cells (right); values adjacent to gates indicate percentage ($n = 13$). **e**, Representative confocal images of GCs at day 10 after NP-OVA immunization in the iLNs of aged C57BL/6 mice that received either an injection of PBS (top) or CD4⁺ OT-II T cells (bottom). Images were taken at ×20 magnification; scale bar, 100 μm. LN sections were stained for

IgD (green), Ki67 (blue), CD35 (white) and CD45.1 (magenta). **f–h**, Quantification of the total area of GCs (**f**), the CD35⁺ FDC network area (**g**) and the percentage of transferred OT-II cells in the light or dark zones (**h**) in the iLNs of aged C57BL/6 mice that received an injection of PBS or CD4⁺ OT-II T cells ($n = 12$). **i**, Enumeration of NP20 (left) and NP2 (middle) IgG1 ASCs and affinity maturation indicated by the ratio of NP2/NP20 ASCs (right) in the bone marrow of aged mice that received OT-II cells or PBS at 35 d postimmunization with NP-OVA/alum ($n = 17$). For all bar graphs, bar height indicates the median, each symbol represents a mouse and $P$ values were obtained by performing an unpaired, two-tailed Mann–Whitney $U$ test. In **h**, $P$ value is from a paired $t$-test, and individual mice are connected with a line. Data are representative of two independent experiments.

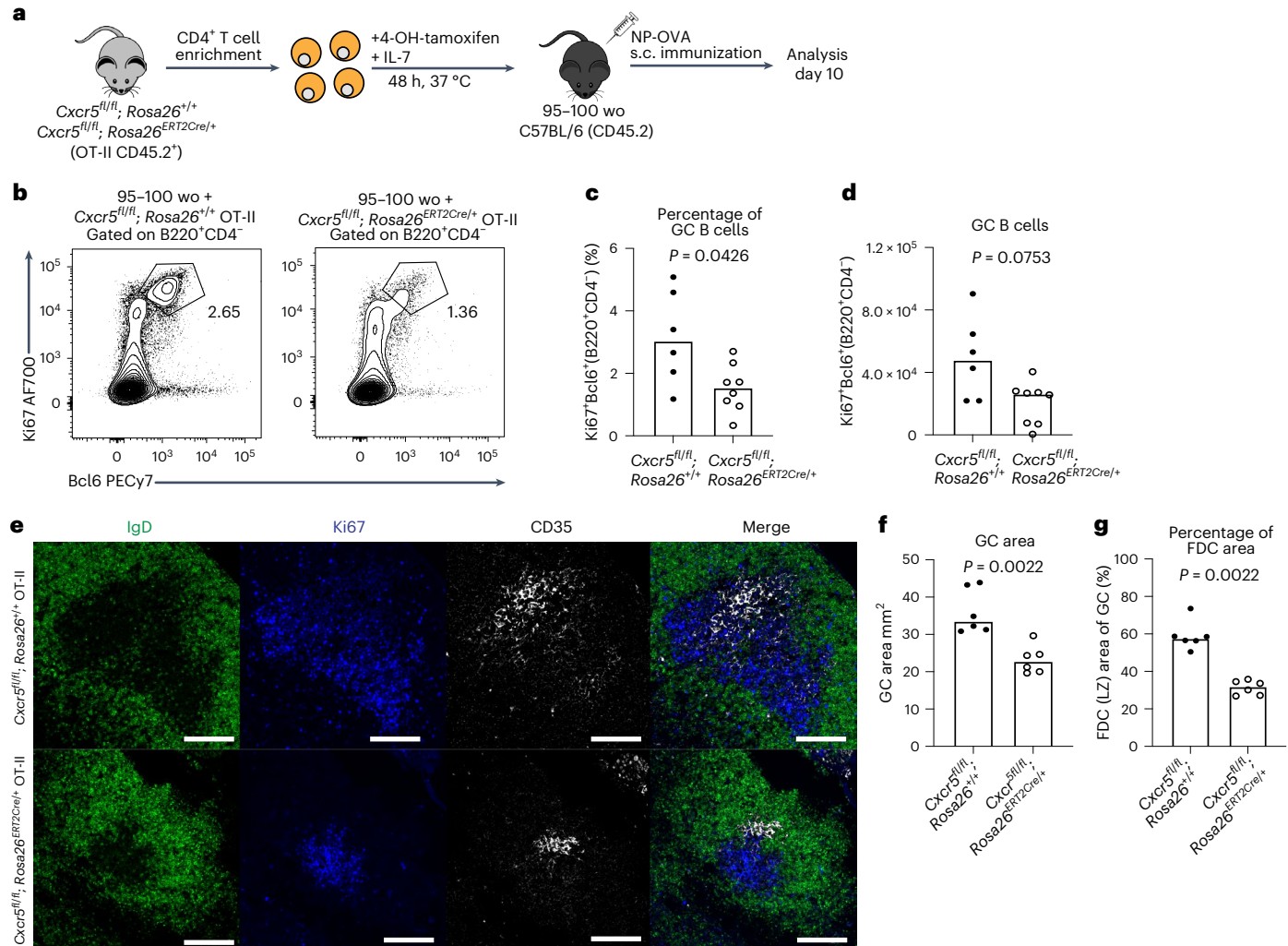

**Fig. 8 | T cell light zone positioning can boost the aged GC response and FDC expansion. a**, Experimental outline of in vitro 4-OH-tamoxifen treatment of CD4$^+$ T cells isolated from *Cxcr5$^{fl/fl}$;Rosa26$^{ERT2Cre/+}$* OT-II mice that were treated for 48 h, after which the cells were transferred into aged C57BL/6 recipient mice. Recipient mice were immunized subcutaneously with NP-OVA in alum and analysis was performed after 10 d. **b**, Representative flow cytometry plots identifying Ki67$^+$Bcl6$^+$ GC B cells in aged C57BL/6 mice that received either *Cxcr5$^{fl/fl}$; Rosa26$^{+/+}$* (left) or *Cxcr5$^{fl/fl}$; Rosa26$^{ERT2Cre/+}$* (right) CD4$^+$ OT-II T cells; values adjacent to gates indicate percentage. **c,d**, Quantification of the percentage (**c**) and total number (**d**) of Ki67$^+$Bcl6$^+$ GC B cells in aged C57BL/6 mice that received either *Cxcr5$^{fl/fl}$; Rosa26$^{+/+}$* or *Cxcr5$^{fl/fl}$;Rosa26$^{ERT2Cre/+}$* OT-II cells (*n* = 14). **e**, Representative ×20

confocal images of GCs at day 10 after NP-OVA immunization in the iLNs of aged C57BL/6 mice that received an injection of either *Cxcr5$^{fl/fl}$;Rosa26$^{+/+}$* (top) or *Cxcr5$^{fl/fl}$;Rosa26$^{ERT2Cre/+}$* (bottom) OT-II cells; scale bar, 100 μm. LN sections were stained for IgD (green), Ki67 (blue) and CD35 (white). **f,g**, Quantification of the total area of GCs (**f**) and the CD35$^+$ FDC network area representative of the light zone compartment within the GCs (**g**) of aged C57BL/6 mice that received an injection of either *Cxcr5$^{fl/fl}$;Rosa26$^{+/+}$* (left) or *Cxcr5$^{fl/fl}$;Rosa26$^{ERT2Cre/+}$* (right) OT-II cells (*n* = 12). For bar graphs, bar heights represent the median, each symbol represents a mouse and *P* values were obtained by performing an unpaired, two-tailed Mann–Whitney *U* test. Data are representative of two independent experiments.

that T cell-intrinsic alterations in CXCR4 internalization and/or degradation are likely responsible for its increased surface expression in aging. After binding CXCL12, CXCR4 is internalized and sorted through the endosomal compartment, where it is either recycled back to the plasma membrane or is degraded. Our data show that ligand-dependent CXCR4 internalization is defective on T$_{FH}$ cells from aged mice, indicating that CXCR4 proteostasis is impaired in aging, likely due to impaired ubiquitination previously described for peripheral blood T cells from older people[38,39].

T$_{FH}$ cells are known to act within the light zone of the GC, where they provide cytokine and costimulatory signals to B cells to promote their survival and proliferation in the GC. The data presented here show that an imbalance in the light zone to dark zone distribution of T$_{FH}$ cells has a profound effect on GC function, and identified CXCR4 is a key controller of this distribution. This complements previous work

showing that light zone polarization of T$_{FH}$ cells requires CXCR5 (ref. 35). By genetically modulating CXCR4 and CXCR5, we demonstrated that T$_{FH}$ cell positioning is important for both the magnitude and quality of the GC response. Moving T$_{FH}$ cells away from the light zone reduces the size of the GC and diminishes its output, consistent with a role for T$_{FH}$ cells in supporting the proliferation of GC B cells. By polarizing T$_{FH}$ cells to the light zone, the affinity of GC B cells and their progeny was increased. This was an unexpected finding as the current model of T$_{FH}$ cell-driven B cell selection in the GC suggests that high-affinity clones compete with each other for T cell help[41], and enhancing the number of T$_{FH}$ cells in the light zone should relax this competition. The enhanced affinity observed would be consistent with prolonged retention of B cells and increased exposure to antigen within the GC[31,42], which may be in turn regulated by the T$_{FH}$ cell location-dependent expansion of the FDC network we report here.

Upon immunization, FDCs increase in number, expand into the GC, upregulate the expression of various cell-surface receptors and display immune complexes on their surface that provide an antigen depot for GC B cells to access[33,43]. It is known that FDCs can sense danger signals directly to facilitate their response to immunization[44], but to our knowledge it has not been reported that interactions with lymphocytes support the expansion of mature FDCs. Our data demonstrate that FDC expansion into the GC requires $T_{FH}$ cells to express CXCR5 which facilitates their colocalization with the CXCL13-producing FDCs. This prompts the question of whether the interaction is direct, or whether it occurs via additional $T_{FH}$ help to GC B cells which in turn promote FDC responses to vaccines. Of note, we did not observe any changes in the number of GC B cells, or their phenotype, in our $Cxcr4^{fl/fl}$ $Cd4^{cre/+}$ mice after immunization, only an increase in the affinity of those GC B cells, indicating that B cell phenotypes are not grossly altered by $T_{FH}$ cell polarization to the light zone. Thus, this study brings to light a role for $T_{FH}$ cells in helping the GC stroma upon vaccination, but how this help is given remains to be elucidated.

Aging is a multifaceted process, and the mechanisms by which it alters the GC reaction are complex due to the number of processes required to coordinate key cellular interactions across time and space for a successful response[1,45]. There are three clear defects in the GC reaction with age: its formation is delayed, its size is smaller at its peak and fewer high-affinity plasma cells are produced. We and others have previously shown that the delayed formation of the GC is due to the age of the LN microenvironment, caused by a diminished response in both conventional dendritic cells and MAdCAM-1-expressing LN stromal cells. However, correction of these impairments through the use of TLR7 and TLR4 agonists could not restore the diminished size of the GC at its peak or the impaired production of high-affinity ASCs[12,46–48]. Here, we show that the transfer of T cells can both boost GC size at the peak of the response (day 10 postimmunization) and increase the number of high-affinity bone marrow plasma cells. However, we did not boost the GC at day 7 postimmunization simply by giving T cells alone, suggesting T cells cannot rescue the delay in GC formation. Together, these studies show that the delay in GC formation is caused by nonmigratory cells in the LN that cannot be corrected by young T cells, while the defective GC size at its peak and output of high-affinity cells are driven by CXCR4-dependent mislocalization of $T_{FH}$ cells. This indicates that effective strategies for enhancing vaccine responses in older people will need to concomitantly address age-dependent changes in both the microenvironment and $T_{FH}$ cells.

## Online content

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

## Methods

### Human cohort and blood sample isolation

Healthy adults were recruited through the NIHR Bioresource before and 7 d after the seasonal influenza vaccine[19,49]. Samples were collected between October 2014 and February 2015, and between October and December 2016, $n = 37$ participants 18–36 yr old, $n = 39$ participants 66–98 yr old. Of the participants, 58% self-identified as female. Venous blood was collected into EDTA-coated tubes and peripheral blood mononuclear cells were isolated using Histopaque-1077 (Sigma) density gradient, then frozen in FBS supplemented with 10% dimethylsulfoxide (Sigma) and stored in liquid nitrogen before flow cytometry.

### Mouse maintenance and husbandry

C57BL/6, BALB/c, OT-II TCR-Tg (ref. 50), B6SJL, $SW_{HEL}$ BCR-Tg (ref. 28), B1.8i BCR-Tg (ref. 51), $Cd4^{Cre/+}$ (ref. 52), $Rosa26^{ERT2Cre/+}$ (ref. 53), $Cxcr4^{fl/fl}$ (ref. 54) and $Cxcr5^{fl/fl}$ (ref. 55) mice were bred and maintained at the Babraham Institute Biological Support Unit; $Cd4^{ERT2Cre/+}$ (ref. 56) and $Cxcr5^{fl/fl}$ (ref. 55) mice were bred and maintained at the Core Facility Animal Models of the Biomedical Center of LMU Munich. Mice were housed under pathogen-free conditions and were kept at an ambient temperature of -19–21 °C with 52% relative humidity. Once weaned, mice were kept in individually ventilated cages with 1–5 mice per cage and were fed CRM (P) VP diet (Special Diet Services) ad libitum. Aged male C57BL/6 and male and female $SW_{HEL}$ mice were 95–108 weeks old at the start of experiments, while aged female BALB/c mice were 90–95 weeks old. All other male and female adult mice used were 8–14 weeks old.

### Subcutaneous immunizations

Mice were immunized subcutaneously on both lower flanks with 100 µl of: 50 µg of NP-KLH (no. N-5060-25, Biosearch Technologies), 20 µg of 1W1K-NP (ref. 12), 50 µg of OVA (no. A5503, Sigma-Aldrich), 10 µg of OVA-HEL (no. 62970, Sigma-Aldrich; conjugated in-house using the SoluLink Protein-Protein Conjugation Kit, no. S-9010-1, TriLink Bio-Technologies) or 50 µg of NP-OVA (no. N-5051-100, Biosearch Technologies) in a 1:1 ratio of PBS with Imject Alum (no. 77161, Thermo Fisher Scientific).

### Flow cytometry

For mouse lymphocyte staining, single-cell suspensions from iLNs were prepared by mechanical disruption of the tissues through a 70-µm mesh in 2% FBS in PBS. For stromal cell staining, single-cell suspensions from iLNs were prepared by enzymatic digestion with 0.2 mg ml⁻¹ Collagenase P (no. 11213865001, Sigma), 0.8 mg ml⁻¹ Dispase II (no. 4942078001, Sigma) and 0.1 mg ml⁻¹ DNase I (no. 10104159001, Sigma) in plain RPMI medium (no. 11875093 Gibco). The cell number and viability of samples were acquired using a CASY TT Cell Counter (Roche). For both mouse and human work, cells were stained with surface antibody stains (Supplementary Tables 1 and 2) for 30 min to 2 h at 4 °C in Brilliant stain buffer (no. 563794, BD Biosciences), then washed with 2% FBS in PBS and fixed using the Foxp3/Transcription Factor Staining Buffer Set (no. 00-5323-00, eBioscience). For intracellular staining, cells were incubated for 1 h at 4 °C with the appropriate antibodies. Samples were acquired on an LSR Fortessa (BD Biosciences) using BD FACSDiva software v.9.0 or on a Cytek Aurora Spectral Cytometer (Cytek) using SpectroFlo Software v.3.0, and analysis was done using FlowJo v.10 software (Tree Star).

### Single-cell sorting for $V_H186.2$ PCR and sequencing

Single B220⁺IgM-GR1-NP⁺IgG1⁺ GC B cells were single-cell sorted into 96-well plates containing 10 µl of reverse transcription lysis buffer (2 U µl⁻¹ RNase inhibitor (no. EO0381, Thermo Fisher Scientific), 4 mM dithiothreitol (no. 43816, Sigma), 30 ng µl⁻¹ Random Hexamers (no. SO142, Thermo Fisher Scientific), 1% NP40 and 0.2 × PBS) using an Aria III Cell Sorter (BD Biosciences). Reverse transcription, nested PCR

and sequencing were performed according to a published protocol[57]. Briefly, complementary DNA was made from single cells then was used for nested PCR with 20 pmol of the following primers: forward, GCTG-TATCATGCTCTTCTTG; and reverse, GGATGACTCATCCCAGGGTCAC-CATGGAGT. The PCR product was then diluted 30 times and 1 µl was used in the second round of PCR, which was prepared with the HotStar Taq DNA polymerase kit (no. 203205, QIAGEN) and 20 pmol of the following primers: forward, GGTGTCCACTCCCAGGTCCA; and reverse, CCAGGGGCCAGTGGATAGAC. The PCR product was purified using the ExoSAP-IT PCR Product Cleanup Reagent (no. 78201, Applied Biosystems) and purified samples were sent for Sanger sequencing to Source Bioscience, UK. Analysis was performed using an automated alignment pipeline in R v.4.0.4 which aligned sequences to the $V_H186.2$ sequence, and the frequency of the affinity-inducing W33L mutation was identified for each sequence. Briefly, sequence trace analysis was automated in R, as follows. Sequence traces were read using the *readsangerseq* function with default parameters from the *sangerseqR* package[58]. Quality control was applied: sequences shorter than 300 nucleotides or containing more than one N base call were removed. The W33L locus was identified using *matchPattern* function—allowing 2 nucleotide mismatches and indels—from the Biostrings package, searching for ACCAGCTACTNNATGCACTGG in the reverse complemented sequence data. A W or L call was assigned as follows: if TNN (in the appropriate position in the nucleotide string above) was TGG, the assignment was W; if TNN was TTA or TTG, this was assigned L; any alternative sequences for TNN were assigned other. Per sample calls were exported as a .csv for downstream analysis in Prism.

### Chemotaxis assays

Chemotaxis assays for CXCL12 were performed using lymphocytes isolated from draining LNs at day 14 postimmunization and washed in complete RPMI medium (no. 11875093 Gibco supplemented with 10% (v/v) FBS (no. F9665, Sigma), 1% (v/v) Penicillin-Streptomycin (no. 15140-122, Thermo Fisher Scientific), 10 mM HEPES (pH 7.2–7.5, no. 15630-056, Gibco), 0.1 mM MEM nonessential amino acid solution (no. 11140-035, Gibco), 1 mM sodium pyruvate (no. 11360-039, Gibco), 55 µM β-mercaptoethanol (no. 21985023, Thermo Fisher Scientific)). Cells were resuspended at $1 \times 10^7$ cells per ml in complete RPMI supplemented with Protease-free BSA (no. 05479-10G, Merck Life Sciences) and rested for 30 min at 37 °C, 5% $CO_2$. Recombinant murine CXCL12 (no. 250-20A, Peprotech) was serially diluted and seeded into the bottom compartment of a 6.5-mm Transwell plate with 5.0-µm-pore polycarbonate membrane inserts (no. CLS3421, Corning Sigma-Aldrich) and plates were incubated for 15 min at 37 °C, 5% $CO_2$. Cells were then left to migrate for 2.5 h at 37 °C, 5% $CO_2$, after which they were collected from the bottom well and stained for flow cytometry analysis as previously described.

### Tamoxifen CD4⁺ OT-II T cell cultures

Cell suspensions were prepared as described above under sterile conditions and were enriched for CD4⁺ T cells using the MagniSort Mouse CD4 T cell Enrichment Kit (no. 8804-6821-74, Thermo Fisher Scientific) according to the manufacturer's instructions. Cells were cultured at a concentration of $2 \times 10^6$ cells per well in complete RPMI medium containing 200 nM 4-OH-tamoxifen (no. SML1666, Sigma) and 2 ng ml⁻¹ recombinant murine IL-7 (no. 217-17, Peprotech) for 48 h at 37 °C, 5% $CO_2$. Following incubation, cells were washed with prewarmed RPMI medium and used for adoptive transfer.

### Adoptive cell transfers

To perform adoptive T cell and B cell transfers, lymphocytes were isolated from spleens and peripheral skin-draining LNs (brachial, axial, superficial cervical and inguinal LNs) of $SW_{HEL}$, B1.8i, OT-II, $Cxcr4^{fl/fl}$; $Rosa26^{ERT2Cre/+}$ OT-II or $Cxcr5^{fl/fl}$; $Rosa26^{ERT2Cre/+}$ OT-II mice. Cell suspensions were prepared as previously described under sterile conditions.

For B1.8i or SW$_{HEL}$ B cell transfers, B cells were enriched using the MagniSort Mouse B cell Enrichment Kit (no. 8804-6827-74, Thermo Fisher Scientific), according to the manufacturer's instructions. For OT-II cell transfers, CD4$^+$ T cells were enriched using the MagniSort Mouse CD4 T cell Enrichment Kit (no. 8804-6821-74, Thermo Fisher Scientific), according to the manufacturer's instructions. For transfer assessing the proliferation of OT-II cells, the CellTrace Violet Cell Proliferation Kit (no. C334557, Invitrogen) was used to stain CD4$^+$ OT-II cells for 15 min at 37 °C. For all adoptive cell transfers, aliquots of donor cells were taken and stained to determine the proportion of antigen-specific cells by flow cytometry before transfer. For SW$_{HEL}$ B cell and OT-II T cell cotransfers, cells were resuspended in 2% FBS in PBS at a concentration of 2 × 10$^6$ HEL-binding B220$^+$ B cells and 2 × 10$^6$ TCRVa2$^+$TCRVb5$^+$CD4$^+$ T cells per ml and mixed at a 1:1 ratio for transfer. For CellTrace Violet-stained OT-II T cell transfers, cells were resuspended at a concentration of 5 × 10$^6$ TCRVa2$^+$TCRVb5$^+$CD4$^+$ T cells per ml, and for all other OT-II transfers, cells were resuspended at a concentration of 5 × 10$^5$ TCRVa2$^+$TCRVb5$^+$CD4$^+$ T cells per ml. For B1.8i B cell transfers, cells were resuspended at a concentration of 1 × 10$^5$ NP$^+$B220$^+$ B cells per ml. Donor cells were then injected intravenously into the tails of congenic recipient mice, and each mouse received 100 µl of cells. Recipient mice were then immunized subcutaneously with HEL-OVA, OVA or NP-OVA as described above, and iLNs were collected at the appropriate time points for flow cytometry and microscopy analysis.

## ELISAs
ELISA plates (no. 456537, Thermo Fisher Scientific) were coated overnight at 4 °C with 10 µg ml$^{-1}$ NP20-BSA (no. N-5050H-100, Biosearch Technologies), 2.5 µg ml$^{-1}$ NP7-BSA (no. N-5050L-100, Biosearch Technologies) or 2.5 µg ml$^{-1}$ NP2-BSA (no. N-5050L-100, Biosearch Technologies) in PBS. Plates were washed and blocked with 2% (w/v) BSA in PBS for 1 h at 20 °C, then washed, and sera were loaded at a starting dilution of 1:200 in 1% (w/v) BSA in PBS and titrated down the plate at a 1:4 ratio. The plates were incubated for 2 h at 20 °C and then washed. Detection of NP-specific antibodies was performed with either polyclonal goat anti-mouse IgG1 (no. ab97240, Abcam) or IgM (no. ab97230, Abcam) HRP-conjugated antibodies. The plates were developed with 100 µl per well of TMB solution (no. 421101, BioLegend) for up to 20 min, when the reaction was stopped with 50 µl per well of 0.5 M H$_2$SO$_4$. The absorption was measured at 450 nm using the PHERAstar FD microplate reader (BMG Labtech) with PHERAstar FSX software v.5.7.

## Bone marrow enzyme-linked immunosorbent spot assays
MultiScreen-HA mixed cellulose ester plates (no. MAHAS4510, Millipore Merck) were coated with 10 µg ml$^{-1}$ NP23-BSA, 10 µg ml$^{-1}$ NP20-BSA or 5 µg ml$^{-1}$ NP2-BSA in PBS overnight at 4 °C. Plates were then washed with PBS and blocked with complete DMEM medium (no. 41965-039, Gibco, supplemented with 10% (v/v) FBS (no. F9665, Sigma), 1% (v/v) Penicillin-Streptomycin (no. 15140-122, Thermo Fisher Scientific) and 55 µM β-mercaptoethanol (no. 21985023, Thermo Fisher Scientific)) for 1 h at 20 °C. Bone marrow cell suspensions were diluted (1:2) in complete DMEM down the plate with a starting concentration of 2 × 10$^6$ cells per well, and were incubated at 37 °C, 5% CO$_2$ overnight. Plates were then washed with 0.05% (v/v) Tween20 PBS, PBS and H$_2$O. Detection was performed with either anti-mouse HRP-conjugated IgG1 (no. ab97240, Abcam) or IgM (no. ab97230, Abcam) in 0.1% (w/v) BSA, 0.05% (v/v) Tween20 PBS. Plates were developed using the AEC staining kit (no. AEC101, Sigma-Aldrich). The number of ASCs was determined using a CTL ELISPOT reader (Cell Technologies) and the ImmunoSpot v.5.0 (Cellular Technology).

## Immunofluorescence staining for confocal microscopy
Immunofluorescence staining for confocal imaging of GCs was done as previously described[59]. In brief, LN sections were stained with the primary antibodies listed in Supplementary Table 3 and the secondary antibodies listed in Supplementary Table 4. Images were acquired using the ×10, ×20 and ×40 objectives on the Zeiss 780 confocal microscope using Zen Microscopy software v.3.2-3.5. A minimum of six sections were analyzed for each GC found within each LN to capture the center-most sections of the GC. All GCs per LN sample were imaged, resulting in a total of 3–8 GCs per section, with the exception of some LNs from aged mice which did not generate GCs. Image processing was done using ImageJ and quantitative analysis was done using an automated pipeline on the Cell Profiler software v.3.19. The GC area was defined as IgD$^-$Ki67$^+$CD35$^+$, the GC light zone area was defined as the IgD$^-$CD35$^+$ FDC network area and the GC dark zone as the Ki67$^+$CD35$^-$IgD$^-$ area. Using ImageJ (Fiji) v.2.0.0-rc-69/1.52p and Cell Profiler v.3.1, the number of CD3$^+$ T cells within each compartment of the GC was identified and reported as a percentage of total T cells within the GC.

CXCL12 staining in tissue sections was performed as follows. LN sections were rehydrated with PBS and permeabilized with PBS/0.1% Triton for 30 min at 20 °C, then incubated in PBS/5% BSA/0.05% Saponin and 0.1% goat serum for 15 min at 20 °C. The purified CD35 antibody was incubated overnight at 4 °C. After washing with PBS, the AF555-conjugated secondary antibody was added for 45 min at 20 °C. Then, the CXCL12-AF647 or AF647-conjugated corresponding isotype control staining was performed along with IgD-AF488 for 1.5 h at 20 °C, followed by washes with PBS and DAPI labeling. After mounting the slides using ProLong Gold Antifade mounting medium (Thermo Fisher Scientific), the images were obtained using the LSM800 confocal microscope (Carl Zeiss).

## In silico GC modeling
In silico mathematical modeling of the GC response was carried out using the agent-based model, known as hyphasma[31,32] and previously calibrated to accurately describe the three-dimensional dynamics of cells in GCs from experimental datasets. Briefly, each simulated cell represents an agent with properties such as direction of movement, position and antigen receptor. The GC is modeled as a grid within a sphere, where each grid point is the size of a cell (5 µm) and has a defined concentration of the chemokines CXCL12 and CXCL13 present in the dark zone and light zone, respectively. Each simulated cell's properties, that is, migration, position and antigen receptor, can be manipulated computationally, and it carries a polarity that defines the direction of movement and a persistence time defining the frequency of possible re-orientation in complex chemokine fields. The direction of movement follows a persistent random walk, with a turning angle predefined from experimental 2-photon microscopy[46]. Chemotaxis is implemented by skewing the turning angles towards the gradient of a chemokine. In each simulation, B cells first search for, and acquire, antigen from FDCs, then search for, and interact with, T$_{FH}$ cells to receive a proliferation signal. T$_{FH}$ cell help is given to those B cells with the highest amount of captured antigen, which also indicates B cell receptor affinity. The state of centroblasts and centrocytes is updated at each time point, leading to a choice of death, differentiation or recycling, subsequent to T$_{FH}$ cell-mediated selection. If the fate is recycling, the selected B cell becomes sensitive to CXCL12 and re-enters the cell cycle. In the model used here, the GC is seeded with a constant inflow of founder B cells, each of which divides six times[60]; this models the in vivo findings that 10–100 B cell clones seed a founder GC[61]. A full list of parameters is available in ref. 62.

To simulate the aged GC, two mechanisms have been added: the distribution of the FDCs has been relocated to a smaller area taken from measurements on aged mice, but keeping the total antigen density, and T$_{FH}$ cells have been given a different chemotaxis responsiveness to CXCL12. The strength of chemotaxis in silico was chosen to reproduce the changes of T$_{FH}$ location observed in aged mice in vivo. The chosen parameters and their original values are shown in Supplementary Table 5. Once the sensitivity was established, the biologically relevant

changes that best reflect in vivo data were chosen to simulate the aged GC response. Each simulation was repeated at least 20 times and the curves, and their standard deviations, were plotted using GLE v.4.2 (Graphics Layout Engine) software

### Statistics

All mouse experiments were performed 2–4 times with 4–10 mice per group, and all significant changes reported were reproducible between experimental repeats. No statistical methods were used to predetermine sample sizes but our sample sizes are similar to those reported in previous publications[12,46]. Where possible, mice were randomly allocated into age- and sex-matched experimental groups by staff of the Babraham Institute Biological Services Unit. For experiments with aged mice, blinding was not possible due to overt phenotypic differences between adult and aged mice. All data points were analyzed including outliers unless there were technical errors. Any aged mice with lymphoma and/or solid tumors were also excluded from analysis. Differences between experimental groups were assessed using a two-sided nonparametric Mann–Whitney $U$ test for unpaired comparisons, while grouped analyses were performed by a two-way analysis of variance (ANOVA) with Sidak's multiple comparisons. Normality testing was not performed on the data before analyses because the small sample sizes used for in vivo mouse work were not well powered for normality testing. All statistical tests were performed using GraphPad Prism v.6-9 software and $P$ values $\leq 0.05$ were considered statistically significant.

### Ethical approval

All research complies with the relevant ethical regulation. All mouse experimentation, except the work using $Cxcr5^{fl/fl}$; $Cd4^{ERT2Cre/+}$ mice, was performed in the United Kingdom with approval from the Babraham Institute Animal Welfare and Ethical Review Body, and complied with European Union and UK Home Office legislation (Home Office License P4D4AF812). Experiments with $Cxcr5^{fl/fl}$; $Cd4^{ERT2Cre/+}$ were done at the Core Facility Animal Models of the Biomedical Center of Ludwig Maximilian University of Munich in accordance with European regulation and federal law of Germany, and approved by the Regierung von Oberbayern. Human samples were collected in accordance with the latest revision of the Declaration of Helsinki and the Guidelines for Good Clinical Practice from the International Council for Harmonisation of Technical Requirements for Pharmaceuticals for Human Use. Informed consent was obtained from all participants. The samples were collected with UK local research ethics committee approval (National Research Ethics Service Committee South Central–Hampshire A, Research Ethics Committee reference 14/SC/1077), using the facilities of the National Institute for Health and Care Research Cambridge Bioresource (Research Ethics Committee reference 04/Q0108/44).

### Reporting summary

Further information on research design is available in the Nature Portfolio Reporting Summary linked to this article.

## Data availability

Source data are provided with this paper.

## Code availability

The code used for in silico modeling is available at: https://gitlab.com/simm/gc/hyphasma/-/releases/Denton2020. The code used for the W33L VH186.2 sequencing analysis is available at: https://github.com/lintermanlab/W33L_caller.

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

## Acknowledgements

We thank C. Vinuesa and A. Liston for critical feedback on this paper. We thank the Babraham Institute Biological Support Unit staff, who performed in vivo treatments of our animals and took care of animal husbandry. We thank the staff of the Babraham Flow Cytometry and Imaging Facilities for their technical support. The National Institute for Health Research Cambridge Biomedical Research Center is a partnership between Cambridge University Hospitals NHS Foundation Trust and the University of Cambridge, funded by the National Institute for Health Research. We thank the National Institute for Health Research Cambridge Biomedical Research Center volunteers for their participation and thank staff for their contribution in coordinating the vaccinations and venipuncture. This study was supported by funding from the Biotechnology and Biological Sciences Research Council (grant nos. BB/W001578/1, BBS/E/B/000C0407, BBS/E/B/000C0427 to M.A.L.; grant no. BBSRC BB/N011740/1 to A.E.D; and the Campus Capability Core Grant to the Babraham Institute), the European Union's Horizon 2020 research and innovation program 'ENLIGHT-TEN' under the Marie Sklodowska-Curie grant agreement no. 675395 to M.A.L., a grant from IdEx Université de Paris (grant no. ANR-18-IDEX-0001 to M.E.) and by an ANR PRC grant (grant no. ANR-17-CE14-0019 to K.B.). M.A.L. is an EMBO Young Investigator and a Lister Institute Prize Fellow. D.B. was supported by the Deutsche Forschungsgemeinschaft (DFG, German Research Foundation) under Emmy Noether Programs BA 5132/1-1 and BA 5132/1-2 (grant no. 252623821 to D.B.), SFB 1054 Project B12 (grant no. 210592381 to D.B.) and Germany's Excellence Strategy EXC2151 (grant no. 390873048 to D.B.). P.A.R. was supported

by the Human Frontier Science Program (grant no. RGP0033/2015 to P.A.R.) and a PhD fellowship granted by École Normale Supérieure de Lyon. J.L.L. is supported by a National Science Scholarship (PhD) by the Agency for Science, Technology and Research, Singapore. D.L.H. received a National Health and Medical Research Council Australia Early-Career Fellowship (grant no. APP1139911). A.R.B. received a Sir Henry Wellcome Postdoctoral Fellowship (grant no. 222793/Z/21/Z). J.P.L. was a recipient of the People Program (Marie Curie Actions) of the European Union's Seventh Framework Program (FP7/2007-2013) under REA grant agreement no. PCOFUND-GA-2013-609102.

## Author contributions
A.S.-C. designed and performed experiments, analyzed data, constructed the figures and cowrote the paper. S.F.-B., S.I., A.R.B., E.M.W., J.L.L., L.M.C.W., R.C.J.M., A.B., I.V., D.A., J.P.L., E.J.C., D.L.H., I.C., W.S.F. and A.E.D. designed and performed experiments and analyzed data. P.A.R., M.M.-H. and A.S.-C. performed in silico modeling of the germinal center. K.B., D.B., M.E., M.M.-H., A.E.D. and M.A.L. provided supervision and contributed to experimental design and funding for this study. M.A.L. designed the study and cowrote the paper. All authors contributed intellectually to the work, and reviewed, edited and approved the paper.

## Competing interests
The authors declare no competing interests.

## Additional information
**Extended data** is available for this paper at https://doi.org/10.1038/s41590-023-01519-9.

**Correspondence and requests for materials** should be addressed to Michelle A. Linterman.

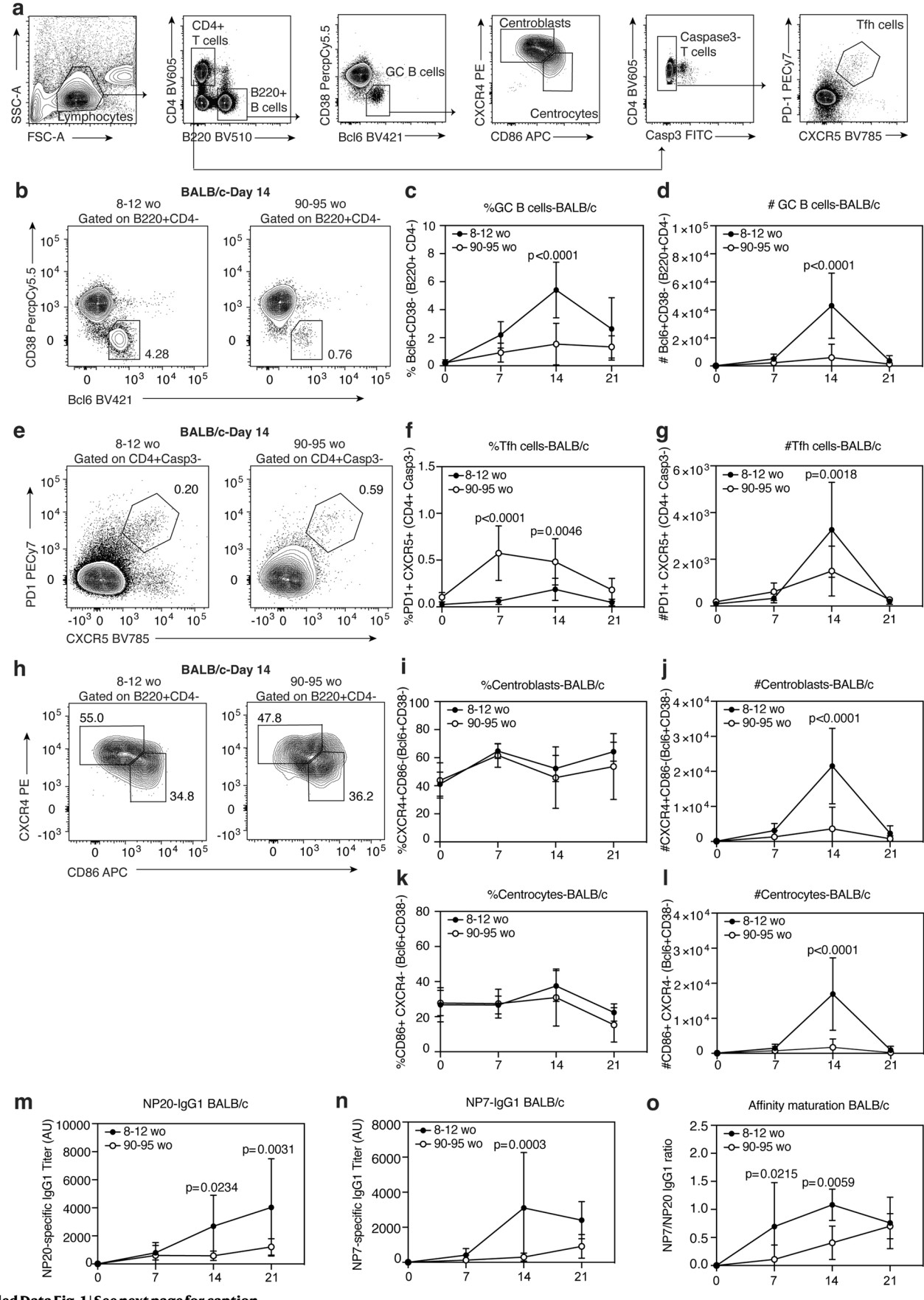

**Extended Data Fig. 1 | See next page for caption.**

**Extended Data Fig. 1 | Kinetic analysis of GC response in aged BALB/c mice after NP-KLH/Alum immunization. (a)** Gating strategy for the analysis of GC B cells, centroblasts, centrocytes and Tfh cell populations in the iLNs of adult and aged BALB/c mice immunized with NP-KLH in alum. **(b)** Representative flow cytometry plots identifying GC B cells (Bcl6 + CD38⁻B220 + CD4-) in the inguinal lymph nodes (iLNs) of adult and aged BALB/c mice and their frequency **(c)** and total number **(d)** at the indicated timepoints post immunization. **(e)** Representative flow cytometry plots identifying Tfh cells (PD1 + CXCR5 + CD4 + Casp3-B220-) and their frequency **(f)** and total number **(g)** in the iLN at the indicated timepoints post immunization. **(h)** Representative flow cytometry plots identifying centroblasts (CXCR4$^{hi}$CD86$^{lo}$Bcl6$^+$CD38$^-$) and centrocytes (CD86$^{hi}$CXCR4$^{lo}$Bcl6$^+$CD38$^-$) and their frequency and number **(i-l)** at the indicated timepoints after immunization. Adult and aged BALB/c serum antibody titers of NP20 **(m)** and NP7 **(n)** specific IgG1 and antibody affinity maturation **(o)** as determined by the ratio of NP7/NP20 IgG1 titers, at the indicated time point post immunization. The data is representative of two independent experiments (n = 14) where each symbol represents the mean ± SD and p-values were generated by performing a two-way ANOVA with Sidak's multiple comparisons test.

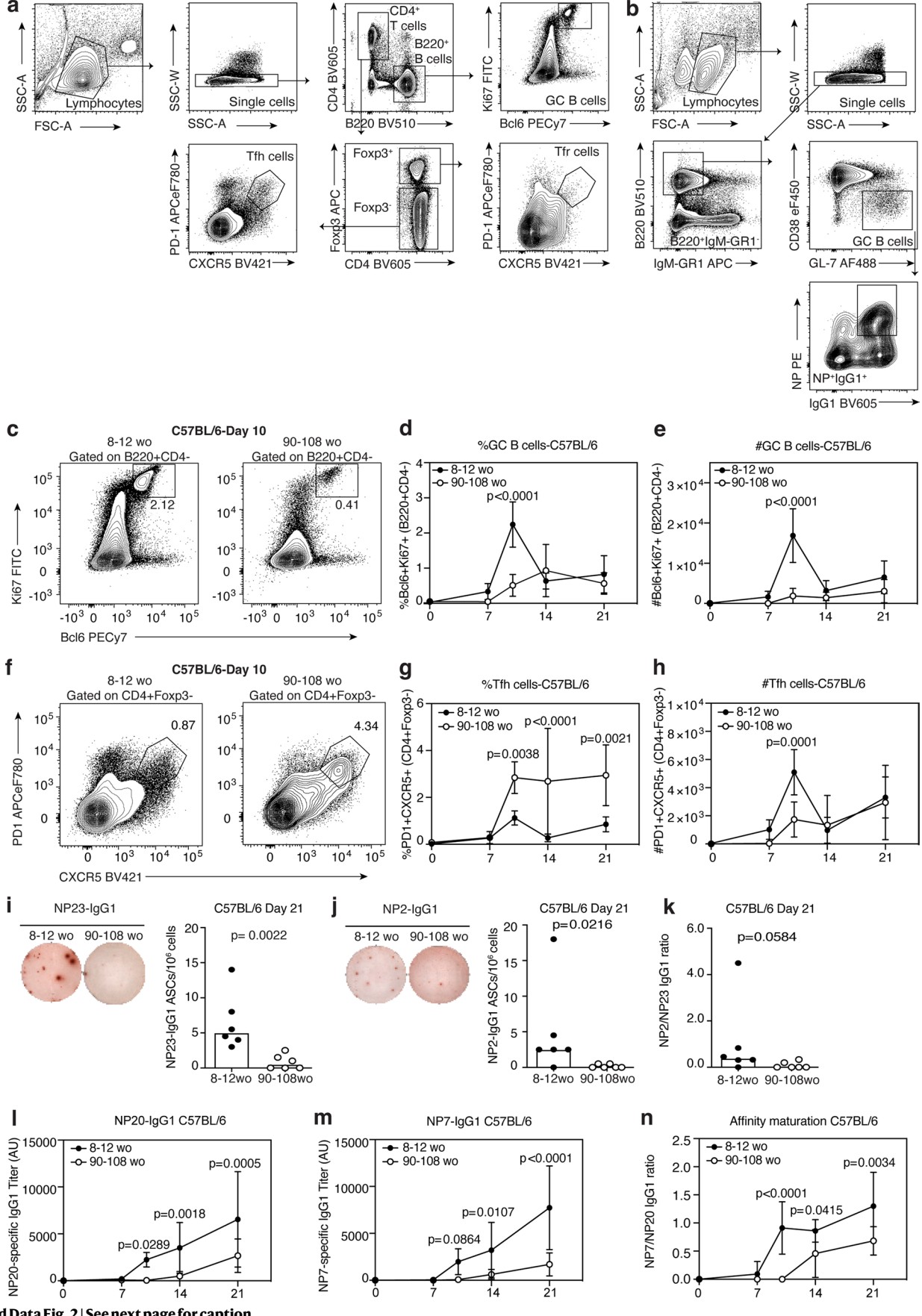

**Extended Data Fig. 2 | See next page for caption.**

**Extended Data Fig. 2 | Kinetic analysis of GC response in aged C57BL/6 mice after 1W1K-NP in alum immunization. (a)** Gating strategy for the analysis of GC B cells, Tfh and Tfr populations in the iLNs of adult and aged C57BL/6 mice immunized with 1W1K-NP in alum. **(b)** Single-cell sorting strategy of NP + IgG1+ GC B cells from iLNs of C57BL/6 mice 21 days after immunization with 1W1K-NP in alum for $V_H$186.2 sequencing. **(c)** Representative flow cytometry plots identifying GC B cells (Ki67$^+$Bcl6$^+$B220$^+$CD4$^-$) in the inguinal lymph nodes (iLNs) of adult and aged C57BL/6 mice 10 days after subcutaneous immunization with 1W1K-NP in alum and their frequency **(d)** and total number **(e)** at the indicated timepoints post immunization. **(f)** Representative flow cytometry plots identifying Tfh cells (PD1$^+$CXCR5$^+$CD4$^+$Foxp3$^-$B220$^-$) in the iLNs of adult and aged C57BL/6 mice 10 days post-immunization with 1W1K-NP in alum and their frequency **(g)** and total number **(h)** at the indicated timepoints post immunization. **(i-k)** Representative ELISpot well images (left) and quantification (right) of bone marrow NP23 **(i)** and NP2 **(j)** specific IgG1 antibody-secreting cells (ASCs) in C57BL/6 mice 21 days after immunization. **(k)** Affinity maturation of bone marrow ASCs from C57BL/6 mice as determined by the ratio of NP2/NP23-specific ASCs. For **(i-k)** the data is representative of two independent experiments (n = 12) where each symbol represents a mouse, and the bar height represents the median. The p-values were obtained by performing an unpaired, two-tailed Mann Whitney U test. **(l-n)** Adult and aged C57BL/6 serum antibody titers of NP20 **(l)** and NP7 **(m)** specific IgG1 at the indicated timepoints after immunization with NP-KLH in alum and antibody affinity maturation **(n)** as determined by the ratio of NP7/NP20 IgG1 titers. For **(l-n)** antibody titers shown are displayed as arbitrary units (AU). For **(c-h, l-n)** The data are representative of two independent experiments (n = 16) where each symbol represents the mean ± SD and p-values were generated by performing a two-way ANOVA with Sidak's multiple comparisons test.

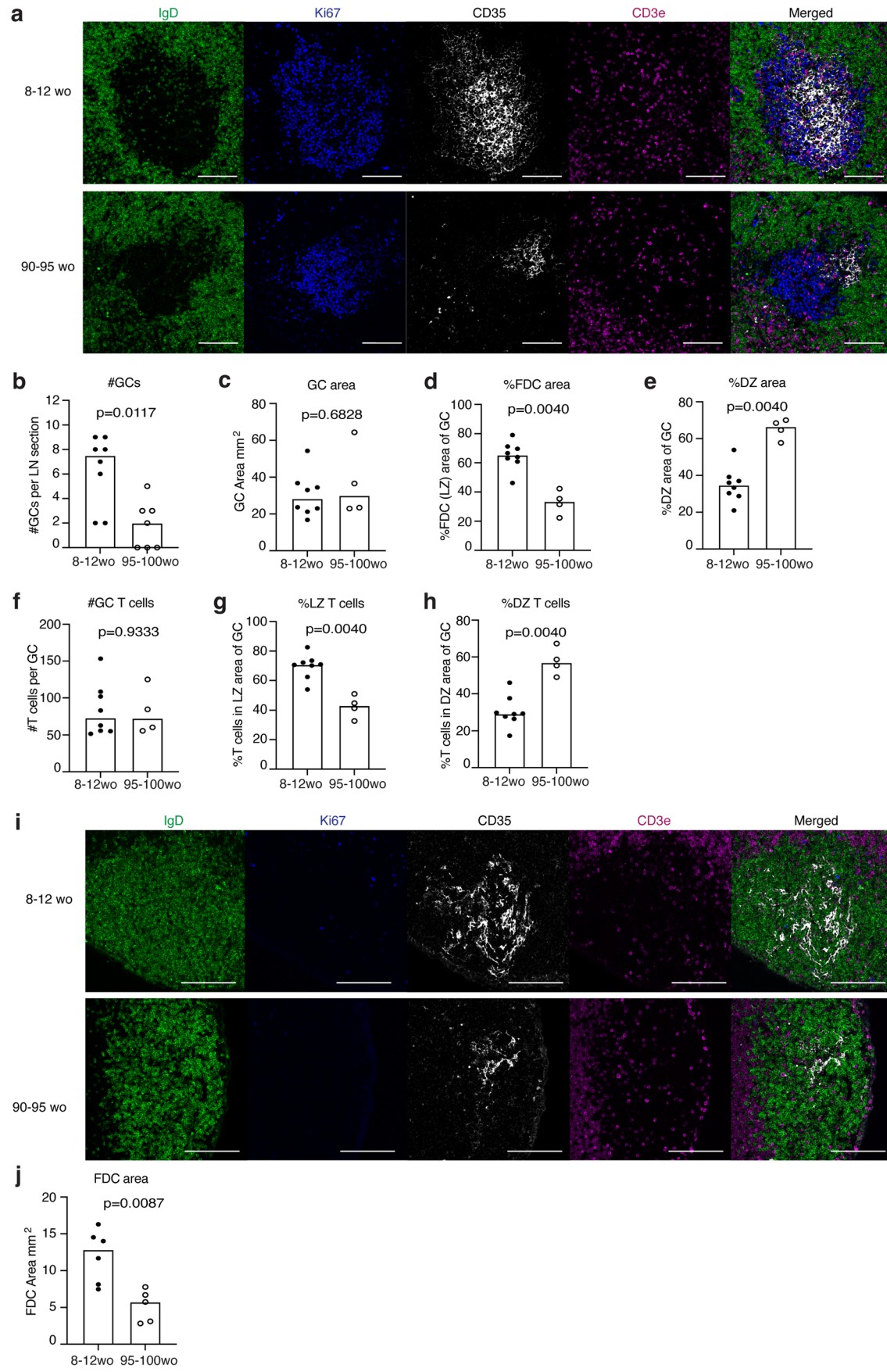

**Extended Data Fig. 3 | See next page for caption.**

**Extended Data Fig. 3 | The spatial organization of the GC is disrupted in aged C57BL/6 mice.** (a) Representative confocal images of GCs at 20x from adult and aged C57BL/6 mice 12 days after immunization with NP-OVA in alum. Scale bars are 100 μm. LN sections were stained for IgD (green), CD35 (white), Ki67 (blue) and CD3 (magenta). Enumeration of GC number (**b**) and area (**c**) per LN was performed by examining 6–10 sections throughout each LN and identifying GCs as CD35$^+$Ki67$^+$IgD$^-$ structures. (**d**) Quantification of the CD35$^+$ FDC network light zone area of the GC. (**e**) Quantification of the Ki67$^+$CD35$^-$ dark zone area of the GC. (**f**) Quantification of the number of CD3$^+$ T cells identified within the total Ki67$^+$CD35$^+$IgD$^-$ GC area. Quantification of the proportion of T cells positioned in the CD35$^+$ FDC light zone area (**g**) and Ki67$^+$CD35$^-$ dark zone (**h**) of the GC (n = 15). (**i**) Representative confocal images of GCs at 40x from unimmunized naive adult and aged C57BL/6 mice and (**j**) quantification of the CD35$^+$ FDC network in the primary B cell follicle (n = 11). For (**b-h, j**), quantification of the GC compartments and T cell positioning was performed using an automated Cell Profiler pipeline. The data is representative of two independent experiments where each symbol on the graph represents a mouse and the bar height represents the median. The p-values were generated by performing an unpaired, two-tailed Mann Whitney U test.

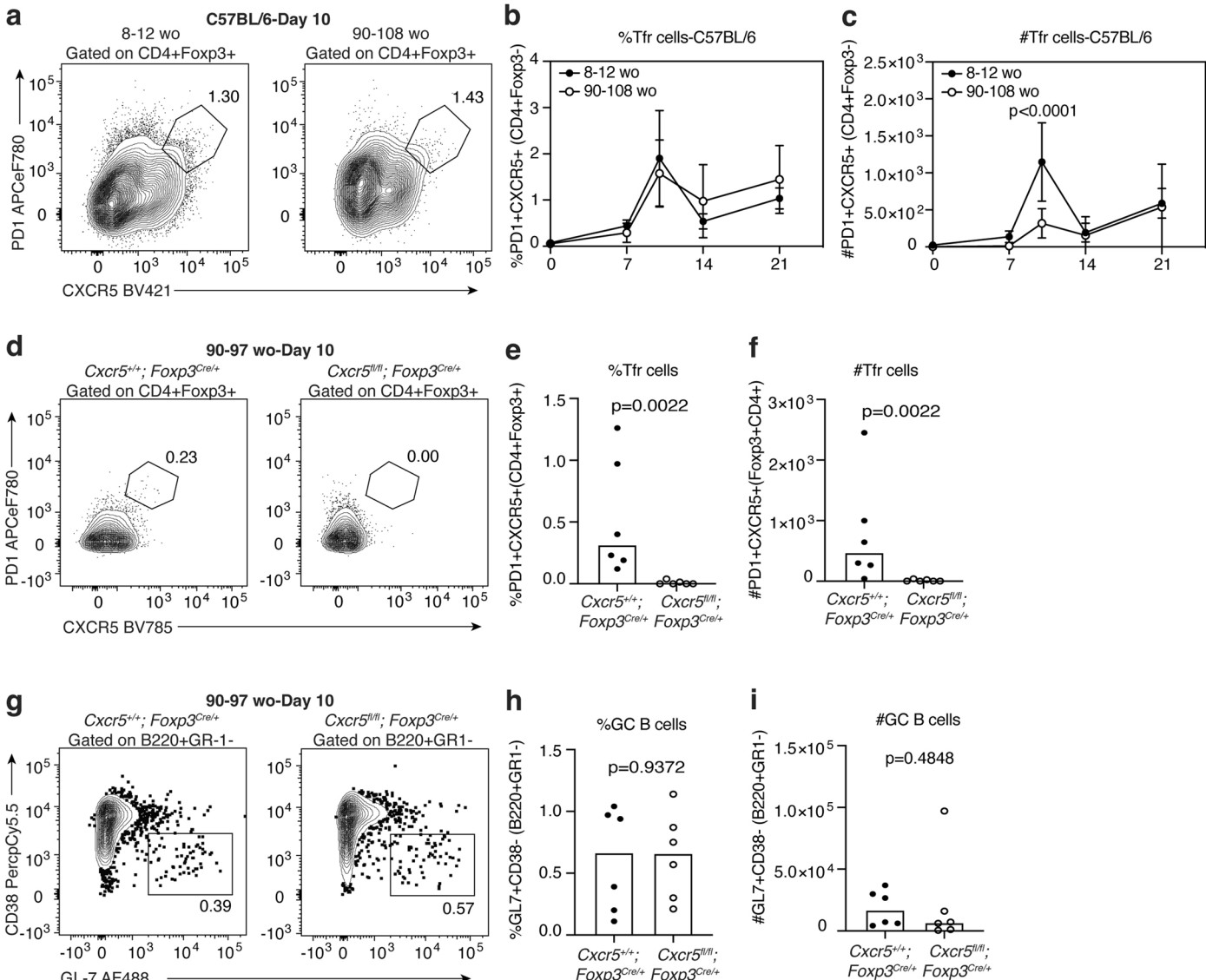

**Extended Data Fig. 4 | Tfr cell reduction in aged mice does not enhance GC responses. (a)** Representative flow cytometry plots identifying Tfr cells (PD1[+] CXCR5[+]CD4[+]Foxp3[+]B220[−]) in the iLNs of adult and aged C57BL/6 mice 10 days post-immunization with 1W1K-NP in alum. The frequency **(b)** and total number **(c)** of Tfr cells in adult and aged C57BL/6 mice was quantified at days 0, 7, 10, 14 and 21 after immunization with 1W1K-NP in alum. The data is representative of two independent experiments (n = 14) where each symbol represents the mean ± SD and p-values were generated by performing a two-way ANOVA with Sidak's multiple comparisons test. **(d)** Representative flow cytometry plots indicating deletion of Tfr cells in aged 90–97-week-old *Foxp3^cre^Cxcr5^fl/fl^* mice

compared to aged littermate control *Foxp3^cre^* mice 10 days after immunization with NP-KLH in alum. Quantification of the frequency **(e)** and total number **(f)** of Tfr cells (n = 12). **(g)** Representative flow cytometry plots identifying GC B cells (GL7[+]CD38[−]B220[+]GR-1[−]) in *Foxp3^cre^Cxcr5^fl/fl^* mice 10 days after immunization with NP-KLH in alum. Quantification of the frequency **(h)** and total number **(i)** of GC B cells (n = 12). The values next to the gates on flow cytometry plots indicate the population percentage. The data is representative of two independent experiments where each symbol represents a mouse, and the bar height represents the median. The p-values were obtained by performing an unpaired, two-tailed Mann-Whitney U test.

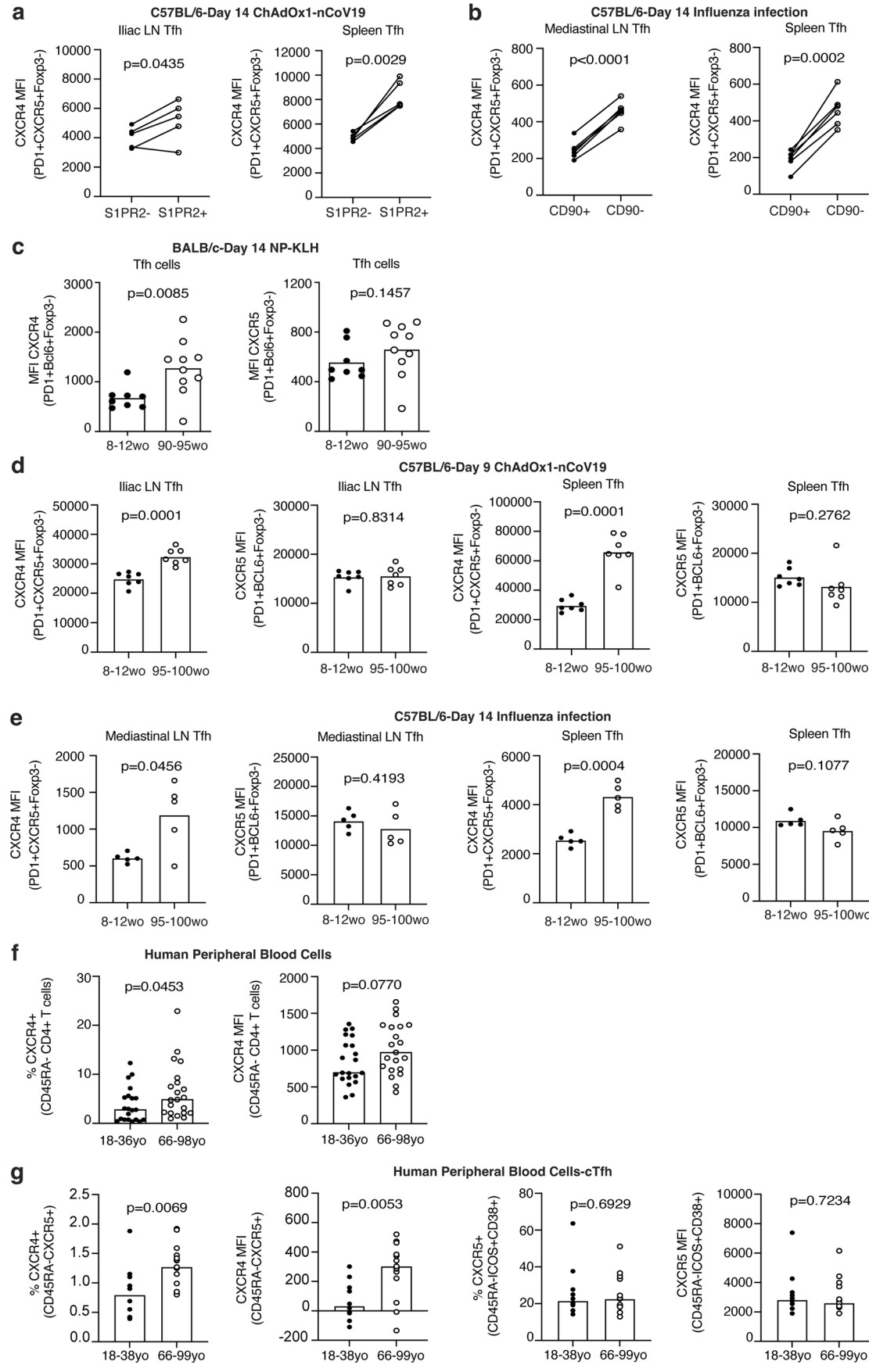

**Extended Data Fig. 5 | See next page for caption.**

**Extended Data Fig. 5 | A proportion of GC Tfh cells express CXCR4 and its expression is also increased in Tfh cells from aged mice upon ChAdOx1-nCoV19 immunization and influenza infection as well as in CD4+ T cells from human peripheral blood cells.** (a) Five *S1pr2*^ERTcre *Rosa26*^stop-flox-RFP mice were immunized with 50uL of ChAdOx1 nCoV-19 intramuscularly, followed by oral gavage of Tamoxifen at 8- and 10-days post immunization. Dots show CXCR4 expression on S1PR2- Tfh cells and S1PR2-fatemapped GC Tfh cells 14 days after immunization (n = 10 biologically independent samples). (b) Six C57BL/6 mice were infected with influenza and CXCR4 expression assessed on CD90 + Tfh cells and CD90low GC Tfh cells 14 days after infection (n = 12 biologically independent samples). (c) Median fluorescence intensity of CXCR4 on PD1$^+$CXCR5$^+$Foxp3$^-$ Tfh cells and CXCR5 MFI on Bcl6$^+$PD-1$^+$Foxp3$^-$ Tfh cells isolated from the draining iliac of adult and aged mice 14 days after immunization with NP-KLH/Alum (n = 18). (d) Median fluorescence intensity of CXCR4 on PD1$^+$CXCR5$^+$Foxp3$^-$ Tfh cells and CXCR5 MFI on Bcl6$^+$PD-1$^+$Foxp3$^-$ Tfh cells isolated from the draining iliac LN (left) and spleen (right) of adult and aged mice 9 days after immunization with

the Oxford/AstraZeneca COVID-19 vaccine candidate ChAOx1-nCoV19 (n = 14). (e) Median fluorescence intensity of CXCR4 on PD1$^+$CXCR5$^+$Foxp3$^-$ Tfh cells and CXCR5 MFI on Bcl6$^+$PD-1$^+$Foxp3$^-$ Tfh cells isolated from the draining mediastinal LN (left) and spleen (right) of adult and aged mice 14 days after infection with influenza (n = 11). (f) Flow cytometric quantification of the percentage (left) of CXCR4$^+$ cells and CXCR4 MFI (right) from CD45RA$^-$CD4$^+$ T cells from human peripheral blood mononuclear cells (n = 42). (g) Flow cytometric quantification of the %CXCR4$^+$ cells and CXCR4 MFI from CXCR5$^+$CD45RA$^-$CD4$^+$ circulating Tfh cells, and %CXCR5$^+$ cells and CXCR5 MFI on ICOS$^+$CD38$^+$CD45RA$^-$CD4$^+$ circulating Tfh cells from human peripheral blood mononuclear cells seven days after seasonal influenza vaccination (n = 34). Bar heights indicate median, each symbol represents a biological replicate and p-values were obtained by performing an unpaired, two-tailed Mann-Whitney U test. In (a-b) each symbol represents an animal, with each mouse connected by a line. P-value is from a paired non-parametric t-test.

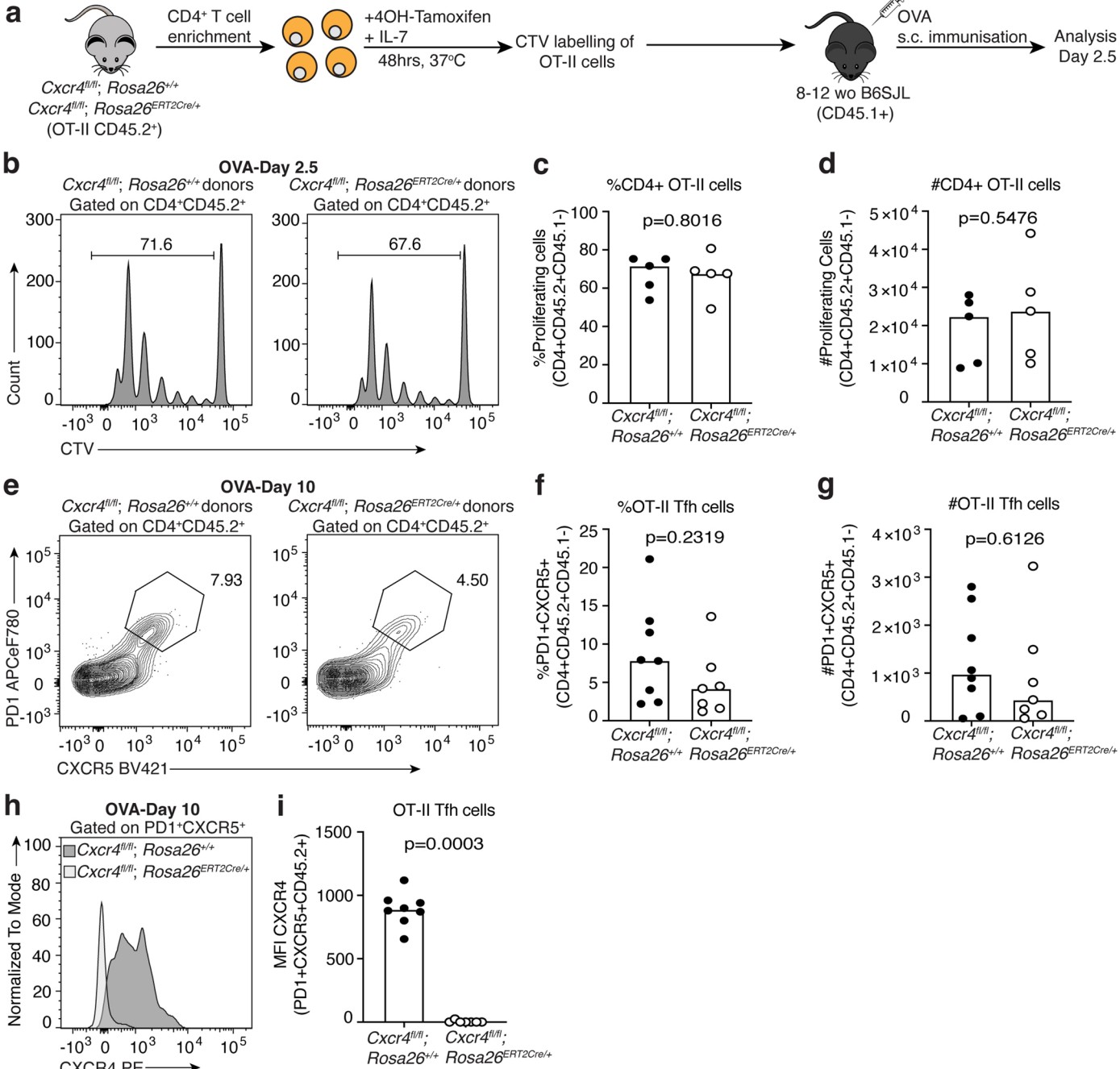

**Extended Data Fig. 6 | Deletion of CXCR4 on T cells does not impair activation and Tfh differentiation. (a)** Experimental outline of *in vitro* 4-OH tamoxifen treatment of CD4⁺ T cells isolated from *Cxcr4^fl/fl; Rosa26^ERT2Cre/+* OT-II mice which were treated with 200 nM of 4-OH tamoxifen for 48 hours after which the cells were labelled with cell trace violet (CTV) and transferred into adult B6SJL recipient mice. Recipient mice were immunized subcutaneously with OVA in alum and analysis was performed after 2.5 days. **(b)** Representative flow cytometry plots indicating CTV dilution by activated CD4⁺CD45.2⁺ T cells derived from control *Cxcr4^fl/fl; Rosa26^ERT2+/+* (left) or *Cxcr4^fl/fl; Rosa26^ERT2Cre/+* (right) OT-II mice. Quantification of the percentage **(c)** and total number **(d)** of proliferating CD4⁺CD45.2⁺ T cells that diluted CTV (n = 10). **(e)** Representative flow cytometry plots identifying PD1⁺CXCR5⁺CD4⁺CD45.2⁺ Tfh cells derived from the transferred

*Cxcr4^fl/fl; Rosa26^ERT2+/+* (left) or *Cxcr4^fl/fl; Rosa26^ERT2Cre/+* (right) OT-II cells in the iLNs of recipient mice at day 10 after subcutaneous OVA immunization. Quantification of the percentage **(f)** and total number **(g)** of PD1⁺CXCR5⁺CD4⁺CD45.2⁺ Tfh cells derived from the transferred *Cxcr4^fl/fl; Rosa26^ERT2+/+* or *Cxcr4^fl/fl; Rosa26^ERT2Cre/+* OT-II cells. **(h)** Representative flow cytometry plot indicating the expression of CXCR4 by PD1⁺CXCR5⁺CD4⁺CD45.2⁺ Tfh cells derived from the transferred *Cxcr4^fl/fl; Rosa26^ERT2+/+* or *Cxcr4^fl/fl; Rosa26^ERT2Cre/+* OT-II cells. **(i)** Median fluorescence intensity (MFI) of CXCR4 by PD1⁺CXCR5⁺CD4⁺CD45.2⁺ Tfh cells derived from the transferred *Cxcr4^fl/fl; Rosa26^ERT2+/+* or *Cxcr4^fl/fl; Rosa26^ERT2Cre/+* OT-II cells (n = 15). Bar height on graphs is indicative of the median, each symbol represents a mouse and p-values were obtained by performing an unpaired, two-tailed Mann-Whitney U test. The data are representative of two independent experiments.

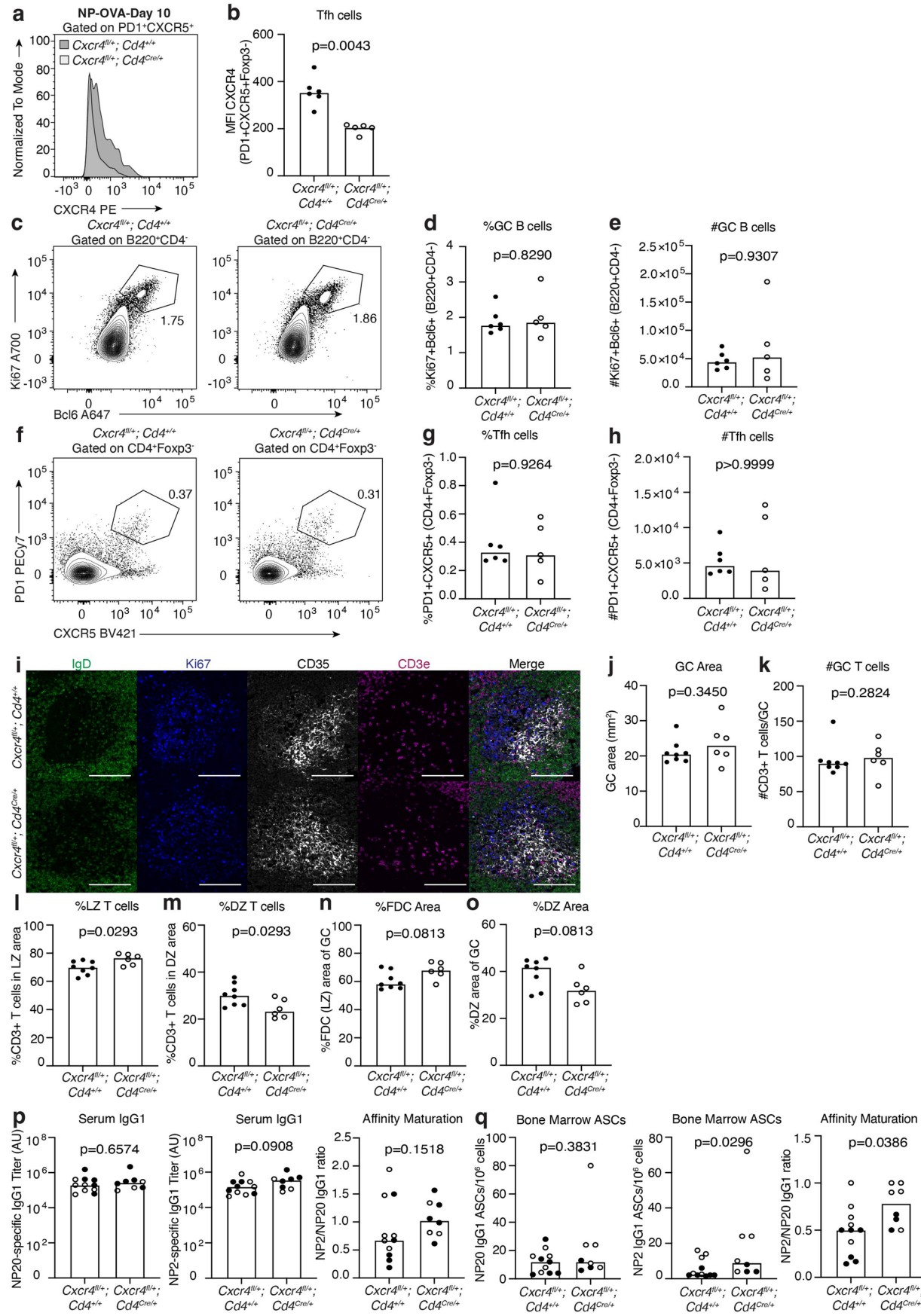

**Extended Data Fig. 7 | See next page for caption.**

**Extended Data Fig. 7 | T cell-specific heterozygosity of CXCR4.** Flow cytometry plot indicating the expression of CXCR4 **(a)** and median fluorescence intensity (MFI) of CXCR4 **(b)** by PD1$^+$CXCR5$^+$Foxp3$^-$ Tfh cells in iLNs of *Cxcr4$^{fl/+}$;Cd4$^{+/+}$* or *Cxcr4$^{fl/+}$;Cd4$^{Cre/+}$* mice 10 days after NP-OVA/Alum immunisation. Flow cytometry plots **(c)** and quantification **(d, e)** of Ki67$^+$Bcl6$^+$ GC B cells 10 days after NP-OVA/Alum immunisation of *Cxcr4$^{fl/+}$;Cd4$^{+/+}$* and *Cxcr4$^{fl/+}$;Cd4$^{Cre/+}$* mice. Flow cytometry plots **(f)** and quantification **(g, h)** PD1$^+$CXCR5$^+$ Tfh cells 10 days after NP-OVA immunisation of *Cxcr4$^{fl/+}$;Cd4$^{+/+}$* and *Cxcr4$^{fl/+}$;Cd4$^{Cre/+}$* mice (n = 11). **(i)** Representative 20x confocal images of GCs at day 10 after NP-OVA immunisation in the iLNs of *Cxcr4$^{fl/+}$;Cd4$^{+/+}$* mice (top) and *Cxcr4$^{fl/+}$;Cd4$^{Cre/+}$* mice (bottom); scale bar is 100 μm. IgD (green), Ki67 (blue), CD35 (white) and CD3e (magenta). Quantification of the total GC area **(j)**, the total number of CD3$^+$ T cells within the GC **(k)**, the percentage of CD3$^+$ T cells localising to the CD35$^+$ light zone area **(l)** and the Ki67$^+$CD35$^-$ dark zone area **(m)**, the CD35$^+$ FDC network light zone area **(n)** and the Ki67$^+$CD35$^-$ dark zone area **(o)** in iLNs of *Cxcr4$^{fl/+}$;Cd4$^{+/+}$* and *Cxcr4$^{fl/+}$;Cd4$^{Cre/+}$* mice (n = 14). **(p)** Serum titres of NP20 and NP2 specific IgG1 and their ratio in *Cxcr4$^{fl/+}$;Cd4$^{+/+}$* and *Cxcr4$^{fl/+}$;Cd4$^{Cre/+}$* mice 35 days post-immunisation with NP-OVA/Alum. **(q)** Enumeration of NP20 (left) and NP2 (middle) IgG1 antibody-secreting cells (ASCs) and their ratio in the bone marrow of *Cxcr4$^{fl/+}$;Cd4$^{+/+}$* and *Cxcr4$^{fl/+}$;Cd4$^{Cre/+}$* mice at 35 days post-immunisation with NP-OVA. Data on **(p)** and **(q)** are pooled from two independent experiments and symbol colour represents different experiments (n = 19). Bar height indicates the median, each symbol represents a mouse, and p-values from an unpaired, two-tailed Mann-Whitney U test. Data are representative of two independent experiments.

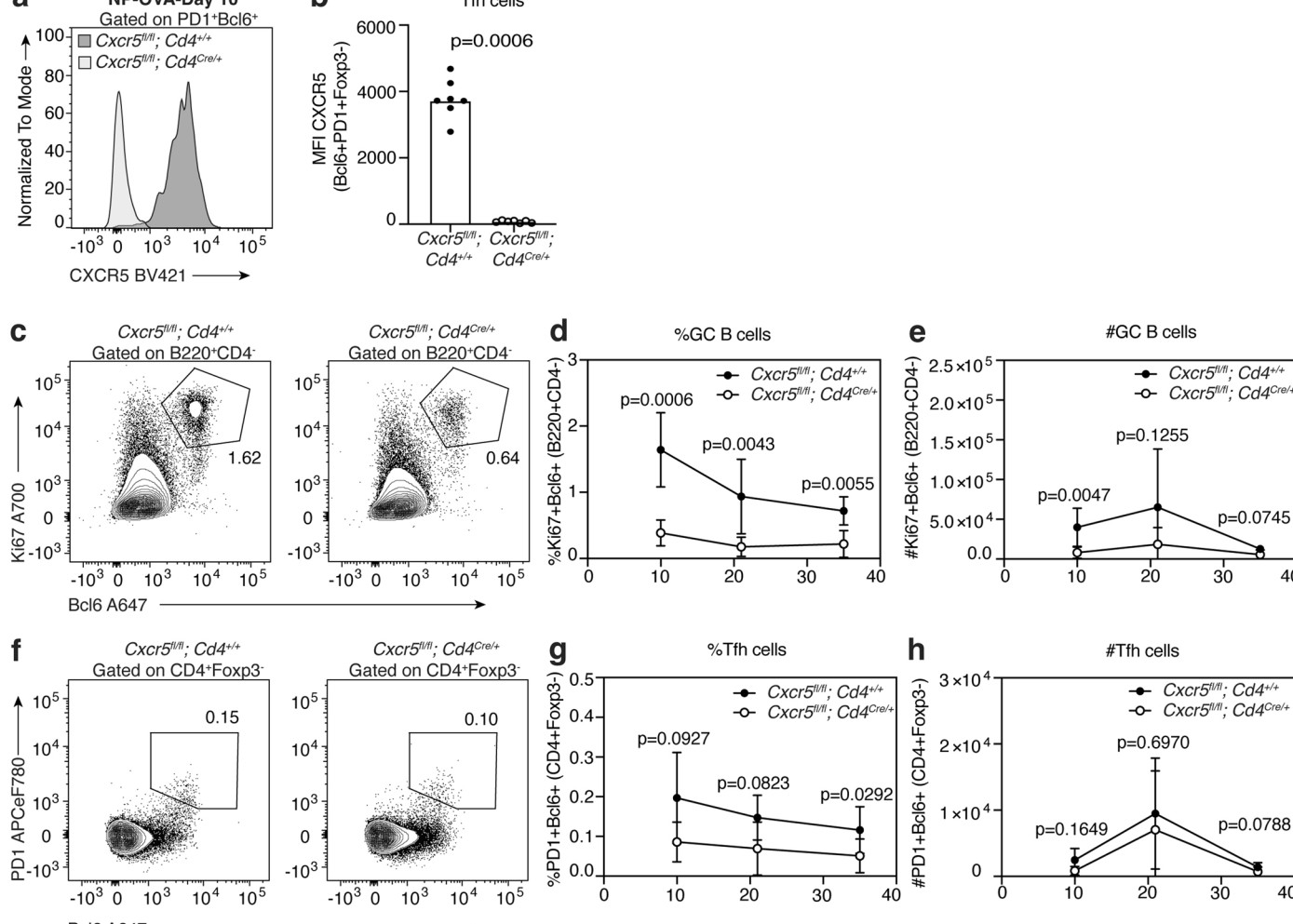

**Extended Data Fig. 8 | CD4⁺ T cell-specific deletion of CXCR5, flow cytometric analysis of immunized *Cxcr5*^fl/fl^; *Cd4*^Cre/+^ mice. (a)** Representative flow cytometry plot indicating the expression of CXCR5 by PD1⁺Bcl6⁺Foxp3⁻ Tfh cells in iLNs of *Cxcr5*^fl/fl^; *Cd4*^+/+^ or *Cxcr5*^fl/fl^; *Cd4*^Cre/+^ mice 10 days after NP-OVA immunization. **(b)** Median fluorescence intensity (MFI) of CXCR5 by PD1⁺Bcl6⁺Foxp3⁻ Tfh cells of *Cxcr5*^fl/fl^; *Cd4*^+/+^ or *Cxcr5*^fl/fl^; *Cd4*^Cre/+^ mice (n = 14). Bar height on graphs is indicative of the median, each symbol represents a mouse and p-values were obtained by performing an unpaired, two-tailed Mann-Whitney U test. The data are representative of two independent experiments. **(c)** Representative flow cytometry plots identifying Ki67⁺Bcl6⁺ GC B cells 10 days after NP-OVA immunization of *Cxcr5*^fl/fl^; *Cd4*^+/+^ (left) and *Cxcr5*^fl/fl^; *Cd4*^Cre/+^

(right) mice; values adjacent to gates indicate percentages. Quantification of the percentage **(d)** and total number **(e)** of Ki67⁺Bcl6⁺ GC B cells at days 10, 21 and 35 after NP-OVA immunization of *Cxcr5*^fl/fl^; *Cd4*^+/+^ and *Cxcr5*^fl/fl^; *Cd4*^Cre/+^ mice. **(f)** Representative flow cytometry plots identifying PD1⁺Bcl6⁺ Tfh cells 10 days after NP-OVA immunization of *Cxcr5*^fl/fl^; *Cd4*^+/+^ (left) and *Cxcr5*^fl/fl^; *Cd4*^Cre/+^ (right) mice; values adjacent to gates indicate percentages. Quantification of the percentage **(g)** and total number **(h)** of PD1⁺Bcl6⁺ Tfh cells at days 10, 21 and 35 after NP-OVA immunization of *Cxcr5*^fl/fl^; *Cd4*^+/+^ and *Cxcr5*^fl/fl^; *Cd4*^Cre/+^ mice. The data is representative of two independent experiments (n = 14) where each symbol represents the mean ± SD and p-values were generated by performing a two-way ANOVA with Sidak's multiple comparisons test.

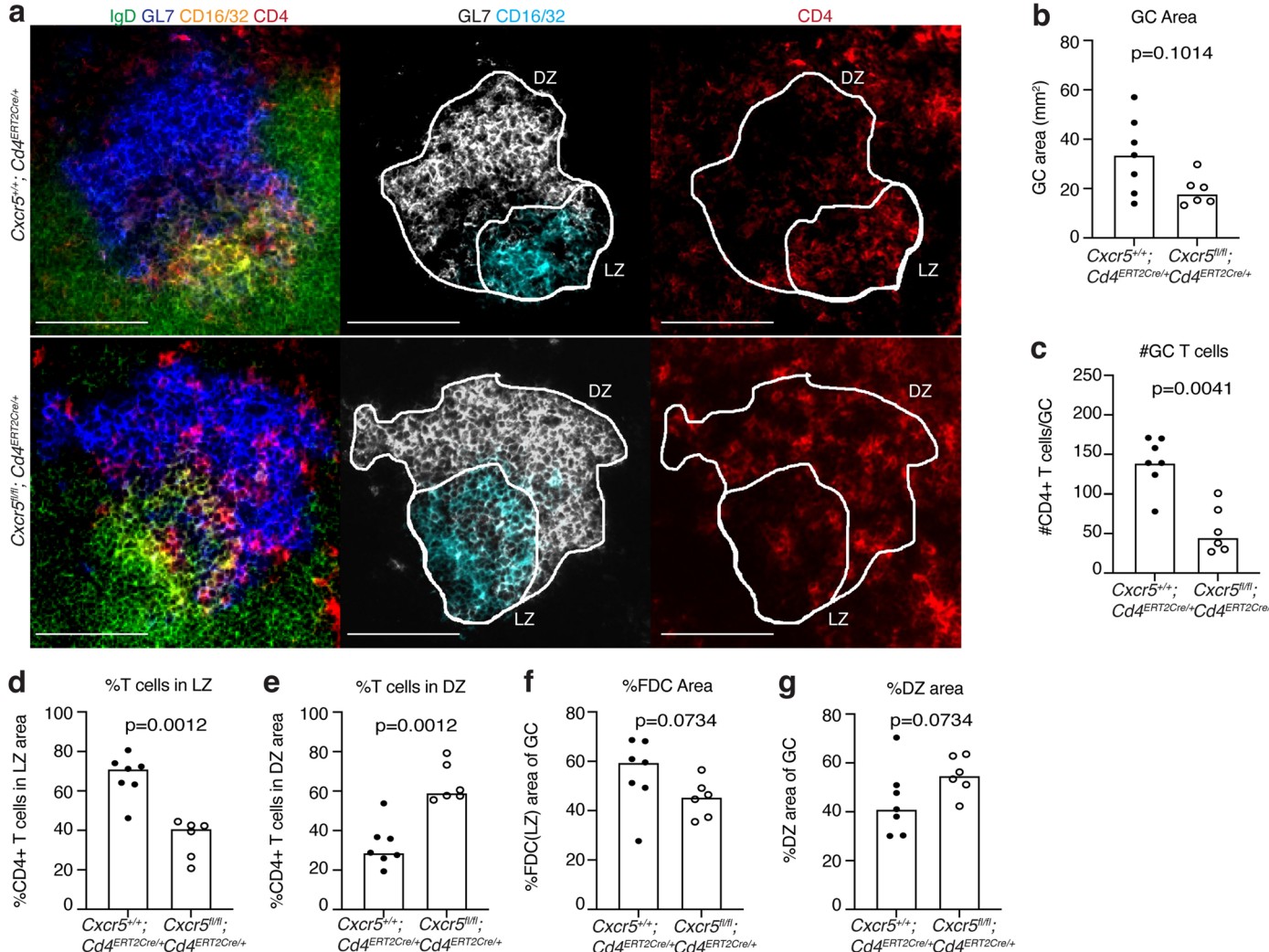

**Extended Data Fig. 9 | Imaging of GCs from *Cxcr5^fl/fl^; Cd4^ERT2Cre/+^* mice after immunization.** (**a**) Representative confocal images of GCs at day 14 after NP-KLH/Alum immunization in the iLNs of control *Cxcr5^+/+^; Cd4^ERT2Cre/+^* mice (left) and *Cxcr5^fl/fl^; Cd4^ERT2Cre/+^* mice (right). Scale bars are 100 μm. LN sections were stained for IgD (green), GL7 (blue), CD16/32 (yellow) and CD4 (red). The outline of the region of interest of the GC used for analysis is outlined in white. (**b**) Quantification of the total area of GCs identified by immunofluorescence as GL7^+^CD16/32^+^IgD^−^ regions in the iLNs of *Cxcr5^+/+^; Cd4^ERT2Cre/+^* mice and *Cxcr5^fl/fl^; Cd4^ERT2Cre/+^* mice. (**c**) Quantification of the total number of CD4^+^ T cells within the GL7^+^CD16/32^+^IgD^−^GC area of *Cxcr5^+/+^; Cd4^ERT2Cre/+^* mice and *Cxcr5^fl/fl^; Cd4^ERT2Cre/+^* mice. (**d**) Percentage of CD4^+^ T cells localizing to the CD16/32^+^ light zone area of

the GC in *Cxcr5^+/+^; Cd4^ERT2Cre/+^* mice and *Cxcr5^fl/fl^; Cd4^ERT2Cre/+^* mice. (**e**) Percentage of CD4^+^ T cells localizing to the GL7^+^CD16/32^−^ dark zone area of the GC in *Cxcr5^+/+^; Cd4^ERT2Cre/+^* mice and *Cxcr5^fl/fl^; Cd4^ERT2Cre/+^* mice. (**f**) Quantification of the CD16/32^+^ FDC network area representative of the light zone compartment of the GCs of *Cxcr5^+/+^; Cd4^ERT2Cre/+^* mice and *Cxcr5^fl/fl^; Cd4^ERT2Cre/+^* mice. (**g**) Quantification of the GL7^+^CD16/32^−^ dark zone area within the GCs of *Cxcr5^+/+^; Cd4^ERT2Cre/+^* mice and *Cxcr5^fl/fl^; Cd4^ERT2Cre/+^* mice. Quantification was performed using ImageJ. Bar heights on graphs represent the median, each symbol represents a mouse and p-values were obtained by performing an unpaired, two-tailed Mann-Whitney U test (n = 13 biologically independent samples).

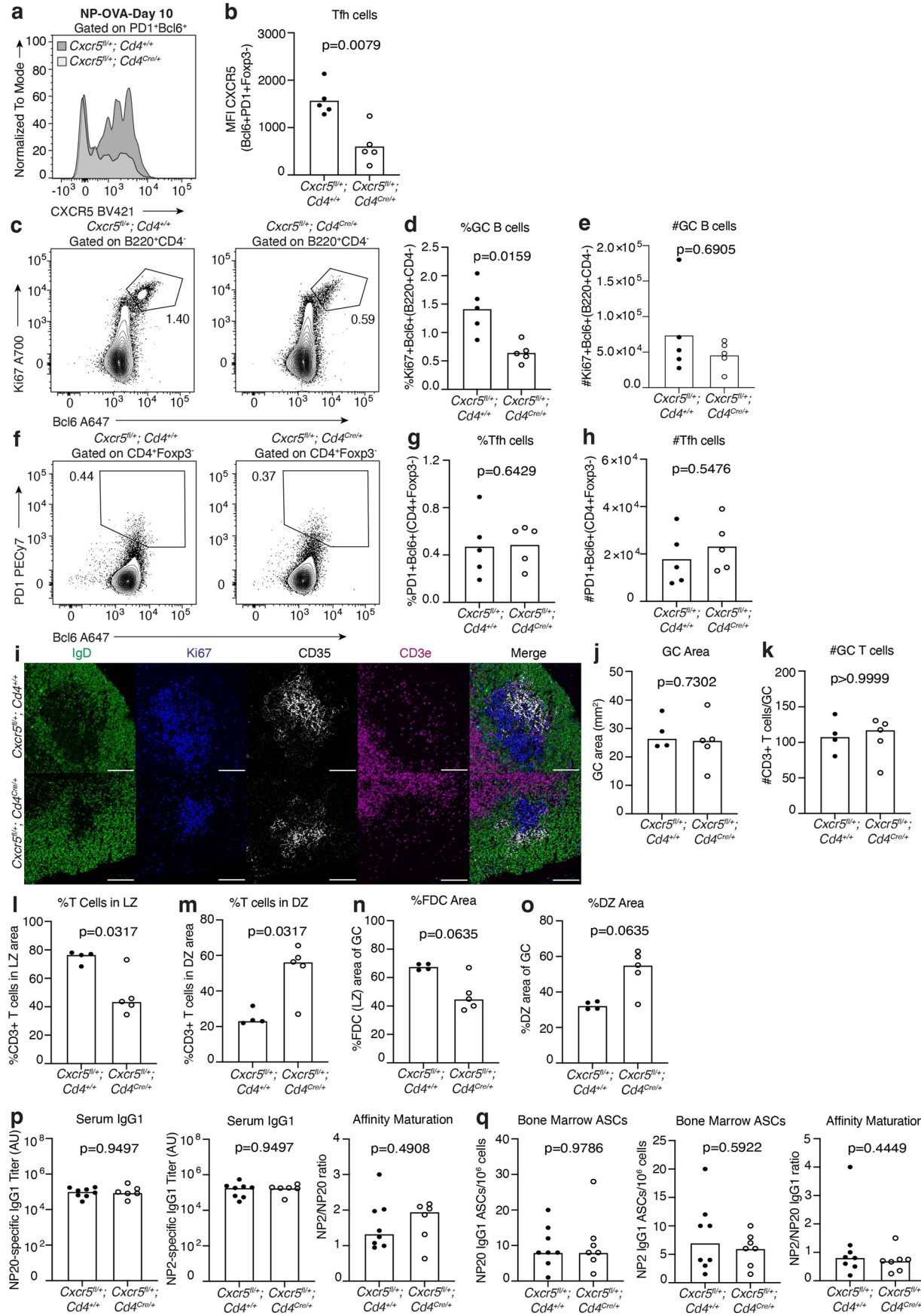

**Extended Data Fig. 10 | See next page for caption.**

**Extended Data Fig. 10 | T cell-specific heterozygosity of CXCR5.** Flow cytometry plot of CXCR5 **(a)** and its median fluorescence intensity (MFI) **(b)** by PD1⁺Bcl6⁺Foxp3⁻ Tfh cells in iLNs of *Cxcr5^{fl/+}; Cd4^{+/+}* or *Cxcr5^{fl/+}; Cd4^{Cre/+}* mice 10 days after NP-OVA/Alum immunisation. Flow cytometry plots **(c)** and quantification **(d, e)** of Ki67⁺Bcl6⁺ GC B cells, and flow cytometry plots **(f)** and quantification **(g, h)** of PD1⁺Bcl6⁺ Tfh cells 10 days after NP-OVA immunisation of *Cxcr5^{fl/+}; Cd4^{+/+}* and *Cxcr5^{fl/+}; Cd4^{Cre/+}* mice (n = 11). **(i)** Representative 20x confocal images of GCs at day 10 after NP-OVA immunisation in the iLNs of control *Cxcr5^{fl/+}; Cd4^{+/+}* mice (top) and *Cxcr5^{fl/+}; Cd4^{Cre/+}* mice (bottom); scale bar is 100 µm. IgD (green), Ki67 (blue), CD35 (white) and CD3e (magenta). Quantification of the total GC area **(j)**, the total number of CD3⁺ T cells within the GC **(k)**, the percentage of CD3⁺ T cells localising to the CD35⁺ light zone area **(l)** and the

Ki67⁺CD35⁻ dark zone area **(m)**, the CD35⁺ FDC network light zone area **(n)** and the Ki67⁺CD35⁻ dark zone area **(o)** in iLNs of *Cxcr5^{fl/+}; Cd4^{+/+}* and *Cxcr5^{fl/+}; Cd4^{Cre/+}* mice (n = 14). **(p)** Serum titres of NP20 and NP2 specific IgG1, and their ratio of *Cxcr5^{fl/+}; Cd4^{+/+}* and *Cxcr5^{fl/+}; Cd4^{Cre/+}* mice 35 days post-immunisation with NP-OVA. **(q)** Enumeration of NP20 (left) and NP2 (middle) IgG1 antibody-secreting cells (ASCs) and affinity maturation indicated by the ratio of NP2/NP20 ASCs (right) in the bone marrow of *Cxcr5^{fl/+}; Cd4^{+/+}* and *Cxcr5^{fl/+}; Cd4^{Cre/+}* mice at 35 days post-immunisation with NP-OVA. Data on **(p)** and **(q)** are pooled from two independent experiments and symbol colour represents different experiments (n = 19). For all bar graphs, bar height indicates the median, each symbol represents a mouse, and p-values were obtained by performing an unpaired, two-tailed Mann-Whitney U test. Data are representative of two independent experiments.

# Reporting Summary

## Statistics

For all statistical analyses, confirm that the following items are present in the figure legend, table legend, main text, or Methods section.

| n/a | Confirmed | |
|---|---|---|
| ☐ | ☒ | The exact sample size (*n*) for each experimental group/condition, given as a discrete number and unit of measurement |
| ☐ | ☒ | A statement on whether measurements were taken from distinct samples or whether the same sample was measured repeatedly |
| ☐ | ☒ | The statistical test(s) used AND whether they are one- or two-sided *Only common tests should be described solely by name; describe more complex techniques in the Methods section.* |
| ☒ | ☐ | A description of all covariates tested |
| ☐ | ☒ | A description of any assumptions or corrections, such as tests of normality and adjustment for multiple comparisons |
| ☐ | ☒ | A full description of the statistical parameters including central tendency (e.g. means) or other basic estimates (e.g. regression coefficient) AND variation (e.g. standard deviation) or associated estimates of uncertainty (e.g. confidence intervals) |
| ☐ | ☒ | For null hypothesis testing, the test statistic (e.g. *F*, *t*, *r*) with confidence intervals, effect sizes, degrees of freedom and *P* value noted *Give P values as exact values whenever suitable.* |
| ☒ | ☐ | For Bayesian analysis, information on the choice of priors and Markov chain Monte Carlo settings |
| ☒ | ☐ | For hierarchical and complex designs, identification of the appropriate level for tests and full reporting of outcomes |
| ☒ | ☐ | Estimates of effect sizes (e.g. Cohen's *d*, Pearson's *r*), indicating how they were calculated |

*Our web collection on statistics for biologists contains articles on many of the points above.*

## Software and code

Policy information about availability of computer code

| Data collection | For collection of flow cytometry data the following software was used: BD FACSDiva Software v9.0 and SpectroFlo Software v3.0<br>For acquisition of confocal microscopy images Zen Microscopy Software v3.2-3.5 was used.<br>For acquisition of ELISPOT images the ImmunoSpot version 5.0 CTL Cellular Technology Ltd Software was used.<br>For acquisition of raw ELISA absorbance values the PHERAstar FSX software v5.7 was used. |
|---|---|
| Data analysis | For analysis of flow cytometry data FlowJo v10.0 software was used.<br>For analysis and quantification of confocal images ImageJ (Fiji) v2.0.0-rc-69/1.52p and Cell Profiler v3.19 were used.<br>Graphs and statistics were generated using Graphpad Prism v6-9<br>Graphs and statistics for in silico modeling were generated using GLE v4.2<br>The code used for in silico modeling is available at: https://gitlab.com/simm/gc/hyphasma/-/releases/Denton2020<br>Analysis of W33L VH186.2 sequencing analysis was done using a custom pipeline on R v4.0.4<br>The code used for the W33L VH186.2 sequencing analysis is available at: https://github.com/lintermanlab/W33L_caller |

For manuscripts utilizing custom algorithms or software that are central to the research but not yet described in published literature, software must be made available to editors and reviewers. We strongly encourage code deposition in a community repository (e.g. GitHub). See the Nature Portfolio guidelines for submitting code & software for further information.

# Data

Policy information about availability of data

All manuscripts must include a data availability statement. This statement should provide the following information, where applicable:

- Accession codes, unique identifiers, or web links for publicly available datasets
- A description of any restrictions on data availability
- For clinical datasets or third party data, please ensure that the statement adheres to our policy

Data availability
Source data are provided with this paper.

Code availability
The code used for in silico modelling is available at: https://gitlab.com/simm/gc/hyphasma/-/releases/Denton2020.  The code used for the W33L VH186.2 sequencing analysis is available at: https://github.com/lintermanlab/W33L_caller.

# Human research participants

Policy information about studies involving human research participants and Sex and Gender in Research.

| | |
|---|---|
| Reporting on sex and gender | Researchers were blinded to the sex or gender of research participants; the effect of sex or gender has not been investigated in this study. |
| Population characteristics | The study population is comprised of two comparison groups: 39 "aged" participants (between 66-98 years of age; mean age = 73.4) and 37 "young" samples (18 - 36 years of age; mean age = 26.5). 58% of participants self-identify as female. All participants were healthy, disease-naive adults. |
| Recruitment | Adult volunteers were recruited to this study on the basis of age. Peripheral blood was collected from healthy UK adults recruited through the NIHR Bioresource prior to and seven days after the seasonal influenza vaccine[26,77]. Samples were collected between October-February 2014-15 and October-December 2016, n = 37 participants 18–36 years old, n = 39 participants 66–98 years old.  Volunteers were recruited through the NIHR BioResource Cambridge, which provided the infrastructure for the screening, recruitment and collection of these samples. Participants identified by NIHR BioResource Cambridge were sent an information sheet and asked to indicate their willingness to participate, as per their usual procedure. Written and informed consent was collected and retained by NIHR BioResource Cambridge. We are not aware of any biases, self-selected or otherwise, that may impact the results presented here. |
| Ethics oversight | Human samples were collected in accordance with the latest revision of the Declaration of Helsinki and the Guidelines for Good Clinical Practice (ICH-GCP). Informed consent was obtained from all participants. The samples collected with UK local research ethics committee approval (NRES Committee South Central - Hampshire A, REC reference 14/SC/1077), using the facilities of the NIHR Cambridge Bioresource (REC reference 04/Q0108/44). |

Note that full information on the approval of the study protocol must also be provided in the manuscript.

# Field-specific reporting

Please select the one below that is the best fit for your research. If you are not sure, read the appropriate sections before making your selection.

☒ Life sciences ☐ Behavioural & social sciences ☐ Ecological, evolutionary & environmental sciences

For a reference copy of the document with all sections, see nature.com/documents/nr-reporting-summary-flat.pdf

# Life sciences study design

All studies must disclose on these points even when the disclosure is negative.

| | |
|---|---|
| Sample size | All mouse experiments were performed with 4-10 mice per group and for aging studies a minimum of 5 aged mice per group were used. Sample size was chosen based on our previous experience working with genetically modified and aged mice, as well as the availability of aged mice or mice of appropriate genotypes. |
| Data exclusions | All data points were analysed including outliers unless there were technical errors. Any aged mice with lymphoma and/or solid tumors were also excluded from analysis. |
| Replication | All experiments were successfully replicated 2-4 times, the data shown is representative of these replicates. |
| Randomization | Where possible, mice were randomly allocated into aged and sex-matched experimental groups by staff of the Babraham Institute Biological Services Unit.  For human blood samples, participants were recruited to the study on the basis of age aiming to recruit an equal number of |

younger and older adults, and samples coded by the NIHR BioResource and analysed in the laboratory. All available samples were used for this study.

| Blinding | For experiments with aged mice, blinding was not possible due to phenotypic differences between adult and aged mice. For experiments with human blood samples, samples were coded by the NIHR BioResource and run blinded. Unblinding was done after data analysis to enable statistical comparisons. |

# Reporting for specific materials, systems and methods

We require information from authors about some types of materials, experimental systems and methods used in many studies. Here, indicate whether each material, system or method listed is relevant to your study. If you are not sure if a list item applies to your research, read the appropriate section before selecting a response.

## Materials & experimental systems

| n/a | Involved in the study |
|-----|----------------------|
| ☐ | ☒ Antibodies |
| ☒ | ☐ Eukaryotic cell lines |
| ☒ | ☐ Palaeontology and archaeology |
| ☐ | ☒ Animals and other organisms |
| ☒ | ☐ Clinical data |
| ☒ | ☐ Dual use research of concern |

## Methods

| n/a | Involved in the study |
|-----|----------------------|
| ☒ | ☐ ChIP-seq |
| ☐ | ☒ Flow cytometry |
| ☒ | ☐ MRI-based neuroimaging |

## Antibodies

| Antibodies used | B220 APCFire810 RA3-6B2 BioLegend Cat#103278 Lot#B323073 (1/400)
B220 BV510 RA3-6B2 BioLegend Cat#103247 Lot#B2603705 (1/400)
B220 BV785 RA3-6B2 BioLegend Cat# 103246 Lot#B338778 (1/400)
Bcl6 A647 K112-91 BD Cat# 561525 Lot#9277097 (1/100)
Bcl-6 BV421 K112-91 BD Cat#563363 (1/100)
Bcl-6 PE-Cy7 K112-91 BD Cat# 563582 Lot#9050793 (1/100)
CD4 APC GK1.5 eBioscience Cat#17-0041-83 Lot#E07038-1635 (1/100)
CD4 BUV395 GK1.5 BD Cat#563790 Lot#0066907 (1/1000)
CD4 BUV496 GK1.5 BD Cat#612952 Lot#0080998 (1/1000)
CD4 BV510 RM4-5 BioLegend Cat#100559 Lot#B252816 (1/200)
CD4 BV605 RM4-5 BioLegend Cat#100547 Lot#B2667915 (1/200)
CD4 PE-Fire640 GK1.5 BioLegend Cat#100482 Lot#B315086 (1/2000)
CD4 SV538 GK1.5 BioLegend Cat#100485 Lot#B329034 (1/2000)
CD19 BUV661 1D3 BD Cat#612971 Lot#1085567 (1/2000)
CD31 PercpCy5.5 MEC13.3 BioLegend Cat#102522 Lot#B284200 (1/100)
CD38 eF450 90 eBioscience Cat# 48-03-81-82Lot#1934068 (1/100)
CD38 PercP-Cy5.5 90 eBioscience Cat# 102722 Lot#B341634 (1/1000)
CD44 BV510 IM7 BioLegend Cat#103044 Lot#B303963 (1/1000)
CD44 Percp-Cy5.5 IM7 BioLegend Cat#103032 Lot#B277590 (1/1000)
CD45 BV510 30-F11 BioLegend Cat#103138 Lot#B333193 (1/100)
CD45.1 A700 A20 BioLegend Cat#110724 Lot#B254605 (1/100)
CD45.1 PE eFluor610 A20 eBioscience Cat#61-0453-82 Lot#2024809 (1/100)
CD45.1 PE-Cy7 A20 eBioscience Cat#25-0453-82 Lot#2055156 (1/100)
CD45.2 A700 104 eBioscience  Cat#56-0454-82 Lot#2123867 (1/100)
CD45.2 Percp-Cy5.5 104 eBioscience Cat#45-0454-82 Lot#4336370 (1/100)
CD54 (ICAM) PacBlue YN1/1.7.4 BioLegend Cat#116116 Lot#B241838 (1/100)
CD86 APC GL-1 eBioscience Cat#17-0862-82 Lot#19954468 (1/100)
CD90.2 AF790 30-H12 BioLegend Conjugated in-house (1/1000)
CD95 BV510 Jo2 BD Cat#563646 Lot#9325255 (1/1000)
CD95 PE-Cy7 Jo2  BD Cat#557653 Lot#7174919 (1/1000)
CXCR4 APC L276F12 BioLegend Cat#146508 Lot#B278598 (1/200)
CXCR4 BUV563 2B11 BD Cat#741313 Lot#1302258 (1/200)
CXCR4 PE L276F12 BioLegend Cat#146506 Lot#B242855(1/200)
CXCR4 Percp-eF710 2B11 eBioscience Cat#46-9991-82 Lot#2139606 (1/200)
CXCR5 APC L138D7 BioLegend Cat#145506 Lot#B270112 (1/100)
CXCR5 BV421 L138D7 BioLegend Cat#145512 Lot#B281252 (1/100)
CXCR5 BV785 L138D7 BioLegend Cat#145523 Lot#B310538 (1/100)
EpCAM-1 BV711 G8.8 BioLegend Cat#118233 Lot#B306636 (1/100)
Foxp3 APC FJK-16S eBioscience Cat#17-5773-82 Lot#1984797 (1/400)
Foxp3 AF488 FJK-16s eBioscience Cat#53-5773-82 Lot#1931448 (1/400)
Foxp3 PE-Cy5.5 FJK-16s eBioscience Cat#35-5773-82 Lot#2248878 (1/400)
GL7 AF488 GL7 eBioscience Cat#53-5902-82 Lot#2312432 (1/1000)
GR1 APC RB6-8C5 eBioscience Cat#56-5931-80 Lot#1920394 (1/100)
Gp38 APC 8.1.1 BioLegend Cat#127410 Lot#B268805 (1/100)
IgG1 BV605 A85-1 BD Cat#563285 Lot#0325234 (1/400) |

IgM  APC R6-60.2 BD Cat#562032 (1/100)
Ki67 AF700 SolA15 eBioscience Cat#56-5698-82 Lot#2261476 (1/1000)
Ki67 APCeF780 SolA15 eBioscience Cat#47-5698-82 (1/1000)
Ki67 FITC SolA15 eBioscience Cat#11-5698-82 Lot#2191034 (1/1000)
MadCAM-1 PE MECA-367 BioLegend Cat#120709 Lot#B299430 (1/100)
CD21/35 PE-Cy7 eBio8D9 eBioscience Cat#25-0211-82 Lot#2284271 (1/1000)
PD-1 FITC RMP1-30 BioLegend Cat#11-9981-81 Lot#2009760 (1/1000)
PD-1 APC eF780 J43 eBioscience Cat#47-9985-82 Lot#2194336 (1/1000)
PD-1 BUV615 RMP1-30 BD Cat#752354 Lot#1130154 (1/1000)
PD-1 PE-Cy7 RMP1-30 BioLegend Cat#109110 Lot#B263885 (1/2000)
TCRVa2 APC B20.1 eBioscience Cat#17-5812-82 Lot# 2142951 (1/100)
Cell Trace Violet eF450 - eBioscience Cat#C34557 (1/1000)
NP PE - Biosearch Technologies Cat#N-5070-1 (1/100)
Cell viability dye Aqua (525/50) - Cat#L34957 eBioscience (1/1000)
Cell Viability Dye Blue (450/50) - Cat#L23105 eBioscience (1/1000)

IgD FITC 11-26c.2a BioLegend Cat#405704 Lot#B271924 (1/400)
IgD AF647 11-26c.2a BioLegend Cat#405708 Lot#B259275 (1/200)
CD3e APC 17A2 BioLegend Cat#100236 Lot#B321239 (1/200)
CD3e Purified 500A Thermo Fisher Cat#14-0033-85 Lot#E03452-1633 (1/200)
CD35 Biotin 8C12 BD Cat#553816 Lot#7200747 (1/400)
Ki67 Purified Polyclonal Abcam Cat#ab15580 (1/200)
Ki67 FITC SolA15 Thermo Fisher Cat#11-5698-82 Lot#2191034 (1/200)
Ki67 eF450 SolA15 eBioscience Cat#48-5698-83 Lot#1998365 (1/100)
CD45.2 A700 104 eBioscience Cat#56-0454-82 Lot#2123867 (1/200)
CD45.1 BV605 A20 BioLegend Cat#110738 Lot#B267743 (1/200)
IgG1 (isotype) AF647 11711 R&D Cat#IC002R Lot#AEIX0219031 (1/200)
CXCL12/SDF-1 AF647 79018 R&D Cat#FAB350R Lot#1561071 (1/200)
CD35 Purified 8C12 BD Cat#558768 Lot#7236858 (1/200)
IgD AF488 11-26 SouthernBiotech Cat#1120-30 Lot#G0815-M329B (1/200)
Goat anti-Rat IgG (H+L) Cross-Adsorbed Secondary Antibody, AF555 Polyclonal Thermo Cat# A-21434 Lot#2089884 (1/1000)
Goat anti-Rat IgG (H+L) Cross-Adsorbed Secondary Antibody, AF568 Polyclonal Thermo Fisher Cat#A11077 Lot#870966 (1/1000)
Streptavidin AF568 - Thermo Fisher Cat#S11226 Lot#2045314 (1/1000)
Streptavidin BV421 - BioLegend Cat#405225 Lot#B279623 (1/1000)
IgD AF488 11-26c.2a BioLegend Cat#405718 (1/200)
CD16/32 BV421 190909 BD Cat#562896 (1/200)
CD4 Dylight550 GK1.4 BIOTREND Cat#C1637-100 (1/200)
GL7 AF647 GL7  Biolegend Cat#144605 (1/200)

CXCR4 PE/Cy5 12G5 BioLegend Cat#306507 (1/200)
CD45RA SB570 F8-11-13 BioRad Cat#MCA88SBV570 (1/200)
CD4 BUV496 M-T477 BD Biosciences Cat#50175 (1/200)
CD3 SparkBlue550 SK7 BioLegend Cat#344851 (1/400)
CD19 BUV615 HIB19 BD Biosciences Cat#751273 (1/200)
Cell viability dye ViaKrome808 - Beckman Coulter Cat#C36628 (1/2000)

| | |
|---|---|
| Validation | All antibodies were validated by the manufacturer, for the species and the specified application (flow cytometry or imaging) for which they were used in this study. Antibodies used for flow cytometry were titrated prior to use and fluorescence minus one (FMO) controls were included to determine true positive staining. Antibodies used for microscopy were also titrated for optimal staining and both FMO controls and isotype controls were included for validation. |

# Animals and other research organisms

Policy information about studies involving animals; ARRIVE guidelines recommended for reporting animal research, and Sex and Gender in Research

| | |
|---|---|
| Laboratory animals | C57BL/6, BALB/c, OT-II TCR-Tg, B6.SJL, SwHEL BCR-Tg, B1.8i BCR-Tg, Cd4Cre/+, Rosa26ERT2Cre/+, Cxcr4flox/flox and Cxcr5flox/flox mice were bred and maintained at the Babraham Institute Biological Support Unit. Cxcr5flox/flox;Cd4ERT2Cre/+ mice were bred and maintained at the Core Facility Animal Models of the Biomedical Centre of LMU Munich. Mice were housed under pathogen-free conditions and were kept at an ambient temperature of ~19-21°C with 52% relative humidity. Once weaned, mice were kept in individually ventilated cages with 1-5 mice per cage and were fed CRM (P) VP diet (Special Diet Services) ad libitum. |
| Wild animals | The study did not involve wild animals |
| Reporting on sex | Experiments with aged C57BL/6 mice were conducted using males due to the limited availability of aged female mice. Experiments with aged BALB/c mice were conducted using females due to the limited availability of aged male mice. All other experiments using only adult mice were performed with both male and female mice. |
| Field-collected samples | The study did not involve samples collected from the field |
| Ethics oversight | All mouse experimentation was approved by the Babraham Institute Animal Welfare and Ethical Review Body. Animal husbandry and experimentation complied with European Union and United Kingdom Home Office legislation and local standards (PPL: P4D4AF812). |

Experiments with Cxcr5flox/flox;Cd4ERT2Cre/+ animals were performed in accordance with European regulation and federal law of Germany, and approved by the Regierung von Oberbayern.

Note that full information on the approval of the study protocol must also be provided in the manuscript.

# Flow Cytometry

## Plots

Confirm that:

☒ The axis labels state the marker and fluorochrome used (e.g. CD4-FITC).

☒ The axis scales are clearly visible. Include numbers along axes only for bottom left plot of group (a 'group' is an analysis of identical markers).

☒ All plots are contour plots with outliers or pseudocolor plots.

☒ A numerical value for number of cells or percentage (with statistics) is provided.

## Methodology

| | |
|---|---|
| Sample preparation | For lymphocyte staining, single cell suspensions from inguinal LNs were prepared by mechanical disruption of the tissues through a 70μm mesh in 2% FBS in PBS. The cell number and viability of samples were acquired using a CASY TT Cell Counter (Roche). 5x106 cells were transferred and stained in 5ml FACS tubes or 96- well plates. Cells were stained with Live/Dead Fixable Blue Dead Cell Stain (#L23105 Invitrogen) diluted at 1:1000 in PBS and incubated for 10 min at 4°C. Surface antibody stains were performed for 30 min-2 hours at 4°C in Brilliant stain buffer (#563794 BD Biosciences) after which cells were washed with 2% FBS in PBS and fixed using the Foxp3/Transcription Factor Staining Buffer Set (#00-5323-00 eBioscience). The antibodies used for flow cytometry of primary mouse cells are listed in Table 1. For intracellular staining, cells were incubated for 1 hour at 4°C with anti-Foxp3, anti-Ki67 and anti-Bcl6 antibodies diluted in 1x Permeabilization buffer (#00-8333-56 eBioscience).<br>For stromal cell staining, single cell suspensions from inguinal LNs were prepared by enzymatic digestion with 0.2mg/ml Collagenase P (#11213865001 Sigma), 0.8mg/ml Dispase II (#4942078001 Sigma) and 0.1mg/ml DNase I (#10104159001 Sigma) in plain RPMI medium (#11875093 Gibco). The LN capsules were penetrated with fine needles and incubated in the digestion buffer at 37°C for 15min and tubes were inverted every 5min. LNs were then triturated with a 1ml pipetted tip and the supernatant containing the released cells was collected in ice-cold 2%FBS in PBS with 2mM EDTA. The remaining fragments were processed in two more rounds as described, triturating every 5min. The cell suspensions were then filtered through a 100μm mesh and stained with the appropriate antibody cocktail. |
| Instrument | Samples were acquired on either an LSR Fortessa (BD Biosciences) with stained UltraComp eBeads Compensation Beads (#01-2222-41 Invitrogen) as single colour compensation controls or on a Cytek Aurora Spectral Cytometer (Cytek) with stained cells as single colour compensation controls. |
| Software | Flow cytometry data was collected using the BD FACSDiva Software v9.0 for samples acquired on an LSR Fortessa (BD Biosciences). For samples acquired on a Cytek Aurora Spectral Cytometer (Cytek), data was collected using the SpectroFlo v3.0 Software. All flow cytometry data was analysed using FlowJo v10.0 software. |
| Cell population abundance | Lymphocytes that were quantified by flow cytometry from a single draining inguinal lymph node yielded between 3-6x10^3 Tfh cells, 0.5-2x10^3 CXCR4+ Tfh cells, 0.5-2x10^3 Tfr cells, 2-6x10^4 GC B cells in adult wild type mice at the peak of the germinal centre response, D10-14. For adoptive transfers of B1.8i cmyc-GFP cells into wild type mice, 0.5-1.5x10^3 c-myc+ GC B cells were recovered from two draining inguinal lymph nodes. For stromal cell stains, roughly 0.1-1.5x10^3 FDCs were identified from two draining inguinal lymph nodes.<br><br>For VH186.2 sequencing, 48 NP+IgG1+ GC B cells were single-sorted per mouse into 96 well plates. From a single lymph node a total of roughly 0.5-3x10^3 NP+IgG1+ GC B cells could be recovered. After nested PCR for VH186.2, DNA from the sorted cells was run on an agarose gel to assess purity. Roughly >80% of sorted samples resulted in a PCR product of the right size. |
| Gating strategy | Lymphocytes were gated based on their size by FSC-A/SSC-A after which Single Cells were gated based on either SSC-W/SSC-A or FSC-H/FSC-A. Live cells from experiments including live/dead stains were then gated based on being negative for the used live/dead stain against FSC-A. Lymphocytes were then separated into either CD4+B220- T cells or B220+CD4- B cells. CD4+B220- T cells were then further gated into either Foxp3+CD4+ T regulatory cells or Foxp3-CD4+ T conventional cells. Follicular T helper cells were gated as PD1+CXCR5+ cells from the Foxp3-CD4+ T conventional cells parent gate while Follicular T regulatory cells were gated as PD1+CXCR5+ cells from the Foxp3+CD4+ T regulatory cell parent gate. CXCR4+CD4+ Follicular helper T cells were gated using a Fluorescence Minus One (FMO) control. For experiments using mouse models of T cell-specific CXCR5 deletion, T follicular helper (Tfh) cells were also gated as PD1+Bcl6+ from a Foxp3-CD4+ T cell parent gate. To identify Germinal Centre (GC) B cells, either Ki67+Bcl6+, Bcl6+CD38-, CD38-GL7+ or CD38-CD95+ were gated from B220 +CD4- B cells. For adoptive transfer experiments, live cells were gated based on their expression of the congenic markers CD45.1 and CD45.2 to separate out transferred cells from host cells, after which GC B cells and Tfh cells were identified as described above. GC B cells were then further divided into either CXCR4hiCD86lo centroblasts, CD86hiCXCR4lo centrocytes and cmyc-GFP+ GC B cells.<br>For Stromal cells gating, cells were first gated based on their size by FSC-A/SSC-A after which Single Cells were gated based on FSC-H/FSC-A. Live cells were then gated based on being negative for the live/dead eF780 stain against FSC-A. From the live cells, mesenchyme stroma was then gated as gp38+CD45- to exclude lymphocytes. This population was then further gated into ICAM+CD31- after which Follicular Dendritic Cells were gated as either EpCAM+CD21/35+ or MadCAM+CD21/35+. |

☒ Tick this box to confirm that a figure exemplifying the gating strategy is provided in the Supplementary Information.

