## [Peer Review File · Nature Immunology]

Peer Review Information

Journal: Nature Immunology

Manuscript Title: Spatial dysregulation of T follicular helper cells impairs vaccine responses in ageing

Corresponding author name(s): Dr. Michelle Linterman

Reviewer Comments & Decisions:

Decision Letter, initial version:
--

10th Jul 2022

Dear Michelle,

Thank you for providing a point-by-point response to the referees' concerns on your manuscript entitled "Spatial dysregulation of T follicular helper cells impairs vaccine responses in ageing". As noted previously, while they find your work of considerable potential interest, they have raised a number of concerns that must be addressed. In light of these comments, we cannot accept the current manuscript for publication, but would be very interested in considering a revised version along the lines proposed in your rebuttal.

We invite you to submit a substantially revised manuscript, however please bear in mind that we will be reluctant to approach the referees again in the absence of major revisions.

Specifically, the revision should include new experiments to address:

- (1) Examine age-related differences in CXCR4 transcription/post-translational modification and surface expression in mouse Tfh cells
- (2) Immunize heterozygous $Cxcr4^{fl/+}$ Cd4-Cre mice and determine GC and antibody responses; same for the WHIM-syndrome (mutant CXCR4) mice
- (3) Address direct vs indirect Tfh-FDC interactions by in vitro co-culture and measurement of multiple GC parameters; likewise co-culture Tfh-FDC-B cells
- (4) Examine conditional FDC-ablation of MHCII expression in Hprt-targeted stromal cells in male mice and whether this ablation influences FDC expansion/GC generation
- (5) Quantify the FDCs in young and aged mice
- (6) Examine CXCR4/CXCR5 expression in human cTfh cells from young adult vs aged individuals
- (7) Overexpress CXCR4 in young transgenic Tfh cells and adoptively transfer into aged recipients to determine if these cells rescue the GC defects

Please include the additional textual clarifications as indicated in your response letter.

When you revise your manuscript, please take into account all reviewer and editor comments, please highlight all changes in the manuscript text file in Microsoft Word format.

* If you have not done so already please begin to revise your manuscript so that it conforms to our Article format instructions at <http://www.nature.com/ni/authors/index.html>. Refer also to any guidelines provided in this letter.

The Reporting Summary can be found here:

When submitting the revised version of your manuscript, please pay close attention to our [href="https://www.nature.com/nature-portfolio/editorial-policies/image-integrity">Digital Image Integrity Guidelines. and to the following points below:](https://www.nature.com/nature-portfolio/editorial-policies/image-integrity)

[REDACTED]

Note: This URL links to your confidential home page and associated information about manuscripts you may have submitted, or that you are reviewing for us. If you wish to forward

this email to co-authors, please delete the link to your homepage.

If you wish to submit a suitably revised manuscript we would hope to receive it within 6 months. If you cannot send it within this time, please let us know. We will be happy to consider your revision so long as nothing similar has been accepted for publication at Nature Immunology or published elsewhere.

Nature Immunology is committed to improving transparency in authorship. As part of our efforts in this direction, we are now requesting that all authors identified as 'corresponding author' on published papers create and link their Open Researcher and Contributor Identifier (ORCID) with their account on the Manuscript Tracking System (MTS), prior to acceptance. ORCID helps the scientific community achieve unambiguous attribution of all scholarly contributions. You can create and link your ORCID from the home page of the MTS by clicking on 'Modify my Springer Nature account'. For more information please visit www.springernature.com/orcid.

Thank you for the opportunity to review your work.

Kind regards,

Laurie

Laurie A. Dempsey, Ph.D.
Senior Editor
Nature Immunology
l.dempsey@us.nature.com
ORCID: 0000-0002-3304-796X

Referee expertise:

Referee #1: Aging in the immune system

Referee #2: Lymphoid tissues & stromal cell environment

Referee #3: Aging in the immune system

Reviewers' Comments:

Reviewer #1:

Remarks to the Author:

The manuscript "Spatial dysregulation of T follicular helper cells impairs vaccine responses in ageing" submitted by Silva-Cayetano and colleagues examines the factors that might account for reduced humoral responses in older individuals. These reduced humoral responses are clinically significant

since they result in less robust responses (lower antibody titers and reduced affinity maturation) to vaccinations in older adults, leaving this vulnerable population less protected when compared to younger adults. The main finding of this study is that there are age-related changes in the spatial organization of germinal centers (GC) following vaccination, including CXCR4-mediated mislocalization of T follicular helper cells (Tfh) in the dark zone.

Overall, this is a very nice study but there are some issues that take away from the overall significance.

1. The studies presented in figures 1 and 2 are largely repetitive of what is already known in this field.
2. The genetic manipulation of CXCR4 expression provides interesting insights, but it is not clear that just because the phenotype that is observed turns out to be similar to what is seen with aging that this is the actual defect impairing humoral responses in aged individuals.
3. The study does not address why CXCR4 expression is dysregulated on Tfh in aged mice. The authors state in the discussion that their results suggest "T cell-intrinsic alterations in CXCR4 internalisation and/or degradation are likely responsible for its increased surface expression in ageing".

Reviewer #2:

Remarks to the Author:

In this study, Silva-Cayetano et al., investigate the role of follicular helper T cells (Tfh) localization in orchestrating the age-associated decline in the magnitude and quality of germinal center (GC) responses. The functional implications of Tfh retention in and exclusion from the GC light zone (LZ) on the magnitude and affinity of GC B cell responses as well as FDC activation are assessed by the complete genetic ablation of *Cxcr4* or *Cxcr5* expression on CD4 T cells, respectively. The observed defects in GC size and GC B cell number resulting from LZ exclusion of *Cxcr5*-deficient CD4 T cells extends the earlier work by Cyster and colleagues (Haynes N.M., et al, 2007), now describing the reduction in FDC network size. The major advances presented in the manuscript are the suggestions that: i) increased CXCR4 surface expression alters the proportion of T cells in the DZ, causing age-related defects in GC responses, and ii) Tfh provide help to FDCs in addition to B cells. Understanding the mechanisms driving age-associated defects in humoral immunity is highly relevant, however the reviewer's enthusiasm is somewhat curbed by the potentially contradictory findings in this study compared to the group's recently published data on the topic (Denton A., et al., 2022. *Sci Immunol.*), and the remaining mechanistic uncertainty as to whether the observed defects in FDC activation are due to the reduced interaction with Tfh or are a consequence of the reduced magnitude of GC responses due to abrogated B cell selection and affinity maturation.

Major comments to consider:

1. Using similar vaccination models, Linterman and colleagues recently demonstrated the age-associated defects in the magnitude and quality of GC B cell responses and stromal cell activation (including FDCs) in aged compared to adult mice (Denton et al., 2022, *Sci. Immunol.*). Using heterochronic parabiosis experiments, the authors demonstrated that circulating immune cells from adult mice are insufficient to restore defects in humoral immunity and the activation of MAdCAM1+ stromal cells and FDCs in aged parabionts. This work is not sufficiently referenced in the current

manuscript, especially given the discrepant observations in terms of the ability of adult T cells to revert age-associated defects in GC responses and FDC activation (Figure 7). This discrepancy should be experimentally addressed and appropriately discussed. To this end, the authors could use a similar approach as depicted in Figure 1 j-m to rule out a B cell-intrinsic defect in the age-associated reduction of GC responses. Transferring either young or old T cells into aged recipients, would show that age-associated, T cell-intrinsic differences in CXCR4+ Tfh (Figure 4a-c) drive altered Tfh positioning and reduced FDC activation and GC responses.

2. A major claim of the study is that increased CXCR4 expression on aged Tfh causes a higher proportion of T cells to localize to the dark zone, thereby accounting for age-associated defects in GC responses. While T cell-targeted *Cxcr4*-deletion is shown to retain T cells in the GC DZ (Figure 4), this key observation should be strengthened to more closely mirror the setting in aged mice and humans by assessing the effects of CXCR4 expression level/ frequency, rather than complete gene deficiency, on GC output and structure. Indeed, CD4-Cre-targeted *Cxcr4*-deficiency places an unphysiological proportion of T cells in the GC LZ (Figure 5d,e), while *Cxcr5*-deficiency limits Tfh entry into the GC (Haynes et al., 2007). To this end, the authors could assess GC responses in heterozygous *Cxcr4*^{fl/-} CD4-Cre mice, or following the transfer of CXCR4-gain of function T cells generated by the studies' co-authors.

3. Along similar lines as the comment above, complete genetic abrogation of CXCR5 or CXCR4 profoundly impact the ability of Tfh to provide help to B cells. A key conceptual advance presented in the manuscript is that Tfh provide help to FDC in addition to B cells. As Tfh localization in the LZ is intimately linked with B cell selection, affinity maturation and the magnitude of the GC responses, it remains unclear whether direct Tfh - FDC interactions drive FDC expansion, or whether the extent of FDC expansion is driven by the magnitude of GC responses. Experimentally untangling these two interactions without either blocking T cell entry into the GC (*Cxcr5*-deficiency) or accumulation in the LZ (*Cxcr4*-deficiency) is needed to substantiate this major claim.

4. The authors should assess whether the number and size of the steady state CD21/35+ FDC network is altered between adult and aged mice prior to vaccination, as this would help to demonstrate that the effect on FDC expansion is induced along with GC and Tfh formation.

Minor comments:

1. The authors should clarify whether there are strain-related differences in the proportion of the GC occupied by the LZ and DZ and FDC yield. In Figure 3, the GC LZ:DZ is roughly 60:40% in adult BALB/c mice and 40:60% in aged BALB/c mice – a 20% shift in relative proportions associated with aging. Genetic perturbation experiments performed for Figures 4-7 employ C57Bl/6 mice, with data in Figures 5-6 collected from adult mice showing a range in the proportion of LZ:DZ area in WT-equivalent mice of 50:50% in Figure 4 and 70:30% in Figure 5. The authors should provide a quantification of the LZ:DZ area in adult and aged C57Bl/6 mice that serves as a comparative baseline for genetic perturbation studies that dissect the requirement for CXCR4 and CXCR5 expression on FDC activation, and substantiate genotype-related differences in LZ:DZ GC organization between chemokine receptor-competent mice.

2. The GC area appears to be reduced following the transfer of *Cxcr4*^{fl/fl} Rosa26 ERT2Cre/+ OT-II T cells into adult mice in Figures 4g (and Extended Figure 7j), but not in *Cxcr4*^{fl/fl}; CD4Cre/+ mice. This

should be quantified and any differences in GC size/output according to Cxcr4-deletion on transferred OT-II T cells as compared to endogenous Cxcr4-deficient T cells should be mentioned in the manuscript.

3. As histological quantification is used throughout the study, the side-by-side single channel images should be provided for:

i) T cells in Figure 4g, Figure 5a, Figure 6a, Figure 7e,l, and corresponding Extended Figures as done for Figure 3a.

ii) Stromal cell CXCL12 staining (Figure 4e, along with secondary Ab control)

4. The current selection of representative images used for quantification of GC T cell localization in Extended Figure 11a that lack a round, discriminate shape, making it difficult to see the LZ:DZ distribution of cells.

5. The recovery of MAdCAM1+ CD21/35+ cells in chemokine receptor-competent (WT-equivalent) mice in Figure 5 and 6 is substantially distinct. Please comment on or address the difference in FDC yield between experiments.

6. Extended data figures should appear in the order referred to in the text. For example, Extended Figure 11a-g, corresponding to Figure 7, are mentioned in relation to Figure 6, before Extended Figures 9 and 10 are referenced (manuscript page 15).

Reviewer #3:

Remarks to the Author:

Silva-Cayetano et. al. demonstrates that TFH cell localization within germinal centers is not only important for high affinity B cell responses via canonical B cell helper function but also for the expansion of FDC networks in GC light zones. Moreover, they suggest that this localization is defective in aging due to CXCR4-mediated mechanisms. These novel findings provide new insight into GC biology and have the potential for identifying targets for amplifying vaccine responses in older individuals. However, there are some gaps in these studies that need to be addressed in order to fully substantiate the authors' conclusions.

Main Comments:

The adoptive transfer experiments in Figure 7 nicely demonstrate that the blunted GC response in older mice can be rescued by CXCR5 expressing CD4 T cells. These data could imply at least two different things; (1) aged T cells don't express CXCR5 as well as young or (2) differences in CXCR4 in aged T cells prevents them from functioning properly. As CXCR4 over-expression by aged TFH cells is defined as a critical mediator of blunted GC responses, it would be of value to demonstrate that CXCR4 OE in young OT-II cells similarly fail to increase high affinity GC responses and FDC network size in an old mouse.

Authors state multiple times without the text that there is no difference in CXCL12 differences within LN across age. However, CXCL12 expression in LN staining is not obviously apparent from the images in Figure 4e. Indeed, it looks like there might be more expression within dark zone in older mice. Authors should better quantify these images, as well as provide single color images for to make it

convincing that there is no CXCL12 difference in LN with age.

In Extended Fig 6e, authors attempt to translate their findings on CXCR4 into human T cell aging, however they solely look at CXCR4 expression on memory CD4 populations. However, it is unclear if these findings translate into the human TFH population. Does CXCR4 expression within the peripheral TFH population change across age? These data would be more convincing that human TFH display similar age-related differences as mice.

Other Comments:

Authors conclude that CXCR4 expression regulate migration into the dark zone and size of FDC networks. Although there is an obvious preferential bias of CXCR4 KO OT-II in the light zone, in these experiments it also appears that there is an overall decrease of in CXCR4 KO T cells in the LN overall. (Figure 4g-I) Authors should explain these findings and what role CXCR4 might play in TFH development and migration overall.

The reduced cmyc-GFP+ GC B cells in older mice appears bi-modal in Figure 2d thus any conclusions about these data are unclear. Authors need to confirm if these results were due to technical error or underlying heterogeneity of aging.

CXCR5 expression is induced by TCR cell activation, which is blunted in human T cells with age. Authors should discuss this in light of their findings. Is it possible that T cells just have reduced CXCR5 with age (independent of CXCR4 expression), and thus are less effective at getting into the GC and helping FDCs out? (see main comment above)

Author Rebuttal to Initial comments
--

See Inserted PDF

We thank the reviewers for their thorough and supportive review of our manuscript, we have extensively revised this manuscript in accordance with their suggestions. All changes and additions are outlined below in purple. Text that is included in the revised manuscript is in blue.

Reviewer #1

The manuscript “Spatial dysregulation of T follicular helper cells impairs vaccine responses in ageing” submitted by Silva-Cayetano and colleagues examines the factors that might account for reduced humoral responses in older individuals. These reduced humoral responses are clinically significant since they result in less robust responses (lower antibody titers and reduced affinity maturation) to vaccinations in older adults, leaving this vulnerable population less protected when compared to younger adults. The main finding of this study is that there are age-related changes in the spatial organization of germinal centers (GC) following vaccination, including CXCR4-mediated mislocalization of T follicular helper cells (Tfh) in the dark zone. Overall, this is a very nice study but there are some issues that take away from the overall significance.

1. The studies presented in figures 1 and 2 are largely repetitive of what is already known in this field.

We have consolidated original figures 1 & 2 into one, and moved the rest of the data to the extended figures. Figure 1 now prioritises the data that makes a novel contribution to the field: The quantitative imaging of GCs (Figure 1a-c), adoptive transfers showing that B cell intrinsic changes with age are not responsible for the diminished GC response (Figure 1d-g) and the impaired positive selection of “young” B cells in an aged microenvironment (Figure 1i, j). In addition, we have chosen to retain the affinity maturation data in the main figure, to provide a baseline for the subsequent figures.

Figure 1

Figure 1 | The GC response, and its output is diminished in aged mice.

(a) Representative confocal images of iLNs at 10x magnification and GCs at 40x from adult and aged BALB/c mice 14 days after immunisation with NP-KLH in alum. Scale bars are 200µm on LN images and 50µm on GC images. LN sections were stained for IgD (green), CD35 (white), Ki67 (blue) and CD3 (magenta). (b) Enumeration of GCs per LN was performed by examining 6-10 sections throughout each LN and identifying GCs as CD35⁺Ki67⁺IgD⁻ structures (c) Quantification of the total GC area was performed using an automated Cell Profiler pipeline for which 5-6 sections were analysed per GC. (d) Experimental outline of the co-transfer of SW_{HEL} B cells from either adult or aged donors alongside OT-II T cells from adult donors into adult C57BL/6 recipient mice in which GC formation was analysed 10 days after immunisation with HEL-OVA in alum. (e) Representative flow cytometry plots identifying SW_{HEL} derived GC B cells (CD95⁺CD38⁻CD45.1⁺B220⁺HEL⁺) in recipient mice. The values next to the gates on flow cytometry plots indicate the population percentage. Quantification of the frequency (f) and total number (g) of SW_{HEL} derived GC B cells. (h) Pie charts indicating the frequency of the affinity-inducing mutation W33L in the CDR1 region of V_H186.2 sequenced from single cell sorted NP⁺IgG1⁺ GC B cells of adult and aged C57BL/6 mice 21 days post-immunisation with 1W1K-NP/Alum. The values in the centre of the pie charts indicate the total number of cells sequenced per group (n=8-10). (i) Experimental outline of the transfer of B1.8ⁱ c-myc^{GFP/GFP} B cells from adult donors into adult or aged C57BL/6 recipient mice 10 days after immunisation with NP-OVA in alum. Quantification of the frequency (j) of B1.8i derived c-myc⁺ GC B cells in adult or aged mice, data are pooled from two independent experiments, first experiment in black, second experiment in white. (k-m) Representative ELISpot well images (left) and quantification (right) of bone marrow NP-23 (k) and NP-2 (l) specific IgG1 antibody secreting cells (ASCs) in BALB/c mice 21 days after immunisation with NP-KLH in alum. (m) Affinity maturation of bone marrow ASCs from BALB/c mice as determined by the ratio of NP2/NP23 specific ASCs. For all experiments 2-4 experimental repeats were performed with 5-10 mice per group. In bar graphs, each symbol represents a mouse and the bar height represents the median. The p-values were generated by performing an unpaired Mann Whitney U test.

2. The genetic manipulation of CXCR4 expression provides interesting insights, but it is not clear that just because the phenotype that is observed turns out to be similar to what is seen with aging that this is the actual defect impairing humoral responses in aged individuals.

Both ageing and the germinal centre reaction are complex biological processes to which multiple cell types contribute. The evidence presented in this paper supports that the mislocalisation of Tfh cells in ageing contributes to the defective function of the germinal centre. Genetic manipulation of CXCR4 on T cells demonstrates, for the first time, that this receptor is required for the dark zone localisation of a proportion of GC-Tfh cells. In the context of ageing, we show that surface CXCR4 expression is increased on Tfh cells in aged mice, and on circulating Tfh-like cells and that this results in enhanced chemotaxis to CXCL12 (Figure 3). We show that supply of T cells that localise correctly to the light zone (CXCR5-sufficient), but not those that localise to the dark zone (CXCR5-deficient) are sufficient to correct the GC reaction at its peak in aged mice. When all these data are taken together it supports that mislocalisation of Tfh cells in ageing is responsible for the defective size of the GC reaction at its peak, and poor expansion of the FDC network into the established GC. In order to directly address the point about whether correcting the GC with LZ localising T cell enhances humoral immunity in aged mice, we repeated of the OT-II transfer into aged mice but waiting until 35days post immunisation to look at GC-derived antibody secreting cells (Figure 6). We found that the frequency of high affinity antibody secreting cells was enhanced by the addition of OT-II T cells seven weeks after immunisation, demonstrating that correctly localised T cells and enhancing expansion of the FDC network does boost humoral immunity in ageing. These data have been included in Figure 6i, and are called out on page 17: “To test whether correction of these age-dependent defects in the GC impacted humoral immunity in ageing, we assessed the ability of OT-II cells to support the formation of high-affinity plasma cells. Indeed, the number of NP2-binding GC-derived plasma cells were increased in aged mice that received OT-II cells, resulting in enhanced affinity maturation (Figure 6i). Together, these data support that supply of

LZ-localising T cells to aged mice can expand the FDC network, increase GC size at the peak of the response and enhance GC-derived humoral immunity."

3. The study does not address why CXCR4 express is dysregulated on Tfh in aged mice. The authors state in the discussion that their results suggest "T cell-intrinsic alterations in CXCR4 internalisation and/or degradation are likely responsible for its increased surface expression in ageing".

Cell surface expression of CXCR4 is downregulated after interaction with its ligand CXCL12. To test whether this process is impaired in ageing, we examined the downregulation of CXCR4 on Tfh cells that have migrated in response to CXCL12 in chemotaxis assays. Tfh cells from younger mice downregulate surface CXCR4 at much lower concentrations of CXCL12 than Tfh cells from aged mice. Although this can be overcome with high concentrations of CXCL12. This suggests that ageing dulls the ligand dependent downregulation of CXCR4, resulting in enhanced cell surface expression. These new data are presented in Figure 3g, and the results (page 9) and discussion (page 21) text have been updated to reflect this new information.

Results, page 9: "In the chemotaxis assays, Tfh cells from younger mice downregulated CXCR4 after migrating to CXCL12, but this did not occur to the same extent in Tfh cells from aged mice (Figure 3g), supporting impaired ligand-dependent internalisation as the cause of enhanced CXCR4 surface expression in ageing."

Discussion, page 21: "After binding CXCL12, CXCR4 is internalised and sorted through the endosomal compartment, where it is either recycled back to the plasma membrane or is degraded. Our data show that ligand-dependent CXCR4 internalisation is defective on Tfh cells from aged mice, indicating that CXCR4 proteostasis is impaired in ageing. This process is regulated by ubiquitination of the receptor at three lysines in the C-terminal region of CXCR4. Mono-ubiquitination of CXCR4 is impaired in CD4⁺ T cells from the blood of older people, as this modification is known to target the receptor for degradation, this is the likely cause the increased expression of CXCR4 on Tfh cells in ageing^{59,60}."

Figure 3 | (g) Median fluorescence intensity (MFI) of cell surface CXCR4 expression on PD1⁺CXCR5⁺Foxp3⁻ Tfh cells that migrated to the indicated concentrations of CXCL12 (n=8). Data is representative of two independent experiments. P-values from 2-way ANOVA with Sidak's multiple comparisons test.

Reviewer #2

In this study, Silva-Cayetano et al., investigate the role of follicular helper T cells (Tfh) localization in orchestrating the age-associated decline in the magnitude and quality of germinal center (GC) responses. The functional implications of Tfh retention in and exclusion from the GC light zone (LZ) on the magnitude and affinity of GC B cell responses as well as FDC activation are assessed by the complete genetic ablation of Cxcr4 or Cxcr5 expression on CD4 T cells, respectively. The observed defects in GC size and GC B cell number resulting from LZ exclusion of Cxcr5-deficient CD4 T cells extends the earlier work by Cyster and colleagues (Haynes N.M., et al, 2007), now describing the reduction in FDC network size. The major advances presented in the manuscript are the suggestions that: i) increased CXCR4 surface expression alters the proportion of T cells in the DZ, causing age-related defects in GC responses, and ii) Tfh provide help to FDCs in addition to B cells.

Understanding the mechanisms driving age-associated defects in humoral immunity is highly relevant, however the reviewer's enthusiasm is somewhat curbed by the potentially contradictory findings in this study compared to the group's recently published data on the topic (Denton A., et al., 2022. *Sci Immunol.*), and the remaining mechanistic uncertainty as to whether the observed defects in FDC activation are due to the reduced interaction with Tfh or are a consequence of the reduced magnitude of GC responses due to abrogated B cell selection and affinity maturation.

Major comments to consider:

1. Using similar vaccination models, Linterman and colleagues recently demonstrated the age-associated defects in the magnitude and quality of GC B cell responses and stromal cell activation (including FDCs) in aged compared to adult mice (Denton et al., 2022, *Sci. Immunol.*). Using heterochronic parabiosis experiments, the authors demonstrated that circulating immune cells from adult mice are insufficient to restore defects in humoral immunity and the activation of MAdCAM1+ stromal cells and FDCs in aged parabionts. This work is not sufficiently referenced in the current manuscript, especially given the discrepant observations in terms of the ability of adult T cells to revert age-associated defects in GC responses and FDC activation (Figure 7). This discrepancy should be experimentally addressed and appropriately discussed.

We appreciate the reviewer highlighting our 2022 *Science Immunology* paper, and apologise for not making the significant advance of this manuscript clear. In brief, Denton *et al.*, 2022 focussed on GC formation (day 7 post immunisation), rather than the peak of the response. The parabionts were analysed seven days after immunisation, whereas the data reported in Silva-Cayetano *et al.*, focusses on days 10-14, the peak of the response. Notably, we show that the transfer of OT-II T cells can boost both the defective FDC and the GC size at day 10 post immunisation (Figure 6, Silva-Cayetano *et al.*), but not at day seven post immunisation (new data included in Extended Figure 18). These data provide the requested experimental evidence that the parabiosis and OT-II transfer data yield similar results of a diminished response seven days after immunisation. This suggests that the location of Tfh cells is important once the GC is established, which is consistent with the light and dark zones forming only after the GC has been populated with clonally expanding GC B cells. We have updated the introduction and discussion to make this point clear:

Introduction, page 2: "We and others have previously identified defects in conventional dendritic cells and lymph node stromal cells as causing delayed GC formation in ageing, but correction of these cellular defects using TLR agonists as an adjuvant was not sufficient to enhance the size of the GC response at its peak, nor the production of high affinity antibody secreting plasma cells^{1,27-29}. Therefore, understanding why the size of the GC, and its output, is impaired in ageing is key to determining why older bodies are less capable of mounting persistent antibody responses to vaccines. "

Discussion, page 22: "Ageing is a multifaceted process, and the mechanisms by which it alters the GC reaction are complex due to the number of processes required to coordinate key cellular interactions across time and

space for a successful response^{66,67}. There are three clear defects in the GC reaction with age: 1) Its formation is delayed, 2) its size is smaller at its peak, and 3) fewer high affinity plasma cells are produced. We and others have previously shown that the first defect (1), namely the delayed formation of the GC, is due to the age of the lymph node microenvironment, specifically caused by a diminished response in both conventional dendritic cells and MAdCAM-1-expressing lymph node stromal cells. However, correction of these impairments through the use of TLR7 and TLR4 agonists could not restore the diminished size of the GC at its peak (defect 2) nor impaired production of high affinity antibody secreting cells (defect 3) in ageing^{1,27-29}. Here, we show that the transfer of T cells can boost both GC size at the peak of the response (day 10 post immunisation), and the number of high affinity bone marrow plasma cells, thereby correcting these latter two defects. It is important to highlight, however, that there was no significant effect on enhancing the GC at day seven post immunisation simply by giving T cells alone (Figure 6 and Extended Figure 18). Together, these studies show that the delay in GC formation is caused by non-migratory cells in the lymph node that cannot be corrected by young T cells, while the defective GC size at its peak and output of high affinity cells is driven by CXCR4-dependent mislocalisation of Tfh cells. This indicates that effective strategies for enhancing vaccine responses in older people will need to concomitantly address age-dependent changes in both the microenvironment, and in Tfh cells.

Importantly, in Denton *et al.*, whilst we were able to rescue the defective marginal reticular cells with a TLR4-stimulating adjuvant, we were not able to correct the defective FDC network expansion, nor the formation of germinal centre derived antibody secreting cells in aged mice (Figure 7, Denton *et al.*, *Sci Immunol* 2022). As researchers, this was disappointing as we have always hoped that we would use our discoveries to correct defects in humoral immunity, which is why we are particularly excited about the data presented here in Silva-Cayetano *et al.*, including a new experiment (Figure 6i) performed to further differentiate these two manuscripts. Knowing we were able to correct the defective FDC network by supplying T cells that correctly localise to the light zone we wanted to test whether this enhanced GC-derived antibody secreting cells. To this end, we repeated the OT-II transfer experiments into aged mice (Figure 6). In two independent experiments, the frequency of high affinity antibody secreting cells was enhanced by the addition of OT-II T cells seven weeks after immunisation, unlike the use of a TLR4-stimulating adjuvant in Denton *et al.* These data are a clear advance in our understanding of the causes of poor humoral immunity in ageing. These new data have been included in Figure 6i: and are called out on page 17: “To test whether correction of these age-dependent defects in the GC impacted humoral immunity in ageing, we assessed the ability of OT-II cells to support the formation of high-affinity plasma cells. Indeed, the number of NP2-binding GC-derived plasma cells were increased in aged mice that received OT-II cells, resulting in enhanced affinity maturation (Figure 6i). Together, these data support that supply of LZ-localising T cells to aged mice can expand the FDC network, increase GC size at the peak of the response and enhance GC-derived humoral immunity.”

To this end, the authors could use a similar approach as depicted in Figure 1 j-m to rule out a B cell-intrinsic defect in the age-associated reduction of GC responses.

The data presented in Figures 1d-g (condensed from prior Figures 1 and 2 in response to reviewer 1) directly address the contribution of B cell intrinsic age-dependent changes to the age-associated reduction in the GC response. We transferred SW_{HEL} B cells from either adult or aged into young mice, and showed that there is no B cell-intrinsic ageing defect in GC responses.

Figure 1| The GC response, and its output is diminished in aged mice.

(d) Experimental outline of the co-transfer of SW_{HEL} B cells from either adult or aged donors alongside OT-II T cells from adult donors into adult C57BL/6 recipient mice in which GC formation was analysed 10 days after immunisation with HEL-OVA in alum. **(e)** Representative flow cytometry plots identifying SW_{HEL} derived GC B cells (CD95⁺CD38⁻CD45.1⁺B220⁺HEL⁺) in recipient mice. The values next to the gates on flow cytometry plots

indicate the population percentage. Quantification of the frequency (**f**) and total number (**g**) of SW_{HEL} derived GC B cells.

Transferring either young or old T cells into aged recipients, would show that age-associated, T cell-intrinsic differences in CXCR4+ Tfh (Figure 4a-c) drive altered Tfh positioning and reduced FDC activation and GC responses.

We agree that transferring transgenic T cells from an aged mouse would be an interesting experiment, but it is not technically feasible. We have tried to age HEL-specific TCR7 mice specifically for these purposes but the mice develop lymphoma around 12 months of age before they are appropriately aged to use in for these experiments. Therefore, despite our wish to perform these experiments, we do not have aged TCR transgenic mice available. We had chosen to age TCR7 mice because a previous study reported a loss of OT-II cells in aged B6 OT-II mice due to expression of an endogenous superantigen in B6 mice (PMID: 18354168), and were concerned that using any remaining OT-II cells would be a serious experimental confounder.

2. A major claim of the study is that increased CXCR4 expression on aged Tfh causes a higher proportion of T cells to localize to the dark zone, thereby accounting for age-associated defects in GC responses. While T cell-targeted *Cxcr4*-deletion is shown to retain T cells in the GC DZ (Figure 4), this key observation should be strengthened to more closely mirror the setting in aged mice and humans by assessing the effects of CXCR4 expression level/ frequency, rather than complete gene deficiency, on GC output and structure. Indeed, CD4-Cre-targeted *Cxcr4*-deficiency places an unphysiological proportion of T cells in the GC LZ (Figure 5d,e), while *Cxcr5*-deficiency limits Tfh entry into the GC (Haynes et al., 2007). To this end, the authors could assess GC responses in heterozygous *Cxcr4*^{fl/-} CD4-Cre mice, or following the transfer of CXCR4-gain of function T cells generated by the studies' co-authors.

As suggested, we have performed immunisation studies in heterozygous *Cxcr4*^{fl/+}; *Cd4*^{cre/+} mice, and additionally in heterozygous *Cxcr5*^{fl/+}; *Cd4*^{cre/+} mice for completeness. Overall, these data support the original conclusions presented in the paper, and we thank the reviewer for suggesting these experiments, as we agree that they strengthen the manuscript.

The *Cxcr4*^{fl/+}; *Cd4*^{cre/+} experiment has been included as Extended figure 13. These data show the same trends as the homozygous deletion of CXCR4 on T cells. As expected, the effect size is smaller in the heterozygous than homozygous animals, but the original conclusions are supported by these data. These results are called out on page 13: "We also observed increased in high-affinity antibody production in heterozygous *Cxcr4*^{fl/+}; *Cd4*^{cre/+} mice, which have a mild skewing of Tfh cells to the LZ (Extended Figure 13). Although the effect size was not as large as in homozygous *Cxcr4*^{fl/fl}; *Cd4*^{cre/+} mice, likely due to heterozygous mice having only a ~40% reduction in expression of CXCR4 on the Tfh cell's surface (Extended Figure 13)."

Extended Figure 13 | T cell-specific heterozygosity of CXCR4, confocal imaging and flow cytometric analysis of immunised *Cxcr4^{fl/+}; Cd4^{Cre/+}* mice

(a) Representative flow cytometry plot indicating the expression of CXCR4 by PD1⁺CXCR5⁺Foxp3⁻ Tfh cells in iLNs of *Cxcr4^{fl/+}; Cd4^{+/+}* or *Cxcr4^{fl/+}; Cd4^{Cre/+}* mice 10 days after NP-OVA immunisation. (b) Median fluorescence intensity (MFI) of CXCR4 by PD1⁺CXCR5⁺Foxp3⁻ Tfh cells of *Cxcr4^{fl/+}; Cd4^{+/+}* or *Cxcr4^{fl/+}; Cd4^{Cre/+}* mice. The data are representative of two independent experiments. (c) Representative flow cytometry plots identifying Ki67⁺Bcl6⁺ GC B cells 10 days after NP-OVA immunisation of *Cxcr4^{fl/+}; Cd4^{+/+}* and *Cxcr4^{fl/+}; Cd4^{Cre/+}* mice; values adjacent to gates indicate percentages. Quantification of the percentage (d) and total number (e) of Ki67⁺Bcl6⁺ GC B cells at days 10, 21 and 35 after NP-OVA immunisation of *Cxcr4^{fl/+}; Cd4^{+/+}* and *Cxcr4^{fl/+}; Cd4^{Cre/+}* mice. (f) Representative flow cytometry plots identifying PD1⁺CXCR5⁺ Tfh cells 10 days after NP-OVA immunisation of *Cxcr4^{fl/+}; Cd4^{+/+}* and *Cxcr4^{fl/+}; Cd4^{Cre/+}* mice; values adjacent to gates indicate percentages. Quantification of the percentage (g) and total number (h) of PD1⁺CXCR5⁺ Tfh cells at days 10, 21 and 35 after NP-OVA immunisation of *Cxcr4^{fl/+}; Cd4^{+/+}* and *Cxcr4^{fl/+}; Cd4^{Cre/+}* mice. (i) Representative confocal images of GCs at day 10 after NP-OVA immunisation in the iLNs of control *Cxcr4^{fl/+}; Cd4^{+/+}* mice (left) and *Cxcr4^{fl/+}; Cd4^{Cre/+}* mice (right). Images were taken at 20x magnification and scale bar is 100µm. LN sections were stained for IgD (green), Ki67 (blue), CD35 (white) and CD3e (magenta). (j) Quantification of the total area of GCs identified by immunofluorescence as Ki67⁺CD35⁺IgD⁻ in the iLNs and of (k) the total number of CD3⁺ T cells within the Ki67⁺CD35⁺IgD⁻ GC area of *Cxcr4^{fl/+}; Cd4^{+/+}* and *Cxcr4^{fl/+}; Cd4^{Cre/+}* mice. (l) Percentage of CD3⁺ T cells localising to the CD35⁺ light zone area and (m) percentage of CD3⁺ T cells localising to the Ki67⁺CD35⁻ dark zone area of the GC in *Cxcr4^{fl/+}; Cd4^{+/+}* and *Cxcr4^{fl/+}; Cd4^{Cre/+}* mice. (n) Quantification of the CD35⁺ FDC network area representative of the light zone and of (g) the Ki67⁺CD35⁻ dark zone area within the GCs of *Cxcr4^{fl/+}; Cd4^{+/+}* and *Cxcr4^{fl/+}; Cd4^{Cre/+}* mice. For (i-o), quantification was performed using an automated Cell Profiler pipeline for which 5-6 sections were analysed per GC. Data are representative of two independent experiments. (p) Serum titres of NP-20 (left) and NP-2 (middle) specific IgG1 of *Cxcr4^{fl/+}; Cd4^{+/+}* and *Cxcr4^{fl/+}; Cd4^{Cre/+}* mice and antibody affinity maturation indicated by the NP-2/NP-20 antibody ratio (right) at 35 days post-immunisation with NP-OVA. Titres were normalised to a positive control and are displayed as arbitrary units. (q) Enumeration of NP-20 (left) and NP-2 (middle) IgG1 antibody secreting cells (ASCs) and affinity maturation indicated by the ratio of NP-2/NP-20 ASCs (right) in the bone marrow of *Cxcr4^{fl/+}; Cd4^{+/+}* and *Cxcr4^{fl/+}; Cd4^{Cre/+}* mice at 35 days post-immunisation with NP-OVA. Bar height on graphs indicates the median, each symbol represents a mouse (n=5-8), and p-values were obtained by performing an unpaired Mann-Whitney U test. Data are representative of two independent experiments.

The *Cxcr5^{fl/+}; Cd4^{Cre/+}* experiment has been included as Extended figure 17. Here, the data also follow similar trends to the homozygous T cell deficient mice, with two notable exceptions, the GC area is not diminished in these mice nor are the number of GC-Tfh cells. These data are called out on page 15: “To further evaluate how Tfh cell positioning towards the DZ influences the GC, we performed similar experiments, but with mice whose T cells have only one functional allele of CXCR5, *Cxcr5^{fl/+}; Cd4^{Cre/+}* mice. We hypothesised that Tfh cells from these mice would have less surface CXCR5 expression, and an intermediate phenotype between controls and full CXCR5 T cell knockouts. Ten days after immunisation, Tfh cells had a ~50% reduction in CXCR5 expression, which resulted in normal sized GCs, and comparable numbers of GC-Tfh cells to control *Cxcr5^{fl/+}; Cd4^{+/+}* mice, unlike mice that completely lack CXCR5 on their T cells. Nevertheless, heterozygosity of CXCR5 resulted in Tfh cell skewing to the DZ and, the proportion of the GC occupied by the FDC network was also diminished (Extended figure 17). This provides further evidence in support of Tfh cell localisation influencing the GC stroma.”

Extended Figure 17 | T cell-specific heterozygosity of CXCR5, confocal imaging and flow cytometric analysis of immunised *Cxcr5^{fl/+}; Cd4^{Cre/+}* mice

(a) Representative flow cytometry plot indicating the expression of CXCR5 by PD1⁺Bcl6⁺Foxp3⁻ Tfh cells in iLNs of *Cxcr5^{fl/+}; Cd4^{+/+}* or *Cxcr5^{fl/+}; Cd4^{Cre/+}* mice 10 days after NP-OVA immunisation. (b) Median fluorescence intensity (MFI) of CXCR4 by PD1⁺CXCR5⁺Foxp3⁻ Tfh cells of *Cxcr5^{fl/+}; Cd4^{+/+}* or *Cxcr5^{fl/+}; Cd4^{Cre/+}* mice. (c) Representative flow cytometry plots identifying Ki67⁺Bcl6⁺ GC B cells 10 days after NP-OVA immunisation of *Cxcr5^{fl/+}; Cd4^{+/+}* and *Cxcr5^{fl/+}; Cd4^{Cre/+}* mice; values adjacent to gates indicate percentages. Quantification of the percentage (d) and total number (e) of Ki67⁺Bcl6⁺ GC B cells at days 10, 21 and 35 after NP-OVA immunisation of *Cxcr5^{fl/+}; Cd4^{+/+}* and *Cxcr5^{fl/+}; Cd4^{Cre/+}* mice. (f) Representative flow cytometry plots identifying PD1⁺CXCR5⁺ Tfh cells 10 days after NP-OVA immunisation of *Cxcr5^{fl/+}; Cd4^{+/+}* and *Cxcr5^{fl/+}; Cd4^{Cre/+}* mice; values adjacent to gates indicate percentages. Quantification of the percentage (g) and total number (h) of PD1⁺CXCR5⁺ Tfh cells at days 10, 21 and 35 after NP-OVA immunisation of *Cxcr5^{fl/+}; Cd4^{+/+}* and *Cxcr5^{fl/+}; Cd4^{Cre/+}* mice. (i) Representative confocal images of GCs at day 10 after NP-OVA immunisation in the iLNs of control *Cxcr5^{fl/+}; Cd4^{+/+}* mice (left) and *Cxcr5^{fl/+}; Cd4^{Cre/+}* mice (right). Images were taken at 20x magnification and scale bar is 100µm. LN sections were stained for IgD (green), Ki67 (blue), CD35 (white) and CD3e (magenta). (j) Quantification of the total area of GCs identified by immunofluorescence as Ki67⁺CD35⁺IgD⁻ in the iLNs and of (k) the total number of CD3⁺ T cells within the Ki67⁺CD35⁺IgD⁻ GC area of *Cxcr5^{fl/+}; Cd4^{+/+}* and *Cxcr5^{fl/+}; Cd4^{Cre/+}* mice. (l) Percentage of CD3⁺ T cells localising to the CD35⁺ light zone area and (m) percentage of CD3⁺ T cells localising to the Ki67⁺CD35⁻ dark zone area of the GC in *Cxcr5^{fl/+}; Cd4^{+/+}* and *Cxcr5^{fl/+}; Cd4^{Cre/+}* mice. (n) Quantification of the CD35⁺ FDC network area representative of the light zone and of (g) the Ki67⁺CD35⁻ dark zone area within the GCs of *Cxcr5^{fl/+}; Cd4^{+/+}* and *Cxcr5^{fl/+}; Cd4^{Cre/+}* mice. For (i-o), quantification was performed using an automated Cell Profiler pipeline for which 5-6 sections were analysed per GC. Data are representative of two independent experiments. (p) Serum titres of NP-20 (left) and NP-2 (middle) specific IgG1 of *Cxcr5^{fl/+}; Cd4^{+/+}* and *Cxcr5^{fl/+}; Cd4^{Cre/+}* mice and antibody affinity maturation indicated by the NP-2/NP-20 antibody ratio (right) at 35 days post-immunisation with NP-OVA. Titres were normalised to a positive control and are displayed as arbitrary units. (q) Enumeration of NP-20 (left) and NP-2 (middle) IgG1 antibody secreting cells (ASCs) and affinity maturation indicated by the ratio of NP-2/NP-20 ASCs (right) in the bone marrow of *Cxcr5^{fl/+}; Cd4^{+/+}* and *Cxcr5^{fl/+}; Cd4^{Cre/+}* mice at 35 days post-immunisation with NP-OVA. Bar height on graphs indicates the median, each symbol represents a mouse (n=5-8), and p-values were obtained by performing an unpaired Mann-Whitney U test. Data are representative of two independent experiments.

3. Along similar lines as the comment above, complete genetic abrogation of CXCR5 or CXCR4 profoundly impact the ability of Tfh to provide help to B cells. A key conceptual advance presented in the manuscript is that Tfh provide help to FDC in addition to B cells. As Tfh localization in the LZ is intimately linked with B cell selection, affinity maturation and the magnitude of the GC responses, it remains unclear whether direct Tfh - FDC interactions drive FDC expansion, or whether the extent of FDC expansion is driven by the magnitude of GC responses. Experimentally untangling these two interactions without either blocking T cell entry into the GC (*Cxcr5*-deficiency) or accumulation in the LZ (*Cxcr4*-deficiency) is needed to substantiate this major claim.

We agree with the reviewer that the entangled nature of Tfh-GCB-FDC cell function makes it difficult to draw conclusions about the whether the impact on FDCs caused by the changes in location of Tfh cells is due to direct Tfh-FDC interactions, or via B cells. Our data show that any changes in the FDC network are not driven by changes in the magnitude of GC responses: In our experiments with the *Cxcr4^{fl/fl}; Cd4^{cre/+}* mice the size and magnitude of the GC response does not change (Figure 4). Therefore, the increase in FDC expansion in these mice cannot be explained by differences in GC size or an increased number of GC B cells, as none were observed. In order to ensure that we have accounted for GC size in our analyses, we have always expressed the FDC network as a proportion (%) of the GC throughout the manuscript to ensure that GC size is not skewing our interpretation. In addition, the data described in response to this reviewer's point #2 using heterozygous *Cxcr4^{fl/+}; Cd4^{cre/+}* mice and *Cxcr5^{fl/+}; Cd4^{cre/+}* mice that do not have different sized GCs to control animals, support the original conclusions, without complete loss of either CXCR4 or CXCR5.

To formally test whether the percentage of GC occupied by the FDC network is related to its size independent of changes in Tfh cell localisation, we have reanalysed our imaging of germinal centres in wild type 3-month-old BALB/c mice 14 days after immunisation. The plot on the left shows that when the FDC area is measured in actual size (mm²), there is a positive linear correlation with GC size (mm², left) as expected. However, when FDC area is measured as a proportion of the GC, there is no correlation with GC area (right). These data show that the proportion of the GC occupied by FDCs is not dependent on GC size.

We have added additional text to the results section (page 12) to clarify that the expanded FDC network in *Cxcr4^{fl/fl}; Cd4^{cre/+}* mice is not being caused by changes in GC size: “Together, these data show that Tfh cell restriction to the light zone through CXCR4 deletion can enhance the expansion of the FDC network and results in higher affinity GC responses without affecting the number of GC B cells or the area of the GC. This demonstrates that Tfh cell localisation and the expansion of the FDC network are entangled processes that can be both modulated by CXCR4 expression on T cells alone.”

Nevertheless, we agree that the reviewer’s question of whether direct Tfh-FDC interactions are occurring is an interesting one. To address this requires a reductionist approach, because the interactions of B cells with both Tfh cells and FDCs *in vivo* that cannot be separated without the whole GC reaction collapsing. Therefore *in vitro* co-culture approaches are required. However, in our hands, FDCs isolated from mice (either by FACS or CD35-bead enrichment) do not survive in culture long enough to perform these experiments (48hrs). This was unlike total lymph node fibroblastic reticular cells (FRC), that survive well in culture, and upregulated ICAM in response to co-culture with activated helper T cells (figure inset). This highlights that direct T cell-fibroblast interactions are able to influence the phenotype of lymph node stromal cells.

We recognise the need to ensure that our conclusions do not go beyond what the data show, and have carefully revised the manuscript to state that changes in Tfh cell localisation alter FDC expansion. Including the manuscript discussion, pages 21-22: “Our data demonstrate that FDC expansion into the GC requires Tfh cells to express CXCR5 which facilitates their co-localisation with the CXCL13-producing FDCs. This prompts the question of whether the interaction is direct, or whether it occurs via additional Tfh help to GC B cells who in turn promote FDC responses to vaccines. Disentangling these possibilities *in vivo* is experimentally challenging because deletion of any one of these cell types (Tfh cells, FDCs or GC B cells) results in GC collapse. Of note, we did not observe any changes in the number of GC B cells, or their phenotype, in our *Cxcr4^{fl/fl} Cd4^{cre/+}* mice after immunisation, only an increase in the affinity of those GC B cells. Thus, this study brings to light a previously unappreciated role for Tfh cells in helping the GC stroma upon vaccination, but the precise mechanism of how this help is given remains to be elucidated.”

4. The authors should assess whether the number and size of the steady state CD21/35+ FDC network is altered between adult and aged mice prior to vaccination, as this would help to demonstrate that the effect on FDC expansion is induced along with GC and Tfh formation.

We have quantified the CD35⁺ FDC area in the primary B cell follicle of younger adult and aged mice and find that in the primary follicles, the FDC network is also of reduced size in older animals (Extended Figure 4i, j). Which makes the ability of young T cells to enhance FDC expansion in old mice even more interesting. These data have been included in the results section on page 7: “Quantitative confocal imaging of iLN sections from adult and aged mice (Figure 2a and Extended Figure 3) revealed that the area of the mesenchyme-derived follicular dendritic cell (FDC) network within GCs of aged mice was significantly reduced compared to younger adult mice, resulting in larger dark zone areas and altered GC structure (Figure 2b-c). The reduced FDC area was also observed in aged mice prior to vaccination (Extended Figure 4).”

Extended Figure 4 | (i) Representative confocal images of iLNs at 10x magnification and GCs at 40x from unimmunised naive adult and aged C57BL/6 mice and **(j)** quantification of the CD35⁺ FDC network in the primary B cell follicle.

Minor comments:

1. The authors should clarify whether there are strain-related differences in the proportion of the GC occupied by the LZ and DZ and FDC yield. In Figure 3, the GC LZ:DZ is roughly 60:40% in adult BALB/c mice and 40:60% in aged BALB/c mice – a 20% shift in relative proportions associated with aging. Genetic perturbation experiments performed for Figures 4-7 employ C57BL/6 mice, with data in Figures 5-6 collected from adult mice showing a range in the proportion of LZ:DZ area in WT-equivalent mice of 50:50% in Figure 4 and 70:30% in Figure 5. The authors should provide a quantification of the LZ:DZ area in adult and aged C57BL/6 mice that serves as a comparative baseline for genetic perturbation studies that dissect the

requirement for CXCR4 and CXCR5 expression on FDC activation, and substantiate genotype-related differences in LZ:DZ GC organization between chemokine receptor-competent mice.

We have done the quantitative imaging in adult and aged C57BL/6 mice to serve as a baseline for the genetic perturbation studies, as requested. These data are included in Extended figure 4, and called out in the results text on page 7: “The mislocalisation of Tfh cells in ageing replicated in C57BL/6 mice (Extended Figure 4), indicating that this is a shared feature of ageing across genetic background.”

Extended Figure 4 | The spatial organisation of the GC is disrupted in aged C57BL/6 mice.

(a) Representative confocal images of iLNs at 10x magnification and GCs at 20x from adult and aged C57BL/6 mice 12 days after immunisation with NP-OVA in alum. Scale bars 100µm. LN sections were stained for IgD (green), CD35 (white), Ki67 (blue) and CD3 (magenta). (b, c) Enumeration of GC number (b) and area (c) per LN was performed by examining 6-10 sections throughout each LN and identifying GCs as CD35⁺Ki67⁺IgD⁻ structures. (d) Quantification of the CD35⁺ FDC network light zone area of the GC. (e) Quantification of the Ki67⁺CD35⁻ dark zone area of the GC. (f) Quantification of the number of CD3⁺ T cells identified within the total Ki67⁺CD35⁺IgD⁻ GC area. Quantification of the proportion of T cells positioned in the CD35⁺ FDC light zone area (g) and Ki67⁺CD35⁻ dark zone (h) of the GC. For (b-f), quantification of the GC compartments and T cell positioning was performed using an automated Cell Profiler pipeline for which 5-6 sections were analysed per LN. The data is representative of two independent experiments (n=4-8) where each symbol on

the graph represents a mouse and the bar height represents the median. The p-values were generated by performing an unpaired Mann Whitney U test.

2. The GC area appears to be reduced following the transfer of *Cxcr4*^{fl/fl} *Rosa26*^{ERT2Cre/+} OT-II T cells into adult mice in Figures 4g (and Extended Figure 7j), but not in *Cxcr4*^{fl/fl}; *CD4*^{Cre/+} mice. This should be quantified and any differences in GC size/output according to *Cxcr4*-deletion on transferred OT-II T cells as compared to endogenous *Cxcr4*-deficient T cells should be mentioned in the manuscript.

We have included the requested analyses in Figure 3h-o. These data show a non-significant trend to smaller GC area in CD45.1 recipient mice that received CXCR4-deficient OT-II T cells, and interestingly fewer T cells within the GC, potentially indicating that in the presence of WT host cells the CXCR4-deficient T cells have a competitive disadvantage to become Tfh cells. What is perhaps most striking about these data is that even though there are fewer T cells overall, we observed enhanced expansion of the FDC network when OT-II cells lack CXCR4. Which recapitulates the phenotype of the *Cxcr4*^{fl/fl}; *Cd4*^{cre/+} mice, despite a trend for smaller GCs.

Figure 3 | CXCR4 expression is increased in Tfh cells from aged mice and determines T cell dark zone positioning. (h) Experimental outline of *in vitro* 4-OH tamoxifen treatment of CD4⁺ T cells isolated from *Cxcr4*^{fl/fl}; *Rosa26*^{ERT2Cre/+} OT-II mice which were treated with 200nM of 4-OH tamoxifen for 48hours after which the cells were transferred into adult B6.SJL recipient mice. Recipient mice were immunised subcutaneously with OVA in alum and analysis was performed after 10 days. (g) Representative 20x magnification confocal images of the GCs from the iLNs of B6.SJL mice which received tamoxifen-treated OT-II cells from either *Cxcr4*^{fl/fl}; *Rosa26*^{+/+} or *Cxcr4*^{fl/fl}; *Rosa26*^{ERT2Cre/+} mice; scale bars are 100µm. LN sections were stained for IgD (green), CD35 (white), Ki67 (blue) and CD45.2 (magenta). Quantification of the (j) GC area, (k) number of CD45.2⁺ transferred cells per GC area, percentage of OT-II Tfh cells in the CD35⁺ FDC light zone area (l) and Ki67⁺CD35⁺ dark zone area (m) of the GCs, (n) percentage of the GC occupied by CD35⁺ FDC network and (o) percentage of the GC occupied by the dark zone, from the iLNs of recipient B6SJL mice (n=5). Data is representative of two independent experiments. Quantification of OT-II Tfh cell positioning was performed using an automated Cell Profiler pipeline for which 5-6 sections were analysed per GC. In bar graphs, bar heights represent the median and p-values were obtained by performing an unpaired Mann-Whitney U test. Each symbol represents a single mouse.

3. As histological quantification is used throughout the study, the side-by-side single channel images should be provided for:

- i) T cells in Figure 4g, Figure 5a, Figure 6a, Figure 7e,l, and corresponding Extended Figures as done for Figure 3a.
- ii) Stromal cell CXCL12 staining (Figure 4e, along with secondary Ab control)

We have updated manuscript to include side-by-side single channel images for every figure that shows imaging, including the extended figures.

4. The current selection of representative images used for quantification of GC T cell localization in Extended Figure 11a that lack a round, discriminate shape, making it difficult to see the LZ:DZ distribution of cells.

We have reviewed the images for this experiment, and they are indeed representative. These experiments were done in an independent laboratory in Germany, so the experimental set up is slightly different. But we believe that similar results obtained across countries with similar (but not identical) mouse strains supports the reproducibility of our findings.

5. The recovery of MAdCAM1+ CD21/35+ cells in chemokine receptor-competent (WT-equivalent) mice in Figure 5 and 6 is substantially distinct. Please comment on or address the difference in FDC yield between experiments.

Flow cytometric enumeration of stromal cells requires 3 rounds of enzymatic digestion to isolate the cells from tissues, and we have noted that there is experiment to experiment variation in this. In this manuscript, we have ensured that every experiment also quantifies the FDC network *in situ* using confocal imaging, to ensure that our conclusions are robust.

6. Extended data figures should appear in the order referred to in the text. For example, Extended Figure 11a-g, corresponding to Figure 7, are mentioned in relation to Figure 6, before Extended Figures 9 and 10 are referenced (manuscript page 15).

Thank you, this has been corrected.

Reviewer #3

Silva-Cayetano et. al. demonstrates that TFH cell localization within germinal centers is not only important for high affinity B cell responses via canonical B cell helper function but also for the expansion of FDC networks in GC light zones. Moreover, they suggest that this localization is defective in aging due to CXCR4-mediated mechanisms. These novel findings provide new insight into GC biology and have the potential for identifying targets for amplifying vaccine responses in older individuals. However, there are some gaps in these studies that need to be addressed in order to fully substantiate the authors' conclusions.

Main Comments:

The adoptive transfer experiments in Figure 7 nicely demonstrate that the blunted GC response in older mice can be rescued by CXCR5 expressing CD4 T cells. These data could imply at least two different things; (1) aged T cells don't express CXCR5 as well as young or (2) differences in CXCR4 in aged T cells prevents them from functioning properly. As CXCR4 over-expression by aged TFH cells is defined as a critical mediator of blunted GC responses, it would be of value to demonstrate that CXCR4 OE in young OT-II cells similarly fail to increase high affinity GC responses and FDC network size in an old mouse.

We appreciate these comments, and have done experiments specifically to address these two points.

(1) To determine whether CXCR5 expression changes with age we re-analysed our adult vs. aged immunisation experiments first identifying Tfh cells as PD-1+Bcl6+CD4+Foxp3⁻ cells so we could then assess the level of CXCR5 expression, and whether it changes with age. There was no difference in CXCR5 MFI on Tfh cells with age. This was consistent in both Balb/c and C57BL/6 mice, and with a range of immunogens. These data are presented in Extended Figure 7c-e, and called out in the results text on page 9: “Flow cytometric analysis revealed that Tfh cells from aged mice had more CXCR4⁺ Tfh cells than younger adult mice (Figure 3a-c), while CXCR5 expression on PD-1⁺Bcl6⁺CD4⁺Foxp3⁻ Tfh cells was unaltered by age (Extended figure 7c).”

Extended Figure 7 | (c) Median fluorescence intensity of CXCR4 on PD1⁺CXCR5⁺Foxp3⁻ Tfh cells and CXCR5 MFI on Bcl6⁺PD-1⁺Foxp3⁻ Tfh cells isolated from the draining iliac of adult and aged mice 14 days after immunisation with NP-KLH/Alum (n=8-10). **(d)** Median fluorescence intensity of CXCR4 on PD1⁺CXCR5⁺Foxp3⁻ Tfh cells and CXCR5 MFI on Bcl6⁺PD-1⁺Foxp3⁻ Tfh cells isolated from the draining iliac LN (left) and spleen (right) of adult and aged mice 9 days after immunisation with the Oxford/AstraZeneca COVID-19 vaccine candidate ChAOx1-nCoV19 (n=7). **(e)** Median fluorescence intensity of CXCR4 on PD1⁺CXCR5⁺Foxp3⁻ Tfh cells and CXCR5 MFI on Bcl6⁺PD-1⁺Foxp3⁻ Tfh cells isolated from the draining mediastinal LN (left) and spleen (right) of adult and aged mice 14 days after infection with influenza (n=5-6).

(2) As suggested, we used a retroviral system to overexpress CXCR4 in OT-II T cells from younger mice, these were adoptively transferred into congenically distinct mice who were then immunised and the response assessed ten days later. When we imaged the lymph nodes from these animals we observed that most of the T cells that overexpressed CXCR4 were located in the medullary sinus, and not in the T cell zone or the B cell follicle. CXCL12 is highly expressed at this site (<https://www.frontiersin.org/articles/10.3389/fimmu.2018.02196/full>), and these data suggest that when the T cells enter the lymph node and already have high CXCR4 expression they are drawn to the medulla, rather than using the balance of CCR7 and CXCR5 to draw them into the follicle. This biology limits our ability assess the impact that CXCR4 over expression in T cells has on the GC response. Representative lymph node images from this experiment are included here for the reviewer.

Authors state multiple times without the text that there is no difference in CXCL12 differences within LN across age. However, CXCL12 expression in LN staining is not obviously apparent from the images in Figure 4e. Indeed, it looks like there might be more expression within dark zone in older mice. Authors should better quantify these images, as well as provide single color images for to make it convincing that there is no CXCL12 difference in LN with age.

We agree with the reviewer that quantification and statistical analyses are key for drawing conclusions about CXCL12 in the GC. As suggested, the imaging is presented with side-by-side channels separated, and quantified using Cell Profiler. This analysis did not show a statistically significant difference in CXCL12 within the GC. These data are shown in Figure 3d, e and called out on page 9: “Immunofluorescence staining indicated that there was comparable expression of CXCL12 within the GCs of adult and aged mice (Figure 3d, e), suggesting that the increased dark zone positioning of Tfh cells with age is likely due to an increase in the expression of CXCR4 rather than increased ligand availability in the GC dark zone.”

Figure 3 | Representative 40x confocal images (**d**) and quantification (**e**) of CXCL12 in red within the dark zone of GCs in the iLNs of adult and aged mice at day 14 post-immunisation with NP-KLH; IgD (green), DAPI (blue) and CD35 (white). AIU=Arbitrary Intensity Units. Scale bars are 50µm.

In Extended Fig 6e, authors attempt to translate their findings on CXCR4 into human T cell aging, however they solely look at CXCR4 expression on memory CD4 populations. However, it is unclear if these findings translate into the human TFH population. Does CXCR4 expression within the peripheral TFH population change across age? These data would be more convincing that human TFH display similar age-related differences as mice.

We have performed analysis of a seasonal influenza vaccination cohort from younger and older adults and were able to show that CXCR4 expression was increased on circulating Tfh like cells from older people seven days after vaccination. These data are now included in Extended figure 7g. In addition, we were mindful of the reviewer's previous point about CXCR5 expression in ageing. We also included an analysis of CXCR5 expression in older persons, and as in mice, we do not see age-dependent differences in cell surface CXCR5 expression. These data are also included in Extended figure 7g, for completeness. The data are called out on page 9: "Enhanced CXCR4 expression was likewise seen in older people, with antigen-experienced CD4⁺ T cells from unvaccinated people over 65 years of age having increased surface CXCR4 expression compared to 18-36-year-old adults (Extended Figure 7f). The expression of CXCR4 was also higher on circulating Tfh-like cells from older people seven days after seasonal influenza vaccination compared to younger individuals, with CXCR5 expression being consistent between the age groups (Extended figure 7g)."

Extended Figure 7 | (g) Flow cytometric quantification of the %CXCR4⁺ cells and CXCR4 MFI from CXCR5⁺CD45RA⁺CD4⁺ circulating Tfh cells, and %CXCR5⁺ cells and CXCR5 MFI on ICOS⁺CD38⁺CD45RA⁺CD4⁺ circulating Tfh cells from human peripheral blood mononuclear cells seven days after seasonal influenza vaccination (n=16-18 per group).

Other Comments:

Authors conclude that CXCR4 expression regulate migration into the dark zone and size of FDC networks. Although there is an obvious preferential bias of CXCR4 KO OT-II in the light zone, in these experiments it also appears that there is an overall decrease of in CXCR4 KO T cells in the LN overall. (Figure 4g-l) Authors should explain these findings and what role CXCR4 might play in TFH development and migration overall.

We have included the requested analyses in Figure 3h-o. These data show a non-significant trend to smaller GC area in CD45.1 recipient mice that received CXCR4-deficient OT-II T cells, and interestingly significantly fewer T cells within the GC, potentially indicating that in the presence of WT host cells the CXCR4-deficient T cells have a competitive disadvantage to become Tfh cells (as there are normal numbers of GC T cells when all T cells lack CXCR4, shown in Figure 4). What is perhaps most striking about these data is that even though there are fewer T cells overall, we observed enhanced expansion of the FDC network when OT-II cells lack CXCR4. Which recapitulates the phenotype of the *Cxcr4^{fl/fl}; Cd4^{cre/+}* mice, despite a trend for smaller GCs.

Figure 3 | CXCR4 expression is increased in Tfh cells from aged mice and determines T cell dark zone positioning. (h) Experimental outline of *in vitro* 4-OH tamoxifen treatment of CD4⁺ T cells isolated from *Cxcr4^{fl/fl}; Rosa26^{ERT2Cre/+}* OT-II mice which were treated with 200nM of 4-OH tamoxifen for 48hours after which the cells were transferred into adult B6.SJL recipient mice. Recipient mice were immunised subcutaneously with OVA in alum and analysis was performed after 10 days. (g) Representative 20x magnification confocal images of the GCs from the iLNs of B6.SJL mice which received tamoxifen-treated OT-II cells from either *Cxcr4^{fl/fl}; Rosa26^{+/+}* or *Cxcr4^{fl/fl}; Rosa26^{ERT2Cre/+}* mice; scale bars are 100µm. LN sections were stained for IgD (green), CD35 (white), Ki67 (blue) and CD45.2 (magenta). Quantification of the (j) GC area, (k) number of CD45.2⁺ transferred cells per GC area, percentage of OT-II Tfh cells in the CD35⁺ FDC light zone area (l) and Ki67⁺CD35⁺ dark zone area (m) of the GCs, (n) percentage of the GC occupied by CD35⁺ FDC network and (o) percentage of the GC occupied by the dark zone, from the iLNs of recipient B6SJL mice (n=5). Data is representative of two independent experiments. Quantification of OT-II Tfh cell positioning was performed using an automated Cell Profiler pipeline for which 5-6 sections were analysed per GC. In bar graphs, bar heights represent the median and p-values were obtained by performing an unpaired Mann-Whitney U test. Each symbol represents a single mouse.

The reduced *myc*-GFP⁺ GC B cells in older mice appears bi-modal in Figure 2d thus any conclusions about these data are unclear. Authors need to confirm if these results were due to technical error or underlying heterogeneity of aging.

Ageing is heterogenous, and to have a better picture of whether there is a bimodal distribution of *c-Myc* expression in the data shown in original figure 2d, we have combined the discovery (black symbols) and replication (white symbols) experiments into one graph. This shows that the distribution is similar between experiments, indicating that this is likely represents normal biological variation. It highlights that the proportion of GC B cells that express *Myc* is heterogenous, but is significantly lower in the aged lymph node. These re-graphed data have been included in figure 1 (Original figure 1 and 2 were combined at the request of reviewer 1).

Figure 1 | (i) Experimental outline of the transfer of B1.8ⁱ c-myc^{GFP/GFP} B cells from adult donors into adult or aged C57BL/6 recipient mice 10 days after immunisation with NP-OVA in alum. Quantification of the frequency **(j)** of B1.8i derived c-myc⁺ GC B cells in adult or aged mice, data are pooled from two independent experiments, first experiment in black, second experiment in white.

CXCR5 expression is induced by TCR cell activation, which is blunted in human T cells with age. Authors should discuss this in light of their findings. Is it possible that T cells just have reduced CXCR5 with age (independent of CXCR4 expression), and thus are less effective at getting into the GC and helping FDCs out? (see main comment above).

We have added new data in both mice and humans to the manuscript to address this point, as outlined above. In addition, we have previously shown that naïve flow-sorted T cells from older people have comparable CXCR5 upregulation to cells from younger donors in vitro after CD3/28 bead stimulation (Webb et al., Aging Cell, 2021). We have added a sentence to the discussion on page 21 to highlight these findings, and include them in our interpretation of the data. “while CXCR5 expression on CD4 T cells is unaltered by age (Extended Figure 7 and reference⁵⁸).”

Decision Letter, first revision:

17th Mar 2023

Dear Michelle,

Thank you for submitting your revised manuscript "Spatial dysregulation of T follicular helper cells impairs vaccine responses in ageing" (NI-A34173B). It has now been seen by the original referees and their comments are below. The reviewers find that the paper has improved in revision, and therefore we'll be happy in principle to publish it in Nature Immunology, pending minor revisions to to comply with our editorial and formatting guidelines.

We will now perform detailed checks on your paper and will send you a checklist detailing our editorial and formatting requirements in about a week. Please do not upload the final materials and make any revisions until you receive this additional information from us.

If you had not uploaded a Word file for the current version of the manuscript, we will need one before beginning the editing process; please email that to immunology@us.nature.com at your earliest convenience.

Thank you again for your interest in Nature Immunology Please do not hesitate to contact me if you have any questions.

Kind regards,

Laurie

Laurie A. Dempsey, Ph.D.
Senior Editor
Nature Immunology
l.dempsey@us.nature.com
ORCID: 0000-0002-3304-796X

Reviewer #2 (Remarks to the Author):

The reviewer is satisfied with the revised manuscript.

The additional data demonstrating the rescued defect of humoral immunity following the transfer of adult T cells into aged mice and validation experiments using heterozygous Cxcr4 fl/- and Cxcr5 fl/- mice help to substantiate the major claims of the study. Moreover, the new discussion paragraph incorporating the findings from Denton et al. and Silva-Cayetano et al, offer added insight as to which aspects of the age-related defects in humoral immunity are attributed to FRCs and which can be rescued by adult T cells.

Reviewer #3 (Remarks to the Author):

Overall, the revised manuscript by Silva-Cayetano et. al. addressed my previous concerns and was of excellent caliber. The added studies provided significantly stronger evidence for their novel conclusion that mislocalization of TFH via CXCR4 blunts FDC expansion and GC responses with age, and offers new insight into the complex nature of TFH-FDC-B cell interactions. Moreover, they provide correlative evidence that the same mechanisms may be occurring in humans, which holds interesting translational implications.

Claire E. Gustafson

Final Decision Letter:

Dear Dr. Linterman,

I am delighted to accept your manuscript entitled "Spatial dysregulation of T follicular helper cells impairs vaccine responses in ageing" for publication in an upcoming issue of Nature Immunology.

Over the next few weeks, your paper will be copyedited to ensure that it conforms to Nature Immunology style. Once your paper is typeset, you will receive an email with a link to choose the appropriate publishing options for your paper and our Author Services team will be in touch regarding any additional information that may be required.

Please note that *Nature Immunology* is a Transformative Journal (TJ). Authors may publish their research with us through the traditional subscription access route or make their paper immediately open access through payment of an article-processing charge (APC). Authors will not be required to make a final decision about access to their article until it has been accepted. [Find out more about Transformative Journals](https://www.springernature.com/gp/open-research/transformative-journals).

Authors may need to take specific actions to achieve > **compliance with funder and institutional open access mandates**. If your research is supported by a funder that requires immediate open access (e.g. according to [Plan S principles](https://www.springernature.com/gp/open-research/plan-s-compliance)) then you should select the gold OA route, and we will direct you to the compliant route where possible. For authors selecting the subscription publication route, the journal's standard licensing terms will need to be accepted, including [self-archiving policies](https://www.springernature.com/gp/open-research/policies/journal-policies). Those licensing terms will supersede any other terms that the author or any third party may assert apply to any version of the manuscript.

Your paper will be published online soon after we receive your corrections and will appear in print in the next available issue. Content is published online weekly on Mondays and Thursdays, and the embargo is set at 16:00 London time (GMT)/11:00 am US Eastern time (EST) on the day of publication. Now is the time to inform your Public Relations or Press Office about your paper, as they might be interested in promoting its publication. This will allow them time to prepare an accurate and satisfactory press release. Include your manuscript tracking number (NI-A34173C) and the name of the journal, which they will need when they contact our office.

About one week before your paper is published online, we shall be distributing a press release to news organizations worldwide, which may very well include details of your work. We are happy for your institution or funding agency to prepare its own press release, but it must mention the embargo date and Nature Immunology. Our Press Office will contact you closer to the time of publication, but if you or your Press Office have any enquiries in the meantime, please contact press@nature.com.

Also, if you have any spectacular or outstanding figures or graphics associated with your manuscript - though not necessarily included with your submission - we'd be delighted to consider them as candidates for our cover. Simply send an electronic version (accompanied by a hard copy) to us with a possible cover caption enclosed.

If you have not already done so, we strongly recommend that you upload the step-by-step protocols used in this manuscript to the Protocol Exchange. Protocol Exchange is an open online resource that allows researchers to share their detailed experimental know-how. All uploaded protocols are made freely available, assigned DOIs for ease of citation and fully searchable through nature.com. Protocols

can be linked to any publications in which they are used and will be linked to from your article. You can also establish a dedicated page to collect all your lab Protocols. By uploading your Protocols to Protocol Exchange, you are enabling researchers to more readily reproduce or adapt the methodology you use, as well as increasing the visibility of your protocols and papers. Upload your Protocols at www.nature.com/protocolexchange/. Further information can be found at www.nature.com/protocolexchange/about .

Please note that we encourage the authors to self-archive their manuscript (the accepted version before copy editing) in their institutional repository, and in their funders' archives, six months after publication. Nature Portfolio recognizes the efforts of funding bodies to increase access of the research they fund, and strongly encourages authors to participate in such efforts. For information about our editorial policy, including license agreement and author copyright, please visit www.nature.com/ni/about/ed_policies/index.html

Sincerely,

Laurie A. Dempsey, Ph.D.
Senior Editor
Nature Immunology
l.dempsey@us.nature.com
ORCID: 0000-0002-3304-796X